# UNDERSTANDING LEARNING WITH SLICED-WASSERSTEIN REQUIRES RE-THINKING INFORMATIVE SLICES

## ABSTRACT

The practical applications of Wasserstein distances (WDs) are constrained by their sample and computational complexities. Sliced-Wasserstein distances (SWDs) provide a workaround by projecting distributions onto one-dimensional subspaces, leveraging the more efficient, closed-form WDs for one-dimensional distributions. However, in high dimensions, most random projections become uninformative due to the concentration of measure phenomenon. Although several SWD variants have been proposed to focus on *informative* slices, they often introduce additional complexity, numerical instability, and compromise desirable theoretical (metric) properties of SWD. Amidst the growing literature that focuses on directly modifying the slicing distribution, which often face challenges, we revisit the classic Sliced-Wasserstein and propose instead to rescale the 1D Wasserstein to make all slices equally informative. Importantly, we show that with an appropriate notion of *slice informativeness*, rescaling for all individual slices simplifies to **a single global scaling factor** on the SWD. This, in turn, translates to the standard learning rate search for gradient-based learning in common ML workflows. We perform extensive experiments across various machine learning tasks showing that the classic SWD, when properly configured, can often match or surpass the performance of more complex variants. We then answer the following question:

*Is Sliced-Wasserstein all you need for common learning tasks?*

## 1 INTRODUCTION

Data representation in machine learning involves encoding the unique characteristics of individual data points and capturing the relationships between them, with a focus on optimizing performance for specific downstream tasks. Optimal transport (OT) theory (Villani et al., 2009; Peyré et al., 2019) compares data distributions by finding an optimal transportation plan that minimizes the expected cost of moving mass between them, leading to the popular Wasserstein distance (WD) central to many learning applications (Khamis et al., 2024). However, the computational complexity of OT solvers poses a significant bottleneck when calculating the WD. In cases of discrete measures or sample-based scenarios, which are common in machine learning, the problem typically reduces to linear programming with time complexity $\mathcal{O}(N^3 \log N)$, space complexity $\mathcal{O}(N^2)$, and sample complexity $\mathcal{O}(N^{-\frac{1}{d}})$, where $N$ is the number of support points and $d$ the data dimensionality. These unfavorable scaling properties, particularly the curse of dimensionality in sample complexity, make WD impractical for many real-world applications. To address these challenges, several approaches have been proposed, including entropic regularized OT (Cuturi, 2013), smooth OT (Blondel et al., 2018; Manole et al., 2024)), and sliced OT (Bonneel et al., 2015).

The Sliced-Wasserstein distances (SWD), (Rabin et al., 2012; Bonneel et al., 2015) project high-dimensional distributions onto 1D subspaces and aggregate the closed-form OT solutions in these subspaces. This method is particularly attractive because 1D Wasserstein distances can be computed efficiently with a time complexity of $\mathcal{O}(N \log N)$ and a space complexity of $\mathcal{O}(N)$ for discrete measures. Additionally, SWD provides a metric between probability distributions that retains many desirable properties of the Wasserstein distance (WD), such as being statistically and topologically equivalent to WD, while being more computationally tractable (Nadjahi et al., 2020). Notably, with a sample complexity of $\mathcal{O}(N^{-\frac{1}{2}})$, SWD avoids the curse of dimensionality. However, a key drawback of SWD is its projection complexity, which requires exponentially more slices as the data dimensionality increases.

The projection complexity of SWD has motivated several lines of work that aim to enhance the effectiveness of the slicing approach, especially in addressing variance reduction (Nguyen & Ho, 2023), approximation error reduction (Nguyen et al., 2023), and slicing complexity (Kolouri et al., 2019; Deshpande et al., 2019; Nguyen et al., 2020; 2024a; Nguyen & Ho, 2024; Nguyen et al., 2024b). This is particularly relevant in high-dimensional machine learning settings where data often has supports in low-dimensional subspaces. These SW variants are data-driven, focusing on identifying the most informative slices for capturing distributional differences in the data. For instance, Max-SW (Deshpande et al., 2019) and DSW (Nguyen et al., 2020) seek to find slices/projections that maximize the differences between the data distributions. GSW (Kolouri et al., 2019) and ASW (Chen et al., 2020) extend SW by allowing 'non-linear' projections to capture complex data structures. EBSW (Nguyen & Ho, 2024) designs an energy-based slicing distribution that is parameter-free and has the density proportional to an energy function of the projected 1D distance. MSW (Nguyen et al., 2024a) imposes a first-order Markov structure to avoid redundant, independent projections. More recently, RPSW (Nguyen et al., 2024b) proposes using the normalized differences between random samples from the two distributions to ensure that the projections are sampled from the subspace in which the data resides. These methods improve the performance of SW in various downstream tasks and have significantly expanded the tools at disposal for both researchers and practitioners alike. Nonetheless, the elegant extensions also come with increased computational cost, numerical instability, complicated design choices, and often losing the metricity of the SWD.

In this paper, we argue that the standard SW, with proper hyperparameters, can often match or surpass the performance of more complex variants in many learning tasks while retaining its simplicity and theoretical guarantees. Our key insight is that when $d$-dimensional data have $k$-dimensional supports, where $k \ll d$, almost all random slices $\theta \sim \mathcal{U}(\mathbb{S}^{d-1})$ can be decomposed into an *informative* component $\theta_D \in \mathbb{R}^k$ within the data subspace and its orthogonal complement $\theta_D^\perp \in \mathbb{R}^{d-k}$. This implies most slices still carry relevant information for distinguishing distributions, proportional to $\|\theta_D\|$. By appropriately scaling the distance per slice, we get better gradient for learning. In expectation, we show that, with our defined notion of *informativeness*, scaling for all slices (based on their informativeness) simplifies to scaling the SWD by a single scalar factor. In gradient-based learning, this means finding an appropriate learning rate is equivalent to getting informative slices **for free**. This allows the classical SWD to adapt to the data's intrinsic dimensionality without explicitly limiting the computation to the subspace. We provide theoretical justification and empirical evidence, offering a fresh perspective on SW, particularly in high-dimensional settings.

By revisiting the celebrated SW with these insights, we aim to elucidate the performance gap between the original formulation and recent variants in the existing literature. We emphasize that our work does not diminish the valuable contributions of these variants, which have greatly advanced our understanding of Sliced-Wasserstein. Rather, we offer a complementary perspective that highlights the potential of the standard SW when properly integrated into learning tasks. Along that line, we remark that the related line of specialized methods that respects the data geometry (Rabin et al., 2011; Bonet et al., 2022; Martin et al., 2023; Quellmalz et al., 2023; Bonet et al., 2024; Tran et al., 2024) remains valuable when the manifold constraint on the data is readily known.

In common ML settings where data is supported, or nearly supported, on a $k$-dimensional subspace embedded in a $d$-dimensional space, our specific contributions can be summarized as follows:

- We introduce the $\phi$-weighting formulation unifying various SW variants. In this framework, we propose reweighing all one-dimensional Wasserstein distances based on *slice informativeness* instead of directly modifying the slicing distribution, as commonly done in the literature. We show that with an appropriate notion of *slice informativeness*, in expectation, this leads to an equivalence between the SWD in the ambient space and the data effective subspace. (See 20).
- Our findings translate to scaling the classic SW by a single global constant to get better learning gradients. We show that this reduces solving the problem of *non-informative slices* to the learning rate search for the classic SW, a process that is already a standard in ML workflows. In other words, we get *informative slices* for free with the classic SW.
- We perform a comprehensive learning rate sweep across a wide range of experiments, including gradient flow (on 3 classic toy datasets, MNIST images, CelebA images), color transfer (3 sets of images), deep generative modeling on the FFHQ dataset (unconditional generation and unpaired translation with SW). We show that the classic SW, with appropriate hyperpamers, perform competitively with more advanced methods in these settings.

**Notations.** We let $\mathbb{R}^d$ denote a $d$-dimensional inner product space, and we denote the unit hypersphere in this space by $\mathbb{S}^{d-1} = \{\theta \in \mathbb{R}^d : \|\theta\|_2 = 1\}$. Additionally, we denote by $\mathcal{P}(\mathbb{R}^d)$ the set of probability measures on $\mathbb{R}^d$ endowed with the $\sigma$-algebra of Borel sets, and by $\mathcal{P}_p(\mathbb{R}^d) \subset \mathcal{P}(\mathbb{R}^d)$ the subset of those measures with finite $p$-th moments. For a measurable function $f : \mathbb{R}^d \to \mathbb{R}$ defined by $f(x) = \theta^\top x$ such that $\theta \in \mathbb{S}^{d-1}$, we denote the pushforward of a measure $\mu \in \mathcal{P}(\mathbb{R}^d)$ through $f$ as $f_\# \mu$. Particularly, $\theta_\# \mu$ is the pushforward measure of $\mu$ under the projection $x \mapsto \theta^\top x$.

## 2 BACKGROUND ON SLICED-WASSERSTEIN

Let $\mu \in \mathcal{P}_p(\mathbb{R}^d)$ and $\nu \in \mathcal{P}_p(\mathbb{R}^d)$ be two probability measures of interest.

**The Wasserstein distance (WD).** The p-WD between $\mu$ and $\nu$ is:

$$W_p^p(\mu, \nu) = \inf_{\pi \in \Pi(\mu, \nu)} \int_{\mathbb{R}^d \times \mathbb{R}^d} \|x - y\|_p^p \, d\pi(x, y), \tag{1}$$

with $\Pi(\mu, \nu) = \{\pi \in \mathcal{P}_p(\mathbb{R}^d \times \mathbb{R}^d) : \pi(A \times \mathbb{R}^d) = \mu(A), \quad \pi(\mathbb{R}^d \times A) = \nu(A)\}$ for all measurable sets $A \subset \mathbb{R}^d$. In one dimension ($d = 1$), the p-WD admits the following closed-form solution:

$$W_p^p(\mu, \nu) = \int_0^1 |F_\mu^{-1}(z) - F_\nu^{-1}(z)|^p \, dz, \tag{2}$$

where $F_\mu, F_\nu$ are the cumulative distribution functions (CDF) of $\mu$ and $\nu$, respectively. For empirical measures, 2 becomes a Monte Carlo sum that can be calculated by averaging the $d^p(\cdot, \cdot)$ between sorted samples. In general, this translates to a highly favorable time complexity of $\mathcal{O}(N \log N)$ and gives rise to the following Sliced-Wasserstein distance.

**Sliced-Wasserstein (SW).** The SW distance between $\mu$ and $\nu$ is defined as:

$$SW_p(\mu, \nu; \sigma) := \left( \mathbb{E}_{\theta \sim \sigma} \left[ W_p^p(\theta_\# \mu, \theta_\# \nu) \right] \right)^{\frac{1}{p}} \tag{3}$$

where $\sigma \in \mathcal{P}(\mathbb{S}^{d-1})$ is the reference measure for slicing vector $\theta$. In default setting, $\sigma$ is set to be uniform distribution, denoted as: $\sigma = \mathcal{U}(\mathbb{S}^{d-1})$ and we use $SW_p(\mu, \nu)$ to denote $SW_p(\mu, \nu; \sigma)$ for simplicity. The intractable expectation implies (3) admits a Monte Carlo estimator:

$$SW_p\left(\mu, \nu; \sum_{l=1}^{L} \frac{1}{L} \delta_{\theta_l}\right) = \left( \frac{1}{L} \sum_{l=1}^{L} W_p^p(\theta_\#^l \mu, \theta_\#^l \nu) \right)^{\frac{1}{p}}, \tag{4}$$

where $\{\theta_l\}_{l=1}^{L} \overset{\text{i.i.d.}}{\sim} \sigma$. The MC scheme has the estimation error decreases as $\frac{1}{\sqrt{L}}$ where $L$ is the number of slices. The main issue becomes how much one can simulate (for large $d$), which proves to be challenging since most slices are known to be non-informative. As a result, $SW_p(\mu, \nu; \sum_{l=1}^{L} \frac{1}{L} \delta_{\theta_l})$ often underestimates the distance between $\mu$ and $\nu$ in practice. Moreover, $L$ should be sufficiently large compared to $d$, which is undesirable since the time complexity of SW scales linearly with $L$.

## 3 OTHER RELATED WORK

**Subspace-constrained Optimal Transport.** Recent works propose computing optimal transport (OT) in lower-dimensional subspaces (Paty & Cuturi, 2019; Bonet et al., 2021b; Muzellec & Cuturi, 2019) to improve both efficiency and robustness for high-dimensional data. **1) Subspace Detours** (Bonet et al., 2021b; Muzellec & Cuturi, 2019) constrain transport plans to be optimal when projected onto a chosen subspace. This enables efficient extension of low-dimensional transport solutions to the full space. **2) Subspace Robust Wasserstein** (Paty & Cuturi, 2019) considers the worst-case transport cost over all possible low-dimensional projections. Interestingly, this can be computed by minimizing the sum of the $k$ largest eigenvalues of the transport plan second-order moment matrix $S_k^2(\mu, \nu) = \min_{\pi \in \Pi(\mu, \nu)} \sum_{l=1}^{k} \lambda_l(V_\pi)$ where $V_\pi := \int (x - y)(x - y)^T d\pi(x, y)$ is the second-order displacement matrix for a coupling $\pi$, and $\lambda_l(V_\pi)$ is its $l$-th largest eigenvalue.

**Gaussian Sliced-Wasserstein.** Earlier works ((Sudakov, 1978; Diaconis & Freedman, 1984; Reeves, 2017)) establish several central limit theorems showing that under mild conditions, low-dimensional projections of high-dimensional data converge to Gaussians. Nadjahi et al. (2021) leverages this concentration of measure phenomenon and shows the Gaussian SW distance is equivalent to the classical SWD: $SW_p^p(\mu, \nu; \mathcal{N}(0, \frac{1}{d}I_d)) = C_{d,p} SW_p^p(\mu, \nu; \mathcal{U}(\mathbb{S}^{d-1}))$, where $C_{d,p}$ is a dimensionality-dependent constant. They then propose an efficient approximation of the SWD without simulation.

## 4 REVISITING SLICED WASSERSTEIN DISTANCES: A SUBSPACE PERSPECTIVE

**The main challenge.** Many machine learning problems involve high-dimensional data that has a low-dimensional structure. Formally, this phenomenon, known as the *manifold hypothesis*, states that for a dataset $X \subset \mathbb{R}^d$, there exists a $k$-dimensional manifold $\mathcal{M}$ where $k \ll d$ such that $X$ approximately lies on $\mathcal{M}$ (Fefferman et al., 2016). For instance, rigorous dimensionality estimation methods applied to common datasets like MS-COCO (Lin et al., 2014) and ImageNet (Deng et al., 2009) suggest $k < 50$ (Pope et al., 2021), despite their ambient dimension $d$ being orders of magnitude larger. While these manifolds are generally nonlinear, they admit local linear approximations via their tangent spaces. Moreover, in practice, data features typically have strong linear correlations, allowing techniques like Principal Component Analysis (PCA) to

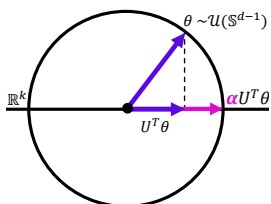

Figure 1: Rescaling the 1D Wasserstein based on slice *informativeness*.

identify a principal subspace that captures most of the data variance. This subspace approximation is particularly relevant in the context of Sliced-Wasserstein distance (SWD). It is known from Kolouri et al. (2019) that when slices $\theta$ are sampled uniformly from $\mathbb{S}^{d-1}$, the probability that a random slice is nearly orthogonal to any fixed direction increases exponentially with dimension. Specifically, for a unit vector $x_0$ representing a principal direction in the data subspace:

$$\Pr\left(|\langle \theta, x_0 \rangle| \leq \epsilon\right) > 1 - e^{-d\epsilon^2}, \quad \theta \sim \mathcal{U}(\mathbb{S}^{d-1}). \tag{5}$$

This concentration of measure phenomenon implies that as dimensionality $d$ grows, most random slices become nearly orthogonal to the principal directions of the data subspace. Consequently, the corresponding 1D Wasserstein distances contribute minimally to the SWD. This effect, which we refer to as the *slice non-informativeness*, limits the effectiveness of SWD in high-dimensional spaces.

**Current approaches: Designing the slicing distribution.** Sampling-based methods seek to define a non-uniform slicing distribution that focuses on *discriminative* directions. Optimization-free methods (Nguyen & Ho, 2024; Nguyen et al., 2024b) are objectively faster but do not yield true metrics. Other methods (Nguyen et al., 2020; 2024a) yield proper metrics but are more computationally expensive due to the optimization involved. In the limit, the Max variants use discrete slicing distributions that require global optimality to be metrics, which is generally intractable in practice. Empirically, without careful hyperparameter tuning, the different variants face numerical instability in the larger learning rate regimes, likely because of the overemphasizing on directions with large projected distances.

**A novel perspective: Rescaling 1D Wasserstein distances.** These challenges in directly redefining the slicing distribution motivates us to take a second look at the conventional wisdom of sampling informative slices. We propose an alternative formulation that reweights each 1D Wasserstein based on the *informativeness* of the corresponding slice/projecting direction (See Figure 1 for illustration). By defining the notion of an *informative slice* based on its alignment with the effective data subspace, we demonstrate that it is possible to reweight for all slices by *a global constant* on the SWD. This maintains the efficiency and theoretical properties of the classical Sliced-Wasserstein. The implications of this finding for using SWD in gradient-based learning will be discussed subsequently. To formalize this approach, we introduce the following assumption and definitions:

**Assumption 4.1** (Effective Subspace Structure). *Let $\mu^d, \nu^d \in P(\mathbb{R}^d)$ be probability measures. We say $(\mu^d, \nu^d)$ has $k$-dimensional effective structure if:*

   *1. There exists a semi-orthogonal matrix $U \in \mathbb{R}^{d \times k}$ (i.e., $U^T U = I_k$) such that*

   $$supp(\mu^d), supp(\nu^d) \subset V_k := col\text{-}span(U).$$

   *2. $k$ is minimal, meaning that there does not exist any $U' \in \mathbb{R}^{d \times k'}$ with $k' < k$ s.t (1) holds.*

*We refer to $V_k$ as the **effective subspace** (ES) of $\mu^d, \nu^d$, and $k$ as their **effective dimensionality** (ED).*

*Note:* In the Appendix A.9, we also discuss results when this assumption does not hold.

**Definition 4.2** (Informative slices). *Let $\phi : \mathbb{S}^{d-1} \to \mathbb{R}_+$ be a function that assigns importance values to projection directions $\theta \in \mathbb{S}^{d-1}$ based on their relevance in comparing data distributions. Different approaches compute this importance in various ways. For instance, Max-SW (Deshpande et al., 2019), Markovian SW Nguyen et al. (2024a), and EBSW (Nguyen & Ho, 2024) implicitly use*

$$\phi_{\mu,\nu}(\theta) = W_p^p(\theta_\# \mu, \theta_\# \nu), \tag{6}$$

*to measure the informativeness of $\theta$. On the other hand, RPSW (Nguyen et al., 2024b) implicitly uses*

$$\phi_{\mu,\nu}(\theta; \mu, \nu, \gamma_\kappa) = \mathbb{E}_{(X,Y)\sim\mu\times\nu}[\gamma_\kappa(\theta; P_{\mathbb{S}^{d-1}}(X - Y))], \tag{7}$$

*where $\gamma_\kappa$ is a location-scale distribution (e.g., vMF) and $P_{\mathbb{S}^{d-1}}$ is the projection onto $\mathbb{S}^{d-1}$.*

*One may also refer to $\phi_{\mu,\nu}$ as the **discriminant function**. However, we define informativeness more broadly, allowing for a broader set of assumptions about data structure. Other ways to quantify informativeness may be appropriate depending on the context where prior information on the data is available. We refer to further related details in Remark 4.6.*

Defining $\phi$ in terms of 1D Wassersteins may not always be desirable. It requires calculating them to find out how informative the slices are, even when the calculations are not always used in computing the final distances (Nguyen et al., 2024a). Furthermore, defining $\phi$ based on the input measures $\mu, \nu$ could introduce complex dependencies that make the triangle inequality difficult to prove (Nguyen & Ho, 2024; Nguyen et al., 2024b). Motivated by 4.1, we propose a principled notion of informativeness that avoids both issues and leads to a significantly simplified solution.

**Definition 4.3** (ES-aligned informative slices). *Given $V_k = span(U)$, where $U \in \mathbb{R}^{d\times k}$ is an orthogonal matrix, we define the ES-aligned informative function $\phi_U : \mathbb{S}^{d-1} \to [0,1]$ as:*

$$\phi_U(\theta) = \|U^\top \theta\|. \tag{8}$$

*Intuitively speaking, $\phi_U$ corresponds to how much information $\theta$ contains about the data if it is projected into the space spanned by $U$. Higher $\phi_U(\theta)$ is considered more (ES-aligned) informative.*

**Remark 4.4.** *$\phi_U(\theta)$ has the following basic properties: **a)** $0 \leq \varphi_U(\theta) \leq 1$ for all $\theta \in \mathbb{S}^{d-1}$, **b)** $\varphi_U(\theta) = 1$ iff $\theta \in span(U) \cap \mathbb{S}^{d-1}$ and $\varphi_U(\theta) = 0$ iff $\theta \perp span(U)$, and **c)** For any orthogonal matrix $Q \in \mathbb{R}^{k\times k}$, $\varphi_U(\theta) = \varphi_{UQ}(\theta)$.*

### 4.1 THE $\phi$-WEIGHTING FORMULATION

Starting from the classical SWD definition in Equation (3), we propose a general framework for reweighting slice contributions:

$$\widetilde{SW}_p(\mu, \nu; \sigma, \rho_\phi) = \left( \int_{\mathbb{S}^{d-1}} \underbrace{\rho_\phi(\phi(\theta))W_p^p(\theta_\#\mu, \theta_\#\nu)}_{\text{Reweighted contribution}} d\sigma(\theta) \right)^{\frac{1}{p}}, \tag{9}$$

where $\rho_\phi : [0,1] \to \mathbb{R}_+$ is the $\phi$-**weighting function** that rescales the contribution of each slice.

**Example 4.5.** *If the goal is to make all slices informative, an appropriate choice for $\rho_\phi$ can be the multiplicative inverse of $\phi(\theta)$ (i.e., more informative slices are scaled less). That is,*

$$\rho_\phi(\phi(\theta)) = \begin{cases} \dfrac{1}{\phi(\theta)^p}, & \text{if } \phi(\theta) > 0, \\ 0, & \text{if } \phi(\theta) = 0. \end{cases} \tag{10}$$

**Remark 4.6.** *Equation (9) notably does not rely on Assumption 4.1 (only our choice of $\phi(\cdot) = \phi_U(\cdot)$ does). By defining the appropriate $\rho_\phi(\cdot)$ and $\phi(\cdot)$, the $\phi$-weighting formulation can be seen as a unifying formulation that recovers different SW variants.*

- *We set $\phi \equiv 1$ and obtain the classical Sliced-Wasserstein distance.*

- *We set $\phi_{\mu,\nu}(\theta) = W_p^p(\theta_\#\mu, \theta_\#\nu)$ and $\rho_\phi(r) = \delta_{r_{max}}, \sigma = \mathcal{U}(\mathbb{S}^{d-1})$ where $r_{max} = \max \phi_{\mu,\nu}(\theta)$, and recover Max-SW (Deshpande et al., 2019).*

- *We set $\phi_{\mu,\nu}(\theta) = W_p^p(\theta_\#\mu, \theta_\#\nu)$, $\rho_\phi(r) = \frac{f(r)}{\int_{\mathbb{S}^{d-1}} f(W_p^p(\theta_\#\mu,\theta_\#\nu))d\sigma(\theta)}$, $\sigma = \mathcal{U}(\mathbb{S}^{d-1})$ where $f : [0,\infty) \to (0,\infty)$ is an increasing energy function (e.g., $f(x) = e^x$), and recover EBSW (Nguyen & Ho, 2024).*

- *We set $\phi_{\mu,\nu}(\theta) = \mathbb{E}_{(X,Y)\sim\mu\times\nu}[\gamma_\kappa(\theta; P_{\mathbb{S}^{d-1}}(X - Y))], \rho_\phi(r) = r, \sigma = \mathcal{U}(\mathbb{S}^{d-1})$, where $\gamma_\kappa$ is a location-scale distribution with parameter $\kappa$, and recover RPSW (Nguyen et al., 2024b).*

## 4.2 MISALIGNED RANDOM PROJECTIONS ARE IMPLICITLY DOWNWEIGHED BY A SCALAR

Under Assumption 4.1, we will show that the 1D Wasserstein corresponding to each random projection is weighted by a scalar related to the (ES-aligned) informativeness of that projection.

**The case for 1D effective subspaces.** Let $V_1 = \text{span}(u)$ where $u \in \mathbb{S}^{d-1}$, and suppose $\text{supp}(\mu^d), \text{supp}(\nu^d) \subset V_1$. Given $\theta \in \mathbb{S}^{d-1}$, we can decompose it uniquely as $\theta = \theta_{V_1} + \theta_{V_1^\perp}$, where $\theta_{V_1} = (u^\top \theta)u$ and $\theta_{V_1^\perp} \perp V_1$. For any $x \in V_1$, we have $x = (x^\top u)u$, and $\theta^\top x$ can thus be decomposed as:

$$\theta^\top x = (\theta_{V_1} + \theta_{V_1^\perp})^\top x = \theta_{V_1}^\top x = (u^\top \theta)^\top (u^\top x). \tag{11}$$

This implies that for any slice $\theta$, the projected distributions $\theta_\# \mu^d$ and $\theta_\# \nu^d$ are equivalent (up to scaling) to the distributions obtained by projecting $\mu^d$ and $\nu^d$ onto $u$. Specifically:

$$W_p^p(\theta_\# \mu^d, \theta_\# \nu^d) = |u^\top \theta|^p W_p^p(u_\# \mu^d, u_\# \nu^d). \tag{12}$$

**Generalizing to higher-dimensional effective subspaces.** We extend the idea from one dimension to a $k$-dimensional subspace $V_k$ and investigate how the reweighting function $\rho_\phi(\phi_U(\theta)) = \|U^\top \theta\|^{-p}$ adjusts the contributions of slices in higher dimensions.

**Proposition 4.7.** *Under Assumption 4.1, let $\mu^k = U_\# \mu^d$ and $\nu^k = U_\# \nu^d$ be the pushforward measures in $\mathbb{R}^k$. Then, for any $\theta^d \in \mathbb{S}^{d-1}$, we have that:*

$$W_p^p(\theta_\#^d \mu^d, \theta_\#^d \nu^d) = W_p^p((U^\top \theta^d)_\# \mu^k, (U^\top \theta^d)_\# \nu^k) = \|U^\top \theta^d\|^p W_p^p(\theta_\#^k \mu^k, \theta_\#^k \nu^k), \tag{13}$$

*where $\theta^k = \frac{U^\top \theta^d}{\|U^\top \theta^d\|}$ with convention $\theta^k = 0_k$ if $\|U^\top \theta^d\| = 0$.*

*Furthermore, we have that:*

$$SW_p^p\left(\mu^k, \nu^k; \frac{1}{L}\sum_{l=1}^L \delta_{\theta_l^k}\right) = \widetilde{SW}_p^p\left(\mu^d, \nu^d; \frac{1}{L}\sum_{l=1}^L \delta_{\theta_l^d}, \rho\right) \tag{14}$$

$$SW_p^p\left(\mu^k, \nu^k\right) = \widetilde{SW}_p^p\left(\mu^d, \nu^d; \mathcal{U}(\mathbb{S}^{d-1}), \rho\right) \tag{15}$$

*Here, we adopt the convention $\frac{1}{0} \cdot 0 = 0$ in (14) if $\|U^\top \theta_l^d\| = 0$.*

The proof is in the Appendix A.4.

**Remark 4.8** (Implicit Downweighting). *Under the conditions of Proposition 4.7, each slice contribution is implicitly downweighted by $\|U^T \theta^d\|^p$. That is, for any $\theta^d \in \mathbb{S}^{d-1}$, we have that $W_p^p(\theta_\#^d \mu^d, \theta_\#^d \nu^d) \leq W_p^p(\mu^k, \nu^k)$. Moreover, the downweighting is maximal if $\theta^d \perp \text{span}(U)$ and vanishing if $\theta^d \in \text{span}(U) \cap \mathbb{S}^{d-1}$.*

**Rescaling to equalize informativeness.** Assumption 4.1 gives rise to the fact that each one-dimensional Wasserstein distance $W_p^p(\theta_\#^d \mu^d, \theta_\#^d \nu^d)$ is implicitly downweighed by $|U^\top \theta^d|^p$. This observation naturally fits into the proposed $\phi$-weighting formulation, as there is an implicit scaling factor associated with each slice. To counteract it and make all slices equally (ES-aligned) informative, we use the reciprocal weighting function (10) to compensate for the implicit down-weighting of misaligned slices. Then, we have that

$$\rho_\phi(\phi_U(\theta^d))W_p^p(\theta_\#^d \mu^d, \theta_\#^d \nu^d) = \begin{cases} W_p^p(\theta_\#^k \mu^k, \theta_\#^k \nu^k), & \text{if } \phi_U(\theta^d) > 0, \\ 0, & \text{if } \phi_U(\theta^d) = 0, \end{cases} \tag{16}$$

where $\theta^k = \frac{U^\top \theta^d}{\|U^\top \theta^d\|}$.

## 4.3 SUBSPACE SLICED-WASSERSTEIN IS RESCALED SLICED-WASSERSTEIN

In this section, we will show that the generalized notion of informative slices (as defined in 4.3) becomes particularly advantageous for equalizing slice informativeness.

Starting from (13), we integrate both sides over $\theta^d \in \mathbb{S}^{d-1}$ wrt the uniform measure $\sigma(\theta^d)$ and obtain

$$SW_p^p(\mu^d, \nu^d) = \int_{\mathbb{S}^{d-1}} W_p^p(\theta_\sharp^d \mu^d, \theta_\sharp^d \nu^d) d\sigma(\theta^d) = \int_{\mathbb{S}^{d-1}} \|U^\top \theta^d\|^p W_p^p\left(\theta_\sharp^k \mu^k, \theta_\sharp^k \nu^k\right) d\sigma(\theta^d). \tag{17}$$

Note that $\theta^k$ depends on $\theta^d$, and the distribution of $\theta^k$ induced by $\theta^d \sim \sigma$ is uniform over $\mathbb{S}^{k-1}$. We introduce the change of variables from $\theta^d$ to $\theta^k$ and express the integral in terms of $\theta^k$:

$$SW_p^p(\mu^d, \nu^d) = \int_{\mathbb{S}^{k-1}} W_p^p\left(\theta_\#^k \mu^k, \theta_\#^k \nu^k\right) \left(\int_{\theta^d: \frac{U^\top \theta^d}{\|U^\top \theta^d\|} = \theta^k} \|U^\top \theta^d\|^p \, d\sigma(\theta^d | \theta^k)\right) dT_\# \sigma(\theta^k),$$
(18)

where $\sigma(\cdot | \theta^k)$ is the conditional distribution of $\theta^d$, and $T : x \mapsto \frac{U^\top x}{\|U^\top x\|}$ is the mapping from $\theta^d$ to $\theta^k$.

The inner integral over $\theta^d$ can be evaluated as a scaling factor $C_{d,k}$ dependent on $\sigma$, $\theta^k$, $U$. When $\sigma = \mathcal{U}(\mathbb{S}^{d-1})$, $C_{d,k}$ is invariant for all $\theta^k$.

Substituting back into (18), and let $\sigma_k = T_\# \sigma = \mathcal{U}(\mathbb{S}^{k-1})$ denote the distribution of $\theta^k$, we obtain

$$SW_p^p(\mu^d, \nu^d) = C_{d,k} \int_{\mathbb{S}^{k-1}} W_p^p\left(\theta_\#^k \mu^k, \theta_\#^k \nu^k\right) \, d\sigma_k(\theta^k).$$
(19)

Since $\sigma_k(\theta^k)$ integrates to 1 over $\mathbb{S}^{k-1}$, and $W_p^p\left(\theta_\#^k \mu^k, \theta_\#^k \nu^k\right)$ is integrated over all $\theta^k$, we can express the right-hand side as $C_{d,k} \cdot SW_p^p(\mu^k, \nu^k; \sigma_k)$. Intuitively speaking, this means the *loss of information* is due to an implicit constant factor on $SW_p^p(\mu^d, \nu^d)$, which we denote as the **Effective Subspace Scaling Factor** (ESSF). Thus, rescaling the one-dimensional Wasserstein for all slices via Equation (16) becomes multiplying the SWD by the reciprocal of the ESSF. We proceed further to make this connection explicit by the following theorem.

**Theorem 4.9** (Effective Subspace Scaling Factor). *Let $\mu^d, \nu^d \in \mathcal{P}(\mathbb{R}^d)$ satisfy Assumption 4.1, and define $\mu^k = U_\# \mu^d$ and $\nu^k = U_\# \nu^d$. Then we have that*

$$SW_p^p(\mu^d, \nu^d) = \frac{C_k}{C_d} \cdot SW_p^p(\mu^k, \nu^k),$$
(20)

*where $C_d = 2^{p/2} \frac{\Gamma\left(\frac{d}{2} + \frac{p}{2}\right)}{\Gamma\left(\frac{d}{2}\right)}$ and $C_k$ is defined analogously, with $\Gamma$ denoting the Gamma function.*

When $k < d$, assuming $\|U^\top \theta_l^d\| \neq 0$ is reasonable since $\mathcal{U}(\mathbb{S}^d)(\{\theta^d \in \mathbb{S}^{d-1} : U^\top \theta = 0\}) = 0$.

The proof is in the Appendix A.4.

**Proposition 4.10.** *Let $\mu^d, \nu^d \in \mathcal{P}(\mathbb{R}^d)$ satisfy Assumption 4.1. Consider the empirical estimator $\widehat{ESSF}(L)$ defined as:*

$$\widehat{ESSF}(L) = \frac{1}{L} \sum_{l=1}^{L} \|U^\top \theta_l^d\|^p,$$
(21)

*where $\{\theta_l^d\}_{l=1}^L \overset{i.i.d.}{\sim} \mathcal{U}(\mathbb{S}^{d-1})$. We have that*

1. *$\mathbb{E}[\widehat{ESSF}(L)] = \frac{C_k}{C_d}$ and $Var(\widehat{ESSF}(L)) = \mathcal{O}(\frac{1}{L})$.*

2. *Let $\epsilon_L = \left| SW_p^p\left(\mu^d, \nu^d; \frac{1}{L} \sum_{l=1}^L \delta_{\theta_l^d}\right) - \widehat{ESSF}(L) \cdot SW_p^p\left(\mu^k, \nu^k; \frac{1}{L} \sum_{l=1}^L \delta_{\theta_l^k}\right) \right|$. Then $\epsilon_L \overset{a.s.}{\longrightarrow} 0$ as $L \to \infty$*

3. *There exists a constant $K > 0$ depending only on $\mu^d$ and $\nu^d$ such that for any $\delta > 0$, we have $\mathbb{P}(\epsilon_L < \delta) \geq 1 - e^{-\delta^2 L / K^2}$.*

The proof of this proposition is in the Appendix A.6.

In Section 5.1 we provide empirical results showing how the variance of $\widehat{ESSF}(L)$ changes wrt $L$.

### 4.4 IMPLICATIONS FOR LEARNING ALGORITHMS: IS SWD ALL YOU NEED?

Assumption 4.1 naturally holds in common machine learning settings. In gradient-based learning, data is typically processed in minibatches, leading to an effective bound on $k$ related to batch size $B \ll d$. Additionally, real datasets often have feature (linear) correlations, potentially reducing $k$ further. Lastly, the $\widehat{ESSF}(L)$ factor, despite its variance, can be absorbed into the learning rate during optimization. This reduces the problem to a single hyperparameter search for the optimal learning rate—a standard practice in machine learning workflows.

**Remark 4.11.** *Let $\{x_i\}_{i=1}^{2B} \subset \mathbb{R}^d$ be a minibatch of $2B$ samples ($B$ from source, $B$ from target). Let $X = [x_1, \ldots, x_{2B}] \in \mathbb{R}^{d \times 2B}$ be the corresponding data matrix. Then the support of the empirical distributions lies in a subspace of dimension $k \leq \min\{2B, d\}$.*

**Proposition 4.12.** *For discrete distributions $\hat{\mu}_d = \sum_{i=1}^{n} q_i^1 \delta_{x_i}$ and $\hat{\nu}_d = \sum_{j=1}^{m} q_j^2 \delta_{y_j}$, we have:*

$$\nabla_x W_p^p(\theta_{\#}\hat{\mu}_d, \theta_{\#}\hat{\nu}_d) = \|U^\top \theta\|^p \nabla_x W_p^p(\theta_{\#}^k \hat{\mu}_k, \theta_{\#}^k \hat{\nu}_k) \tag{22}$$

*where $\theta^k = U^\top \theta / \|U^\top \theta\|$. Define the empirical gradient error for each $x_i$ as $\epsilon_L(x_i) := \|\nabla_{x_i} SW_p^p(\hat{\mu}_d, \hat{\nu}_d; \sum_{l=1}^{L} \delta_{\theta_l^d}) - \widehat{ESSF}(L) \cdot \nabla_{x_i} SW_p^p(\hat{\mu}_k, \hat{\nu}_k; \sum_{l=1}^{L} \delta_{\theta_l^k})\|$, Then the following holds*

1. *$\epsilon_L(x_i) \xrightarrow{\mathbb{P}} 0$ as $L \to \infty$*

2. *$\mathbb{P}(\|\epsilon_L(x_i)\| \leq \epsilon) \geq 1 - 2e^{-\epsilon^2 L/(pq_i^1 K)^2}$, where $K = \max_{x_i, y_j} \|x_i - y_j\|^{p-1} < \infty$.*

We refer readers to the Appendix A.7 for the detailed discussion and proofs.

# 5 EXPERIMENTS

We use 50 random projections for the SWD. For other variants, we use the default hyperparameters provided by the official implementations. More details (results, visualizations) are in the Appendix.

## 5.1 NUMERICAL VALIDATION OF MAIN RESULTS

**Verifying Theorem 4.9 for $p = 1, 2$.** Our setup involves two $k$-dimensional isotropic Gaussians embedded in $\mathbb{R}^d$ ($d \geq k$). We generated 500 samples from each and varied both $d$ and $k$ to observe how the empirical ratio $\widehat{C} = \frac{\widehat{SW}_p^p(\mu^d, \nu^d)}{\widehat{SW}_p^p(\mu^k, \nu^k)}$ behaves under different dimensionality settings for a fixed number of slices ($L = 1000$). **a)** Fixing $k = 2$, varying $p$ across $\{1, 2\}$, and varying $d$ across $\{10, 30, 50, 80, 100, 300, 500, 800, 1000\}$. **b)** Fixing $d = 1000$, varying $p$ across $\{1, 2\}$, and varying $k$ across $\{10, 30, 50, 80, 100, 300, 500, 800, 1000\}$. The results are averaged over 10 runs

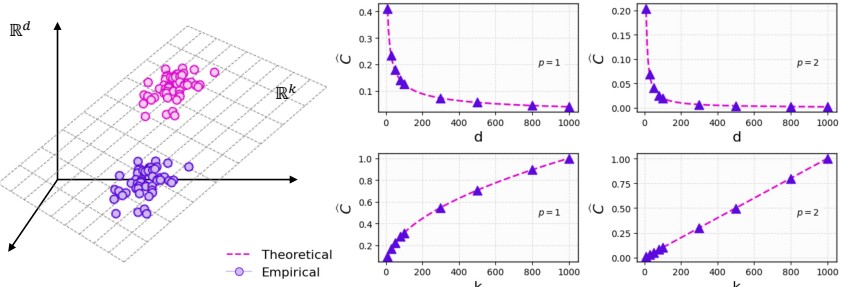

Figure 2: **Left**: Illustration of two embedded Gaussians. **Top row**: Empirical $\widehat{C}$ with varying $d$ for $k = 2$ and $p = 1, 2$. **Bottom row**: Empirical $\widehat{C}$ with varying $k$ for in $d = 1000$ and $p = 1, 2$.

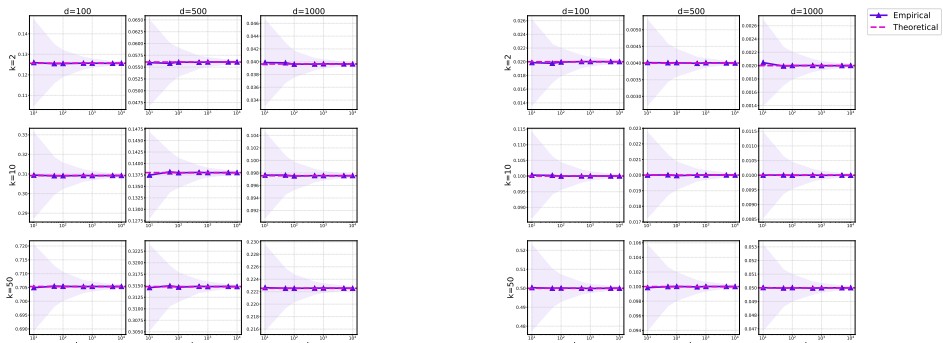

Figure 3: The $\widehat{ESSF}(L)$ for varying $d, k$ over 1000 runs for $p = 1$ (left) and $p = 2$ (right).

**Verifying Proposition 4.10** We proceed further to observe the empirical estimate $\widehat{ESSF}(L)$ and its variance for different values of $L = \{10, 50, 100, 500, 1000, 5000, 10000\}$. Here, we use $d = \{100, 500, 1000\}$ and $k = \{2, 10, 50\}$. The results are across 1000 runs.

## 5.2 GRADIENT FLOW

**On classic synthetic datasets.** We generate 300 particles as target from three classic 2D datasets: Swiss role, 8 Gaussians, and Knot. The source is realized from a 2D isotropic Gaussian. We embed these data into the space with target dimensions of $d = \{2, 50, 100\}$ by padding with 0's and applying a random $d$-dimensional rotation on the $2D$ data plane. We use 10000 iterations with vanilla GD and results are over 3 runs. **Learning rates:** $\{1, 3, 5, 8\} \times 10^{\{-6, -5, -4, -3, -2, -1, 0, 1, 2\}}$.

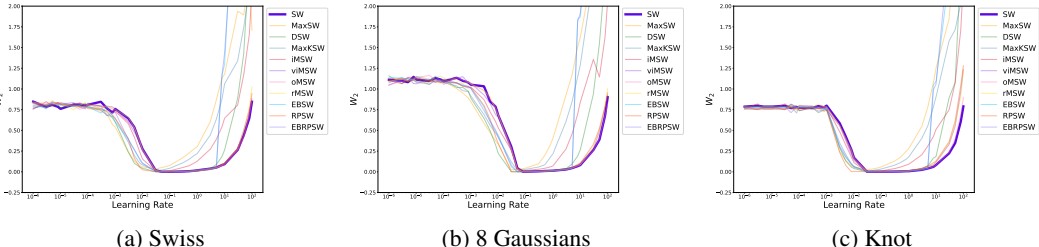

| (a) Swiss | (b) 8 Gaussians | (c) Knot |

Figure 4: Optimal basin plots for Gradient Flow with embedded synthetic datasets.

**On MNIST/CelebA images.** For MNIST, we randomly select a set of 50 samples from digits 0 (as source) and 1 (as target) to perform gradient flow with 200000 iterations. **Learning rates:** $\{1, 5\} \times 10^{\{-3, -2, -1, 0, 1, 2, 3\}}$. For CelebA, we randomly select a set of 50 samples to perform gradient flow from the Gaussian source noises with 200000 iterations. **Learning rates:** $\{1, 4, 8, 16, 64, 128, 256, 512, 1024, 3200\}$.

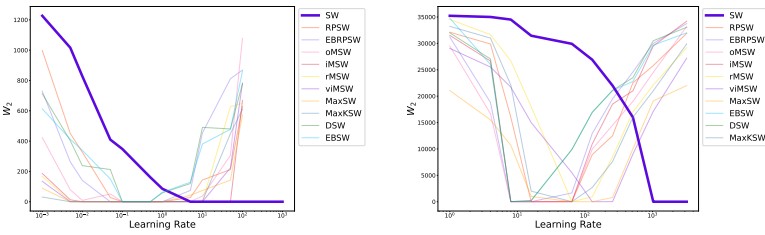

Figure 5: Optimal basin plots for MNIST (left) and CelebA (right).

## 5.3 COLOR TRANSFER

We follow a similar setup as in (Nguyen et al., 2024a; Nguyen & Ho, 2024; Nguyen et al., 2024b) with different hyperparameters. Our experiments are performed over 3 image sets (See Figure 11). The optimization uses $50,000$ iterations. To reduce computational complexity, we optionally apply K-means clustering with $3,000$ clusters, to reduce the colorspace into an empirical measure with $N = 3,000$ particles. **Learning rates:** $\{1, 3, 5, 8\} \times 10^{\{-4, -3, -2, -1, 0, 1\}}, 100$.

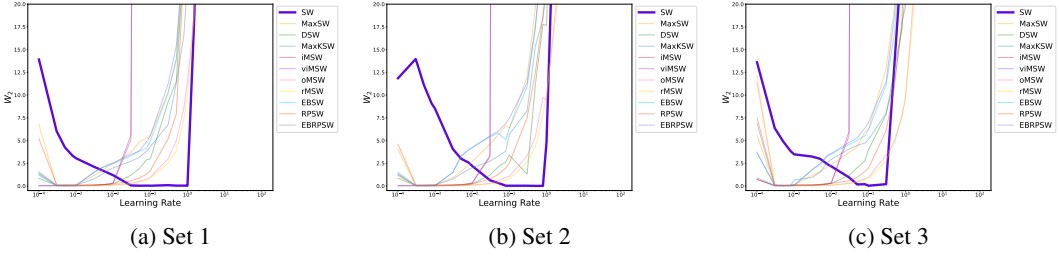

| (a) Set 1 | (b) Set 2 | (c) Set 3 |

Figure 6: Optimal basin plots for Gradient Flow with embedded synthetic datasets.

## 5.4 DEEP GENERATIVE MODELING

There exist various generative modeling setups with Sliced-Wasserstein (Kolouri et al., 2018; Deshpande et al., 2018; Wu et al., 2019; Liutkus et al., 2019; Nguyen et al., 2024b). We restrict our setup to

the latent space ($d = 512$) of an autoencoder (Pidhorskyi et al., 2020) pretrained on the $1024 \times 1024$ FFHQ dataset (Karras et al., 2019). **Learning rates:** $\{1, 3, 5, 8\} \times 10^{\{-6, -5, -4, -3, -2, -1\}}, 1$.

We evaluate SW variants on both unconditional generation and unpaired image-to-image translation tasks. For generation, we follow Deshpande et al. (2018)'s SWG setup using a generator $G_\phi(\cdot)$ to transform $z \in \mathbb{R}^8$ to latents $X \in \mathbb{R}^{512}$. For translation, we modify this to use a residual generator transforming source domain $X$ to target domain $Y$ latents. Following Rombach et al. (2022), Korotin et al. (2023), we operate in an autoencoder's latent space to sidestep the known dimensionality challenges of SWG (Deshpande et al. (2018), Nadjahi et al. (2021)). We train for 10000 iterations using vanilla gradient descent with batch size 2048. For translation, we evaluate on two FFHQ subtasks: Male to Female (M2F) and Adult to Children (A2C) using split training/test sets.

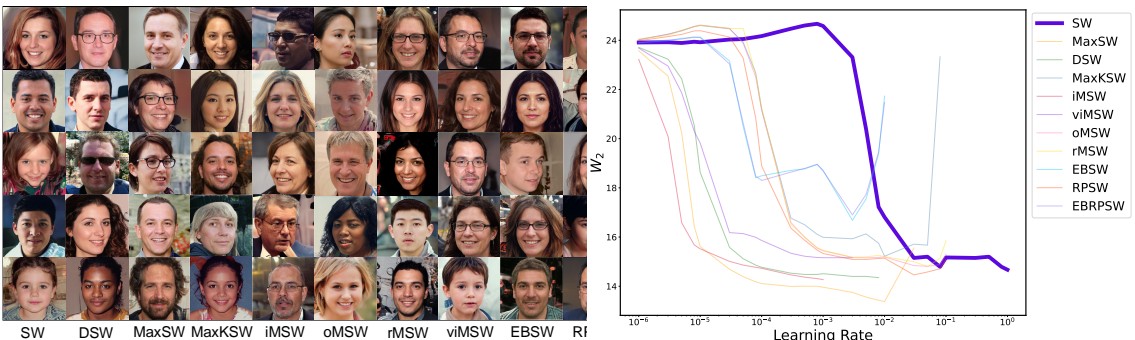

Figure 7: **Left:** Samples generated using different SW variants. **Right:** Optimal basin plot.

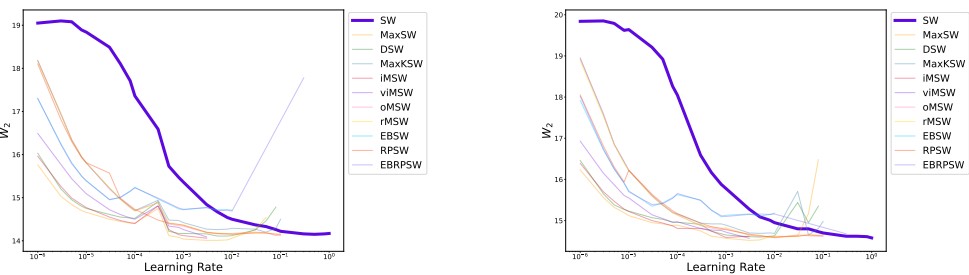

Figure 8: Optimal basin plots for M2F(left) and A2C(right).

## 6 CONCLUSION

In this paper, we revisit the classic Sliced-Wasserstein and rethink the current approaches that modify the slicing distribution to focus on informative slices. We introduce another perspective of rescaling the 1D Wasserstein distances based on slice 'informativeness.' By defining the notion of informativeness in terms of alignment with the data effective subspace, we show this rescaling simplifies to a global scaling factor on the SW. This directly translates to the standard learning rate search in gradient-based optimization (even with a finite number of slices). We then empirically show that, in a wide variety of learning settings, a properly configured SW performs competitively with other complex variants without the additional computation/memory overheads. This challenges the notion that increasingly advanced methods are always necessary for improved performance. We show that while standard SW may not be the universal solution for all scenarios, it remains to be a reliable and efficient solution for common learning tasks.

**Future research:** Our work does not preclude further research to improve the Sliced-Wasserstein using the current approaches, which have their own merits. In fact, the rescaling approach has deep connections to modifying the slicing distributions. Nonetheless, our work provides a novel and generalized perspective to interpret and address a major limitation of the classic SW. From this angle, future research could investigate different choices for the rescaling function $\rho$ under various assumptions about the data, as well as explore alternative notions of *slice informativeness*.

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

## A  PROOFS AND ADDITIONAL THEORETICAL RESULTS

### A.1  NOTATION

- $\mathbb{R}^d$: d-dimensional Euclidean space, where $d$ is a positive integer.
- $\mathbb{S}^{d-1} := \{x \in \mathbb{R}^d : \|x\| = 1\}$: unit sphere defined in $\mathbb{R}^d$.
- $\mathcal{P}(\mathbb{R}^d)$: set of all probability measures defined on $\mathbb{R}^d$.
- $\mathcal{P}_p(\mathbb{R}^d)$: set of probability measures whose $p$-th moment is finite, where $p \geq 1$.
- $\mathbb{V}_{k,d}$: set of all $d \times k$ orthogonal matrices, i.e.

$$\mathbb{V}_{k,d} := \{U \in \mathbb{R}^{d \times k} : U^\top U = I_k\}.$$

  Note, $\mathbb{S}^{d-1} = \mathbb{V}_{1,d}$.
- $U = [U[:,1], U[:,2], \ldots U[:,k]] \in \mathbb{V}_{k,d}$: an orthogonal matrix. For each $i \in [1 : k]$, $U[:,i] \in \mathbb{R}^d$ is the $i$-th column of $U$.
  Note that $U$ induces a linear function from $\mathbb{R}^d$ to $\mathbb{R}^k$, i.e. $x \mapsto U^\top x$. With abuse of notation, we do not distinguish the matrix $U$ and the corresponding linear mapping.
- $\mathrm{Span}(U)$: The linear subspace spanned by $U$, i.e.

$$\mathrm{Span}(U) := \mathrm{Span}(\{U[:,1], U[:,2], \ldots U[:,k]\}) = \left\{\sum_{i=1}^{k} \alpha_i U[:,i] : \alpha_i \in \mathbb{R}\right\}.$$

- $V_k \subset \mathbb{R}^d$: a $k$-dimensional subspace, where $k$ is a positive integer with $k \leq d$. Note, by classical linear algebra theory, we have

$$V_k = \mathrm{Span}(U)$$

  for some $U \in \mathbb{V}_{d,k}$. **Note, given $V_k$, $U$ is not uniquely determined.**
- $V_k^\perp$: perpendicular complement of $V_k$, which is a subspace of dimension $d - k$.
- $\mu^d, \mu, \nu^d, \nu \in \mathcal{P}(\mathbb{R}^d)$: probability measures in $d$-dimensional space.
- $\mathcal{L}^d$: Lebesgue measure in $\mathbb{R}^d$.
- $C_0(\mathbb{R}^d)$: set of all continuous functions defined on $\mathbb{R}^d$ which vanish at infinity.
- $f_\mu = \frac{d\mu^d}{d\mathcal{L}^d}$: density of $\mu$, that is, for all test functions $\phi \in C_0(\mathbb{R}^d)$:

$$\int_{\mathbb{R}^d} \phi(x) d\mu^d(x) = \int_{\mathbb{R}^d} f_\mu(x)\phi(x)dx.$$

- $X \sim \mu$: A random variable/vector $X$ following distribution $\mu$. We say $X$ is a **realization** of $\mu$.
- $\mathbb{E}[X] := \mathbb{E}[\mu]$, where $X \sim \mu$: expected value of $X$, i.e.

$$\mathbb{E}_\mu[X] = \int_{\mathbb{R}^d} x d\mu(x).$$

- $m_k(\mu)$: $k$-th moment of measure $\mu$. That is, given realization $X \sim \mu$, $m_k(\mu)$ is defined by

$$m_k(\mu) := \mathbb{E}[X^k]$$

- $Var(X) := \mathbb{E}[(X - \mathbb{E}(X))^\top (X - \mathbb{E}(X))]$: the covariance matrix of $X$ (or the measure $\mu$).
- $T_\#\mu$, where $T : \mathbb{R}^d \to \mathbb{R}^d$ is a function: push-forward measure $\mu$ under mapping $T$. That is, for all Borel sets $A \subset \mathbb{R}^d$, we have

$$T_\#\mu(A) = \mu(T^{-1}(A)).$$

  Equivalently speaking, suppose $X \sim \mu$ is a realization of $\mu$, then $T(X) \sim T_\#\mu$.
- $\mathcal{N}(e, \Sigma)$: Gaussian distribution, where $e \in \mathbb{R}^d$ is the expected value, $\Sigma \in \mathbb{R}^{d \times d}$ is the covariance matrix.

- $0_d$: $d \times 1$ vector where each entry is $0$. Similarly, we define $1_d$.

- $I_d$: $d \times d$ identity matrix.

- $\mathcal{U}(\mathbb{S}^{d-1})$: Uniform distribution defined on $\mathbb{S}^{d-1}$.

- $\theta^d \sim \mathcal{U}(\mathbb{S}^d)$: a $d$-dimensional random vector. We say $\theta^d$ is a **realization** of $\mathcal{U}(\mathbb{S}^d)$.

- $\theta, \theta^d, \theta^g$: a $d$-dimensional vector.

- $\theta^k$: a $k$-dimensional vector.

- $P_{V_k} := P_U$, where $V_k = \mathrm{Span}(U)$: the projection mapping from $\mathbb{R}^d$ into subspace $V_k$, i.e.

$$P_{V_k}(x) := P_U(x) = UU^\top x, \forall x \in \mathbb{R}^d.$$

  Note, in this case: the mapping $U : \mathbb{R}^d \to \mathbb{R}^k$ with $x \mapsto U^\top x$ is the corresponding parameterization function of projection $P_U$.

- $\Gamma(\mu, \nu)$: set of joint measures whose marginals are $\mu, \nu$ respectively:

$$\Gamma(\mu, \nu) := \{\gamma \in \mathcal{P}((\mathbb{R}^d)^2) : (\pi_1)_\# \gamma = \mu, (\pi_2)_\# \gamma = \nu\},$$

  where $\pi_1 : (x, y) \mapsto x, \pi_2 : (x, y) \mapsto y$ are canonical projection mappings.

- $W_p^p(\mu, \nu)$: Wasserstein problem between $\mu$ and $\nu$:

$$W_p^p(\mu, \nu) := \inf_{\gamma \in \Gamma(\mu, \nu)} \int_{(\mathbb{R}^d)^2} \|x - y\|^p d\gamma(x, y)$$

- $SW(\mu, \nu; \sigma)$, where $\sigma \in \mathcal{P}(\mathbb{S}^{d-1})$: Sliced Wasserstein problem between $\mu$ and $\nu$ with respect to reference measure $\sigma$:

$$SW_p^p(\mu, \nu; \sigma) := \int_{\mathbb{S}^{d-1}} W_p^p(\theta_\# \mu, \theta_\# \nu) d\sigma(\theta)$$

- $\phi_U : \mathbb{S}^{d-1} \to \mathbb{R}_+$: ES-informative aligned mapping. A measurable mapping which describes the information of the projected $\theta$ on the space spanned by $U$.

- $\widetilde{SW}(\mu, \nu; \sigma, \rho)$: rescaled sliced Wasserstein distance:

$$\widetilde{SW}(\mu, \nu; \sigma, \rho) := \int_{\mathbb{S}^{d-1}} r(\phi_U(\theta)) W_p^p(\theta_\# \mu, \theta_\# \nu) d\sigma(\theta)$$

  where $\rho : \mathbb{R}_+ \to \mathbb{R}_+$ is a recalling function. In this paper, we set $\rho$ as the following decreasing function:

$$\rho(x) = \frac{1}{x^p}$$

  When $x = 0$, we adopt the convention $\rho(x) = 0$.

**Remark A.1.** *In this paper, we adopt the following convention.*

*We do not distinguish the scalar/vector/matrix and the corresponding induced linear mapping. For example, $\theta \in \mathbb{R}^d$, induces the mapping*

$$\mathbb{R}^d \ni x \mapsto \theta^\top x \in \mathbb{R}.$$

- *When $\theta$ is a random vector, we refer to it as a "random projection mapping" in both the main text and the appendix. We adopt the same convention for the scalar notation $\alpha$ and the matrix notation $U$.*

- *We use $\theta_\# \mu$ to denote the push-forward measure induced by mapping $x \mapsto \theta^\top x$. Similarly, $(\theta \times \theta)_\# \gamma$ denotes the push-forward measure of joint measure $\gamma \in \mathcal{P}((\mathbb{R}^d)^2)$ induced by mapping $(x, x') \mapsto (\theta^\top x, \theta^\top x')$. The same convention is adopted for $\alpha, U$.*

**Remark A.2.** *For simplicity, in notation $SW(\mu, \nu; \sigma)$, we may relax the restriction that $\sigma$ is a probability measure. We allow $\sigma$ to be a finite positive measure in the main text and appendix.*

## A.2    Wasserstein distances in $\mathbb{R}^d$ and $\mathbb{R}^k$

In this article, we assume the probability measures $\mu^d, \nu^d \in \mathcal{P}_p(\mathbb{R}^d)$ are supported in a lower dimensional subspace. We refer to Assumption 4.1 for details.

Let $P_U$ denote the projection mapping from $\mathbb{R}^d$ to $V_k$:

$$P_U(x) = UU^\top x, \forall x \in \mathbb{R}^d, \tag{23}$$

Then, the corresponding lower-dimensional parameterization mapping is defined as:

$$x \mapsto U^\top x, \forall x \in \mathbb{R}^d. \tag{24}$$

By classical linear algebra theory, it is straightforward to verify the following:

**Proposition A.3.** *[Basic properties of linear projection] Let $P_U, U$ be defined above, then we have:*

*(1) For each $\theta \in \mathbb{R}^d$, $\theta$ can be uniquely decomposed into $V_k, V_k^\perp$, i.e. $\theta = \theta_{V_k} + \theta_{V_k^\perp}$, where $\theta_{V_k} = P_U(\theta) \in V_k, \theta_{V_k^\perp} \in V_k^\perp$.*

*(2) For all $x \in V_k$, $P_U(x) = x$.*

*(3) If we restrict $U$ to space $V_k$, denoted as $U\mid_{V_k}$, then $U\mid_{V_k}: V^k \to \mathbb{R}^k$ is a bijection. The inverse is given by*

$$(U)^{-1}(y) = Uy, \forall y \in \mathbb{R}^k.$$

*In addition, $\|U(x)\| = \|x\|, \forall x \in V_k$.*

*Proof.* It follows directly from the definitions of $P_U, U$. $\qquad\qquad\qquad\qquad\square$

Let $\mu^k = (U)_\# \mu^d, \nu^k = (U)_\# \nu^d$, the above proposition directly induces the following relation between the Wasserstein distance between $\mu^d, \nu^d$ and the Wasserstein distance between $\mu^k, \nu^k$.

**Proposition A.4.** *Under assumption 4.1, we have the following:*

*(1) $\mu^d$ can be recovered by the inverse of $U\mid_{V_k}$, i.e.*

$$\mu^d = U^\top_\# \mu^k.$$

*(2) The mapping*

$$\Gamma(\mu^d, \nu^d) \ni \gamma^d \mapsto \gamma^k := (U \times U)_\# \gamma^d \in \Gamma(\mu^k, \nu^k), \tag{25}$$

*is a well-defined bijection, where $U \times U$ is defined as*

$$\mathbb{R}^d \times \mathbb{R}^d \ni (x, x') \mapsto U \times U((x, x')) = (U(x), U(x')) \in \mathbb{R}^k \times \mathbb{R}^k. \tag{26}$$

*(3) The Wasserstein distance is preserved via the lower-dimensional parameterization:*

$$W_p^p(\mu^d, \nu^d) = W_p^p((P_U)_\# \mu^d, (P_U)_\# \nu^d) = W_p^p(\mu^k, \nu^k) \tag{27}$$

*Proof.* Let $X \sim \mu^d$ be a realization.

(1) We have $U^\top X \sim \mu^k$ since $\mu^k = (U)_\# \mu^d$. In addition, by assumption (4.1), we have $X = UU^\top X$, thus $UU^\top X \sim \mu^d$. That is $U^\top_\# \mu^k = \mu^d$.

(2) Pick $\gamma^d \in \Gamma(\mu^d, \nu^d)$, we have

$$(\pi_1)_\#(U \times U)_\# \gamma^d = (U)_\#((\pi_1)_\# \gamma^d) = (U)_\# \mu^d = \mu^k$$

Similarly, $(\pi_2)_\#(U \times U)_\# \gamma^d = \nu^k$. Thus the mapping defined in (25) is well-defined. Moreover, from statement (1), we have

$$\Gamma(\mu^k, \nu^k) \ni \gamma^k \mapsto (U^\top \times U^\top)_\# \gamma^k \in \Gamma(\mu^d, \nu^d) \tag{28}$$

is well-defined.

Next, we will show the above mapping is the inverse of (25).

Let $(X, Y) \sim \gamma^d$ be a realization. Then

$$(X, Y) = (UU^\top X, UU^\top Y) \sim (U^\top \times U^\top)_{\#}(U \times U)_{\#}\gamma.$$

Thus $(U^\top \times U^\top)_{\#}(U \times U)_{\#}\gamma = \gamma$. Thus, the mapping (28) is inverse of the mapping (25). Thus, (25) is invertible/bijection.

(3) By Proposition A.3 (2), for each $x \in \text{supp}(\mu) \subset V_k$, we have $P_U(x) = x$, thus $(P_U)_{\#}\mu^d = \mu^d$. Similarly, $(P_U)_{\#}\nu^d = \nu^d$. Thus we obtain the first equality:

$$W_p^p(\mu^d, \nu^d) = W_p^p((P_U)_{\#}\mu^d, (P_U)_{\#}\nu^d).$$

For the second equality, we first pick $\gamma^d \in \Gamma(\mu^d, \nu^d)$ and let $\gamma^k = (U \times U)_{\#}\gamma^d$. By statement (2), we have $\gamma^k \in \Gamma(\mu^k, \nu^k)$.

$$\int_{(\mathbb{R}^d)^2} \|x - y\|^p d\gamma^d(x, y)$$

$$= \int_{(\mathbb{R}^d)^2} \|U(x) - U(y)\|^p d\gamma^d(x, y)$$

$$= \int_{(\mathbb{R}^k)^2} \|x' - y'\|^p d(U^\top \times U^\top)_{\#}\gamma^d(x', y')$$

$$= \int_{(\mathbb{R}^k)^2} \|x' - y'\|^p d\gamma^k(x', y')$$

where the first equality follows from Proposition A.3 (3), the second equality follows from the definition of push-forward measure, the third equality holds from statement (2).

Combining the above equality with statement (2), we obtain

$$W_p^p(\mu^d, \nu^d) = \inf_{\gamma^d \in \Gamma(\mu^d, \nu^d)} \int_{\mathbb{R}^d} \|x - y\|^p d\gamma^d(x, y)$$

$$= \inf_{\gamma^k \in \Gamma(\mu^k, \nu^k)} \int_{\mathbb{R}^k} \|x' - y'\|^p d\gamma^k(x', y')$$

$$= W_p^p(\mu^k, \nu^k)$$

$\square$

### A.3 BACKGROUND: RELATIONSHIP BETWEEN THE GAUSSIAN AND SPHERICAL UNIFORM DISTRIBUTION

In this section, we introduce basic properties of multivariate Gaussian and the relation between Gaussian and spherical uniform distribution.

First we consider 1D space $\mathbb{R}$, choose $e \in \mathbb{R}$ and $\sigma > 0$, the Gaussian distribution, denoted as $\mathcal{N}(e, \sigma^2)$, is the probability measure whose density is defined by

$$f(x) := \frac{1}{\sqrt{2\pi\sigma^2}} e^{-\frac{(x-e)^2}{2\sigma^2}},$$

where $e, \sigma^2$ are the expected value and variance of $X$ respectively.

When $e = 0, \sigma^2 = 1$, the induced measure is called standard (1D) Gaussian distribution, whose density is given by

$$f(x) := \frac{1}{\sqrt{2\pi}} e^{-\frac{x^2}{2}} \tag{29}$$

In space $\mathbb{R}^d$, the above density function can be generalized as:

$$f(x) := \frac{1}{(2\pi)^{d/2}} e^{-\frac{\|x\|^2}{2}} \tag{30}$$

and the induced distribution is called $d$-**dimensional Standard Gaussian distribution**.

Given $e \in \mathbb{R}^d$ and positive definite $d \times d$ matrix, $\Sigma = AA^T$ where $A \in \mathbb{R}^{d \times k}$, the Gaussian distribution is denoted as $\mathcal{N}(e, \Sigma)$, can be defined by the following well-known proposition:

**Proposition A.5** (Definition of Gaussian distribution). *Let $X \sim \mathcal{N}(e, \Sigma)$ be a realization, then the following are equivalent:*

- $\mathcal{N}(e, \Sigma)$ *is Gaussian distribution, with expected value $e$ and covariance matrix $\Sigma$.*

- $X = AG + e$, *where $G \sim \mathcal{N}(0, I_d)$, whose density is defined by (30).*

- $\forall \theta \in \mathbb{R}^d$, $\theta^\top X$ *is a 1D Gaussian variable:*

$$\theta^\top X \sim \mathcal{N}(\theta^\top e, (\theta^\top A)^\top (\theta^\top A)).$$

From the proposition, it is straightforward to verify the following:

**Proposition A.6** (Basic properties of Gaussian distribution). *Suppose $X \sim \mathcal{N}(e, \Sigma)$, then we have:*

*(1) If rank$(\Sigma) = d$, then $\mathcal{N}(e, \Sigma)$ admits the density function:*

$$f(x) = \frac{1}{(2\pi)^{d/2} det(\Sigma)^{1/2}} e^{-\frac{(x-e)^T \Sigma^{-1} (x-e)}{2}}$$

*(2) Choose $B \in \mathbb{R}^{d \times k}, \beta \in \mathbb{R}^k$, and let $T_{B,e,\beta}(x) := B(x - e) + \beta$, then we have*

$$B(X - e) + \beta \sim (T_{B,e,\beta})_\# \mathcal{N}(e, \Sigma) = \mathcal{N}(\beta, B^\top \Sigma B).$$

*(3) Suppose $Z \sim \mathcal{N}(0, I_d)$, then the absolute $p$-th power of $Z$ is given by*

$$\mathbb{E}[\|Z\|^p] = 2^{p/2} \frac{\Gamma(\frac{p+d}{2})}{\Gamma(d/2)}.$$

*(4) Suppose $Z \sim \mathcal{N}(0, I_d)$, then $r = \|Z\|, \theta = \frac{Z}{\|Z\|}$ are independent.*

At the end of this section, we introduce the following relation between the Gaussian distribution and the spherical uniform distribution.

**Proposition A.7.** *We define the following function $f$ with*

$$\mathbb{R}^d \setminus \{0\} \ni x \mapsto f(x) = \frac{x}{\|x\|}.$$

*Suppose $\Sigma = AA^\top$ is a full rank positive-semi-definite matrix, then we have*

$$f_\# \mathcal{N}(0_d, \Sigma) = \mathcal{U}(\mathbb{S}^{d-1}).$$

*Proof.* Let $X \sim \mathcal{N}(0_d, \Sigma)$ be a realization of the $d$-dimensional Gaussian, $\Theta = f(X) = \frac{X}{\|X\|}$. Note that $\Theta$ is well defined $\mathcal{N}(0_d, \Sigma)$-a.s.

**Step 1**. Suppose $\Sigma = I_d$, it is equivalent to the following:

Suppose $X_1, \ldots X_d \overset{\text{i.i.d.}}{\sim} \mathcal{N}(0,1)$ and $\Theta = [\frac{X_1}{\sqrt{\sum_{i=1}^d X_i^2}}, \ldots, \frac{X_d}{\sqrt{\sum_{i=1}^d X_i^2}}]^T$, then $\Theta \sim \text{Unif}(\mathbb{S}^{d-1})$. It is a standard result in probability theory. In particular, choose test function $\phi \in C_0(\mathbb{S}^{d-1})$, we have:

$$
\begin{aligned}
\mathbb{E}[\phi(\Theta)] &= \int_{\mathbb{R}^d} \phi(\frac{x}{\|x\|}) f_X(x) dx \\
&= \frac{1}{(2\pi)^{d/2}} \int_{\mathbb{R}^d} \phi\left(\frac{x}{\|x\|}\right) e^{-\frac{\|x\|^2}{2}} dx \\
&= \frac{1}{(2\pi)^{d/2}} \int_{\mathbb{S}^{d-1}} \int_{\mathbb{R}_+} \phi(\theta) e^{-r^2/2} r^{d-1} d\theta dr \qquad r, \theta \text{ are spherical coordinates} \\
&= \int_{\mathbb{S}^{d-1}} \phi(\theta) d\theta \cdot \underbrace{\frac{1}{(2\pi)^{d/2}} \int_{\mathbb{R}_+} e^{-r^2/2} r^{d-1} dr}_{1/\|\mathbb{S}^{d-1}\|}
\end{aligned}
$$

Thus, $\Theta \sim \text{Unif}(\mathbb{S}^{d-1})$.

**Step 2**. Suppose $\Sigma = \text{diag}(\sigma_1, \ldots \sigma_d)$ where $\sigma_1, \ldots \sigma_d > 0$, we have

$$
\Theta = \frac{X}{\|X\|} = \frac{\Sigma^{-1/2} X}{\|\Sigma^{-1/2} X\|},
$$

where $\Sigma^{-1/2} X \sim \mathcal{N}(0, I_d)$. Thus, by step 1, we have $\Theta \sim \mathcal{U}(\mathbb{S}^{d-1})$.

**Step 3**. We consider the general positive definite $\Sigma$. We have $\Sigma = U\Lambda U^\top$ where $U \in \mathbb{V}_{d,d}$ is orthonormal matrix.

We have

$$
U^\top \Theta = \frac{U^\top X}{\|X\|} = \frac{U^\top X}{\|U^\top X\|}
$$

Since $U^\top X \sim \mathcal{N}(0, \Lambda)$ and $\Lambda$ is a positive diagonal matrix, then from step 2, we have $U^\top \Theta \sim \mathcal{U}(\mathbb{S}^{d-1})$. Thus, $\Theta = U(U^\top \Theta) \sim \mathcal{U}(\mathbb{S}^{d-1})$.

$\square$

**Remark A.8.** *Note that the above statement (especially the statement in Step 1) is a well-known result, and that is why isotropic Gaussian distribution is called a "rotationally invariant distritbution." We do not claim this proposition or its proof as contributions of this article; we present the proof merely for completeness.*

### A.4 RELATIONSHIP BETWEEN THE SWD IN $\mathbb{R}^d$ AND $\mathbb{R}^k$

In this section, we discuss the proof of the proposition 4.9. We first introduce some intermediate results in the following subsection.

#### A.4.1 RELATIONSHIP BETWEEN $SW_p^p(\mu^d, \nu^d; \mathcal{U}(\mathbb{S}^{d-1}))$ AND $SW_p^p(\mu^d, \nu^d; \mathcal{N}(0, I_d))$

The main result in this section is the following proposition

**Proposition A.9.** *Choose $\mu, \nu \in \mathcal{P}(\mathbb{R}^d)$, we have*

$$
2^{p/2} \frac{\Gamma(\frac{p+d}{2})}{\Gamma(d/2)} SW_p^p(\mu, \nu; \mathcal{U}(\mathcal{S}^{d-1})) = SW_p^p(\mu, \nu; \mathcal{N}(0, I_d)) \tag{31}
$$

**Remark A.10.** *If we replace $\mathcal{N}(0, I_d)$ by $\mathcal{N}(0, \frac{1}{d} I_d)$, the corresponding conclusion has been proved by (Nadjahi et al., 2021, Proposition 1). Thus, we do not claim the above statement and related proof as part of the contribution in this paper. We present this statement and the related proof for the readers' convenience.*

To prove the above statement, first it is straight forward to verify the following:

**Lemma A.11.** *Given $\alpha \in \mathbb{R}$, with abuse of notations, we let $\alpha_\# \mu$ denote the pushforward measure of $\mu$ under mapping $x \mapsto \alpha x$, then we have*

$$|\alpha|^p W_p^p(\mu, \nu) = W_p^p(\alpha_\# \mu, \alpha_\# \nu) \tag{32}$$

*Proof.* If $\alpha = 0$, then both sides are zero, and we've done.

If $\alpha \neq 0$, it is straightforward to verify the following is a well-defined bijection:

$$\Gamma(\mu, \nu) \ni \gamma \mapsto (\alpha \times \alpha)_\# \gamma \in \Gamma(\alpha_\# \mu, \alpha_\# \nu) \tag{33}$$

where $(\alpha \times \alpha)$ denotes the mapping

$$\mathbb{R}^2 \ni (x, x') \mapsto (\alpha x, \alpha x') \in \mathbb{R}^2.$$

Pick $\gamma \in \Gamma(\mu, \nu)$, we have

$$|\alpha|^p \int_{\mathbb{R}^2} |x - y|^p d\gamma(x, y)$$

$$= \int_{\mathbb{R}^2} |\alpha x - \alpha y|^p d\gamma$$

$$= \int_{\mathbb{R}^2} |x - y|^p d(\alpha \times \alpha)_\# \gamma(x, y)$$

Take the infimum for both sides over $\Gamma(\mu, \nu)$, combine it with the fact that (33) is a bijection. We obtain (32). □

Now we introduce the proof of Proposition (A.9).

*Proof.* Suppose $\theta^g \sim \mathcal{N}(0, I_d)$ and let $\theta = \frac{\theta^g}{\|\theta^g\|}$, we have $\theta \sim \mathcal{U}(\mathbb{S}^{d-1})$ by Proposition A.7. Then we have:

$$SW_p^p(\mu, \nu; \mathcal{N}(0, I_d))$$

$$= \mathbb{E}_{\theta^g \sim \mathcal{N}(0, I_d)}[W_p^p(\theta^g_\# \mu, \theta^g_\# \nu)]$$

$$= \mathbb{E}_{\theta^g \sim \mathcal{N}(0, I_d)}[\|\theta^g\|^p W_p^p(\theta_\# \mu, \theta_\# \nu)] \qquad \text{by Lemma A.11}$$

$$= \mathbb{E}_{\theta^g \sim \mathcal{N}(0, I_d)}[\|\theta^g\|^p] \cdot \mathbb{E}_{\theta \sim \mathcal{U}(\mathbb{S}^{d-1})}[W_p^p(\theta_\# \mu, \theta_\# \nu)] \qquad \text{by Proposition A.6 (4)}$$

$$= 2^{p/2} \frac{\Gamma(\frac{p+d}{2})}{\Gamma(d/2)} \cdot SW_p^p(\mu, \nu; \mathcal{U}(\mathbb{S}^{d-1})) \qquad \text{by Proposition A.6 (3).}$$

□

## A.5 PROOF OF PROPOSITION A.4

We adapt notations $V_k, U$ in previous subsection.

**Lemma A.12.** *Suppose $\mu^d, \nu^d$ satisfy assumption 4.1, pick $\theta^d \in \mathbb{R}^d$ and let $\hat{\theta}^k = U^\top \theta^d$ then we have:*

$$\theta_\# \mu^d = \hat{\theta}^k_\# \mu^k, \theta_\# \nu^d = \hat{\theta}^k_\# \nu^k.$$

*Proof.* For each $x \in \text{Span}(U) = V_k$, we have

$$\theta^\top x = P_U(\theta)^\top x + (\theta - P_U(\theta))^\top x$$

$$= P_U(\theta)^\top x + 0 \qquad \text{Since } \theta - P_U(\theta) \in V_k^\perp$$

$$= (UU^\top \theta)^\top x$$

$$= (U^\top \theta)^\top (U^\top x)$$

Thus,

$$\theta^d_\# \mu^d = (U^\top \theta^d)_\# (U)_\# \mu^d = \hat{\theta}^k_\# \mu^k$$

Similarly, we have $\theta^d_\# \nu^d = \hat{\theta}^k_\# \nu^k$ and we complete the proof. □

**Lemma A.13.** *Suppose* $\theta_1^d, \ldots \theta_L^d \overset{i.i.d.}{\sim} \mathcal{U}(\mathbb{S}^{d-1})$ *and let* $\theta_l^k = \frac{U^\top \theta}{\|U^\top \theta\|}, \forall l \in [1 : L]$, *then* $\theta_1^k, \ldots \theta_L^k \overset{i.i.d}{\sim} \mathcal{U}(\mathbb{S}^{k-1})$.

*Proof.* First, since $k < d$, we have

$$\mathcal{U}(\theta^d \in \mathbb{S}^{d-1} : U^\top \theta^d = 0_k) = 0.$$

Thus, with probability 1, $\theta_l^k$ is well-defined.

By Proposition A.7, with probability 1, we can redefine $\theta_1^d, \ldots \theta_N^d$ by the following way:

Suppose $X_1, \ldots X_n \overset{\text{i.i.d.}}{\sim} \mathcal{N}(0, I_d)$, $\theta_l^d = \frac{X_l}{\|X_l\|}$.

Then

$$\theta_l^k = \frac{U^\top \theta_l^d}{\|U^\top \theta_l^d\|} = \frac{U^\top X_l / \|X_l\|}{\|U^\top X_l / \|X_l\|\|} = \frac{U^\top X_l}{\|U^\top X_l\|}$$

Since $U^\top X_l \sim \mathcal{N}(0, I_k)$, we have $\theta_l^k \sim \mathcal{U}(\mathbb{S}^{k-1})$.

Furthermore, since $X_1, \ldots X_N$ are independent, we have $\theta_1^k, \ldots \theta_N^k$ are independent. Thus, $\theta_1^1, \ldots \theta_N^k \overset{\text{i.i.d.}}{\sim} \mathcal{U}(\mathbb{S}^{k-1})$. $\square$

Now we discuss the proof of Proposition 4.7.

*Proof of Proposition .* Pick $\theta^d \in \mathbb{S}^{d-1}$.

We have

$$W(\theta_\#^d \mu^d, \theta_\#^d \nu^d) = W((U^\top \theta)_\# \mu^k, (U^\top \theta)_\# \nu^k) \qquad \text{By lemma A.12}$$

$$= \|U^\top \theta^d\|^p W(\theta_\#^k \mu^k, \theta_\#^k \nu^k) \qquad \text{By lemma A.11}$$

Thus we prove Equation (13).

Now, we pick $\theta_1^d, \ldots \theta_N^d \in \mathbb{S}^{d-1}$, and thus we have:

$$SW_p^p(\mu^k, \nu^k; \frac{1}{L}\sum_{l=1}^{L}\delta_{\theta_l^k})$$

$$= \frac{1}{L}\sum_{l=1}^{L}W_p^p((\theta_l^k)_\#\mu^k, (\theta_l^k)_\#\nu^k) \qquad \theta_l^k = 0_k \text{ if } \|U^\top\theta_l^d\| = 0$$

$$= \frac{1}{L}\sum_{l=1}^{L}\frac{1}{\|U^\top\theta_l^d\|^p}W_p^p((U^\top\theta_l^d)_\#\mu^k, (U^\top\theta_l^d)_\#\nu^k) \qquad \text{By convention } 0 \cdot \frac{1}{0} = 0$$

$$= \frac{1}{L}\sum_{i=1}^{N}\frac{1}{\|U^\top\theta_l^d\|^p}W_p^p((\theta_l^d)_\#\mu^d, (\theta_l^d)_\#\nu^d) \qquad \text{by equation (13)}$$

$$= \widetilde{SW}_p^p\left(\mu^d, \nu^d; \frac{1}{N}\sum_{i=1}^{N}\delta_{\theta_l^d}, \rho\right)$$

And we prove (14).

Similarly, we obtain the last equation,

$$\widetilde{SW}_p^p(\mu^d, \nu^d; \mathcal{U}(\mathbb{S}^{d-1}), h) = \mathbb{E}_{\theta^d \sim \mathcal{U}(\mathbb{S}^{d-1})}\left[\frac{1}{\|U^\top\theta^d\|^p}W_p^p((\theta^d)_\#\mu^d, (\theta^d)_\#\nu^d)\right]$$

$$= \mathbb{E}_{\theta^d \sim \mathcal{U}(\mathbb{S}^{d-1})}\left[W_p^p((\theta^k)_\#\mu^k, (\theta^k)_\#\nu^k)\right] \qquad \text{By equation (13)}$$

$$= \mathbb{E}_{\theta^k \sim \mathcal{U}(\mathbb{S}^{k-1})}[W_p^p((\theta^k)_\#\mu^k, (\theta^k)_\#\nu^k)] \qquad \text{By lemma A.13}$$

$$= SW_k^k(\mu^k, \nu^k)$$

$\square$

### A.5.1 PROOF OF THEOREM 4.9

In this section, we first discuss the relation between $SW_p^p(\mu^d, \nu^d; \mathcal{N}(0, I_d))$ and $SW_p^p(\mu^k, \nu^k; \mathcal{N}(0, I_k))$ under assumption 4.1. Next, we present the proof of Proposition 4.9.

Based on the above lemma, we can derive the following relation between $SW_p^p(\mu^d, \nu^d; \mathcal{N}(0, I_d))$ and $SW_p^p(\mu^k, \nu^k; \mathcal{N}(0, I_k))$.

**Lemma A.14.** *Under assumption 4.1, we have*

$$SW_p^p(\mu^d, \nu^d; \mathcal{N}(0, I_d)) = SW_p^p(\mu^k, \nu^k; \mathcal{N}(0, I_k)) \tag{34}$$

*Proof.* Suppose $\theta^d \sim \mathcal{N}(0, I_d)$ and let $\theta^k = U^\top \theta^d$. Then by proposition A.6 (1), we have $\theta^k \sim \mathcal{N}(0, U^\top I_d U) = \mathcal{N}(0, I_k)$. Therefore,

$$SW_p^p(\mu^d, \nu^d; \mathcal{N}(0, I_d))$$
$$= \mathbb{E}_{\theta^d \sim \mathcal{N}(0, I_d)}[W_p^p(\theta_\#^d \mu^d, \theta_\#^d \nu^d)]$$
$$= \mathbb{E}_{\theta^d \sim \mathcal{N}(0, I_d)}[W_p^p(\theta_\#^k \mu^k, \theta_\#^k \nu^k)] \qquad \text{By lemma A.12, where } \theta^k = U^\top \theta^d$$
$$= \mathbb{E}_{\theta^k \sim \mathcal{N}(0, I_k)}[W_p^p(\theta_\#^k \mu^k, \theta_\#^k \nu^k)]$$
$$= SW_p^p(\mu^k, \nu^k; \mathcal{N}(0, I_k))$$

and we complete the proof. $\square$

Combine the above lemma and proposition A.9, we can prove the Theorem 4.9

*Proof of Theorem 4.9.* For the first equality, we have

$$SW_p^p(\mu^d, \nu^d; \mathcal{U}(\mathbb{S}^{d-1}))$$
$$= \frac{1}{C_d} SW_p^p(\mu^d, \nu^d; \mathcal{N}(0_d, I_d)) \qquad \text{By proposition A.9} \tag{35}$$
$$= \frac{1}{C_d} SW_p^p(\mu^k, \nu^k; \mathcal{N}(0_k, I_k)) \qquad \text{By lemma A.14}$$
$$= \frac{C_k}{C_d} SW_p^p(\mu^k, \nu^k; \mathcal{U}(\mathbb{S}^{k-1})) \qquad \text{By proposition A.9} \tag{36}$$

where $C_d = \frac{\Gamma(p/2 + d/2)}{\Gamma(d/2)}$ and $C_k$ is defined similarly.

$\square$

### A.6 PROOF OF PROPOSITION 4.10

We first introduce the following lemma:

**Lemma A.15.** *Let $I_{d \times k}$ denote the matrix $\begin{bmatrix} I_{k \times k} \\ 0_{(d-k) \times k} \end{bmatrix}$, and suppose $\theta^d \sim \mathcal{U}(\mathbb{S}^{d-1})$, then $\|U^\top \theta^d\|$, $\|I_{d \times k}^\top \theta^d\|$ have same distribution.*

*Proof.* We write SVD decomposition of $U$, since $U$ is orthonormal matrix, we have $U = V_1 I_{d \times k} V_2$ where $V_1 \in \mathbb{R}^{d \times d}, V_2 \in \mathbb{R}^{k \times k}$ are orthogonormal matrix.

Then we have

$$\|U^\top \theta^d\| = \|V_2^\top I_{d \times k}^\top V_1^\top \theta^d\| = \|I_{d \times k}^\top V_1^\top \theta^d\|$$

Since $\theta^d \sim \mathcal{U}(\mathbb{S}^{d-1})$, then $V_1^\top \theta^d \sim \mathcal{U}(\mathbb{S}^{d-1})$.

Thus, $I_{d \times k}^\top \theta^d, I_{d \times k}^\top V_1^\top \theta^d$ have same distribution. Thus $\|I_{d \times k}^\top \theta^d\|, \|I_{d \times k}^\top V_1^\top \theta^d\| = \|U^\top \theta^d\|$ have same distribution. $\square$

Based on this, we can prove the statment (1) in proposition 4.10.

*Proof of Proposition 4.10 (1).* By the above lemma, it is sufficient to consider $U = I_{d \times k}$.

Let $\theta^{d,g} \sim \mathcal{N}(0, I_d)$, and let $\theta^{d,g}[i], i \in [1 : d]$ denote each component of $\theta^{d,g}$. Thus $\theta^{d,g}[1], \ldots \theta^{d,g}[d] \overset{i.i.d}{\sim} \mathcal{N}(0,1)$. We can redefine $\theta^d$ as $\theta^d = \frac{\theta^{d,g}}{\|\theta^{d,g}\|}$, thus,

$$\|U^\top \theta^d\|^2 = \frac{\|U^\top \theta^{d,g}\|^2}{\|\theta^{d,g}\|^2}$$

$$= \frac{\sum_{i=1}^{k} \theta^{d,g}[i]^2}{\sum_{i=1}^{d} \theta^{d,g}[i]^2} \sim \text{Beta}(\frac{k}{2}, \frac{d-k}{2})$$

Thus, we have

$$\mathbb{E}[\|U^\top \theta^d\|^p] = \mathbb{E}[(\|U^\top \theta^d\|^2)^{p/2}] = \frac{\Gamma(k/2 + p/2)\Gamma(d/2)}{\Gamma(k/2)\Gamma(d/2 + p/2)} = \frac{C_k}{C_d}.$$

Note, $\|U^\top \theta_1^d\|, \ldots \|U^\top \theta_L^d\|$ are i.i.d. random variables, thus, we have

$$\mathbb{E}[\widehat{ESSF(L)}] = \frac{1}{L} \mathbb{E}[\sum_{l=1}^{L} \|U^\top \theta_l^d\|^p] = \frac{C_k}{C_d}.$$

Similarly,

$$\text{Var}[\widehat{ESSF(L)}] = \frac{1}{L} \text{Var}[\|U^\top \theta_l^d\|^p]$$

where $\text{Var}[\|U^\top \theta_l^d\|^p] > 0$, is the variance of the $p/2-$th power of a $\text{Beta}(k/2, (d-k)/2)$ variable, which is a constant only depends on $(d, k, p)$.

$\square$

*Proof of Proposition 4.10(2).* For each $\theta$, we have $W_p^p(\theta_\# \mu^d, \theta_\# \nu^d) = \|U^\top \theta^d\|^p W_p^p(\theta_\# \mu^k, \theta_\# \nu^k)$. Thus

$$\epsilon_L = \frac{1}{L} |\sum_{l=1}^{L} (1 - \sum_{l'=1}^{L} \frac{\|U^\top \theta_{l'}\|^p}{\|U^\top \theta_l\|^p}) \underbrace{W_p^p((\theta_l)_\# \mu^d, (\theta_l)_\# \nu^d)}_{A(\theta_l)}|$$

where $A(\theta_l)$ is a function from $\mathbb{S}^{d-1}$ to $\mathbb{R}$ is a function.

Furthermore, from assumption 4.1, we have $A(\theta_l) = A(UU^\top \theta_l)$, thus,

$$|A(\theta_l)| = |A(UU^\top \theta_l)|$$
$$= |W_p^p((UU^\top \theta_l)_\# \mu, (UU^\top \theta_l)_\# \nu)|$$
$$\leq \max_{x \in \text{supp}(\mu), y \in \text{supp}(\nu)} \|UU^\top \theta_l x - UU^\top \theta_l y\|^p$$
$$\leq \underbrace{\max_{x \in \text{supp}(\mu), y \in \text{supp}(\nu)} \|x - y\|^p}_{K} \cdot \|UU^\top \theta_l\|^p \qquad \text{By Cauchy–Schwarz inequality}$$
$$= K \|U^\top \theta_l\|^p \qquad\qquad\qquad\qquad (37)$$

where constant $K < \infty$ since $\mu, \nu$ are supported on compact sets.

Thus, we have that

$$\epsilon_L \leq K \cdot \left| \underbrace{\frac{1}{L} \sum_{l=1}^{L} \left( \|U^\top \theta_l\|^p - \sum_{l'=1}^{L} \|U^\top \theta_{l'}\|^p \right)}_{B_n} \right|.$$

By law of large numbers, with probability 1, $B_n \to 0$. Thus, $\epsilon_L \to 0$.

It remains to show the convergence rate of $\epsilon_L$. Since each $\|U^\top \theta_l\|^p \in [0,1]$, for each $t > 0$, by Hoeffding's we have

$$\mathbb{P}(|B_n| \geq \delta) \leq e^{-2\delta^2 L}$$

Replacing $\epsilon$ by $\epsilon/K$, we have $\mathbb{P}(\text{error}_L \leq \delta) \geq 1 - 2e^{\frac{2\delta^2 L}{K^2}}$ and we complete the proof.

$\square$

### A.6.1 PROOF OF THEOREM 4.10

### A.7 SPECIAL CASE: LEARNING RATE BOUND FOR THE SW GRADIENT FLOW PROBLEM

In this section, we consider the following sliced gradient flow problem Bonet et al. (2021a):

$$\mu_{t+1} \leftarrow \arg \min_{\mu \in \mathcal{P}_2(\mathbb{R}^k)} \frac{1}{2\tau} SW_2^2(\mu, \mu_t) + F(\mu)$$

$$s.t. \mu_0 = \mu^k$$

$$\text{where } F(\mu) := SW_2^2(\mu, \nu^k), \text{for some } \nu^k, \tau > 0$$

In the discrete setting, $\mu^k = \sum_{i=1}^n q_i^1 \delta_{x_i}, \nu^k = \sum_{j=1}^m q_j^2 \delta_{y_j}$. Furthermore, we assume that the pmf of $\mu_t$ is fixed. Then the above problem can be transferred to the following:

$$X^{t+1} \leftarrow X^t - h_t \odot \nabla_X SW_2^2(\mu_t, \nu^k), \text{ where } \mu_t = \sum_{i=1}^n q_i^1 \delta_{x_i^t}, X^t = [x_1, \ldots, x_n] \quad (38)$$

where $\odot$ denote the element-wise product operator, and $h_t \in \mathbb{R}_+^n$.

We will discuss how to select the appropriate learning rate $h_t$.

**Gradient and Hessian of Sliced Wasserstein distance.** First, we discuss the gradient and Hessian matrix of the function $X \mapsto SW_2^2(\mu, \nu^k)$:

Pick $\theta \in \mathbb{S}^{d-1}$ and suppose that $\gamma_\theta$ is an optimal transportation plan for $W_2^2(\theta_\# \mu, \theta_\# \nu^k)$.

Then by Bonneel & Coeurjolly (2019), we have:

$$\nabla_{x_i} W_2^2(\theta_\# \mu, \theta_\# \nu^k) = 2\theta\theta^\top (q_i^1 x_i - \sum_{j=1}^m y_j \gamma_{i,j}^\theta), \forall x_i$$

Note, when $W_2^2(\theta_\# \mu, \theta_\# \nu^k)$ is induced by a Monge mapping, the above formulation can be simplified to $q_i^1 \theta\theta^\top (x_i - T(x_i))$.

Thus the Hessian matrix is

$$\left[ \frac{\partial^2 W_2^2(\theta_\# \mu, \theta_\# \nu^k)}{\partial x_i[l] \partial x_i[l']} \right]_{l,l' \in [1:d]} = 2q_i^1 \theta\theta^\top.$$

Therefore, the gradient for mapping $X \mapsto SW_2^2(\mu, \nu^k)$ with respect to each $x_i$ is given by:

$$g(x_i) := \nabla_{x_i} SW_2^2(\mu, \nu^k) = 2 \int_{\mathbb{S}^{d-1}} \theta\theta^\top (q_i^1 x_i - \sum_{i=1}^m y_j \gamma_{i,j}^\theta) d\mathcal{U}(\mathbb{S}^{d-1})(\theta)$$

$$\approx \frac{2}{N} \sum_{l=1}^L \theta_l \theta_l^\top (q_i^1 x_i - \sum_{j=1}^m y_j \gamma_{i,j}^{\theta_l})$$

where the second line is the Monte carlo approximation.

Similarly, the Hession matrix and the Monte carlo approximation are given by

$$H(x_i) := H_{x_i}(SW_2^2(\mu, \nu^k)) = 2q_i^1 \int \theta \theta^\top d\mathcal{U}(\mathbb{S}^{d-1})(\theta) = 2q_i^1 \frac{1}{k} I_k$$

$$\approx \frac{2q_i^1}{N} \sum_{i=1}^{N} \theta \theta^\top$$

By classical machine learning theory, the optimal learning rate for $x_i$, is given by

$$(h_t)_i = \frac{g(x_i)^\top g(x_i)}{g(x_i)^\top H g(x_i)} = \frac{k}{2q_i^1}, \forall i \in [1:n] \tag{39}$$

**Remark A.16.** *We consider a simplified case to intuitively understand the above learning rate. Suppose $\mu^k = q_i^1 \delta_{x_i}$ and $\nu^k = q_i^1 \delta_{y_j}$ (relaxing the assumption that $\mu^k$ and $\nu^k$ are probability measures). Then, we have:*

$$\begin{aligned}
&SW_2^2(\mu^d, \nu^d) \\
&= q_i^1 \mathbb{E}_{\theta \sim \mathcal{U}(\mathbb{S}^{d-1})}[(\theta^\top x_i - \theta^\top y_j)^2] \\
&= q_i^1 \mathbb{E}_{\theta \sim \mathcal{U}(\mathbb{S}^{d-1})}[\|\theta \theta^\top x_i - \theta \theta^\top y_j\|^2] \\
&= q_i^1 (x_i - y_j)^\top \mathbb{E}[\theta \theta^\top](x_i - y_j) \\
&= \frac{1}{k} q_i^1 \|x_i - y_j\|_2^2 \\
&= \frac{1}{k} W_2^2(\mu, \nu).
\end{aligned}$$

*Thus the gradient with respect to $x_i$ becomes*

$$g(x_i) = 2 \frac{q_i^1}{k}(x_i - y_j).$$

*Letting $t = 0$, we plug the learning rate from (39) and the gradient into (38), obtaining:*

$$x_i^{t+1} \leftarrow x_i^t - (y_j - x_i) = y_j.$$

*Intuitively, the learning rate $(h_t)_i$ for $x_i$ is chosen such that the (negative) gradient becomes the displacement given by the classical OT transportation plan, i.e.,*

$$-g(x_i) \approx y_j - x_i.$$

*That is, when $\theta$ is sufficiently large (i.e., $\frac{1}{L} \sum_{i=1}^{L} \theta \theta^\top \approx \frac{1}{k} I_k$), $\mu_t^k$ will converge to $\nu^k$ in one step.*

## A.8 PROOF OF PROPOSITION 4.12

Pick $x_i$ from $\{x_1, \ldots, x_n\}$. Note, based on assumption 4.1, $x_i = UU^\top x_i = Ux_i^k, \forall i \in [1:n]$. Thus, we have

$$\nabla_{x_i} SW_p^p(\hat{\mu}, \hat{\nu}; \frac{1}{L} \sum_{l=1}^{L} \delta_{\theta_l})$$

$$= U\nabla_{x_i^k} SW_p^p(\hat{\mu}, \hat{\nu}; \frac{1}{L} \sum_{l=1}^{L} \delta_{\theta_l})$$

$$= U\nabla_{x_i^k} \sum_{l=1}^{L} \frac{1}{L} \sum_{i,j} (q_i^1 \|U^\top \theta_l\| (\theta_l^k)^\top (x_i^k - \frac{1}{q_i^1} y_j^k \gamma_{i,j}^{\theta_l}))^p$$

$$= q_i^1 pU \frac{1}{L} \sum_{l=1}^{L} \|U^\top \theta_l\| (\theta_l^k)^\top (x_i^k - \frac{1}{q_i^1} y_j^k \gamma_{i,j}^{\theta_l})^{p-1}.$$

Similarly,

$$\nabla_{x_i} SW_p^p(\hat{\mu}^k, \hat{\nu}^k; \frac{1}{L} \sum_{l=1}^{L} \delta_{\theta_l^k})$$

$$= q_i^1 pU \frac{1}{L} \sum_{l=1}^{L} (\theta_l^k)^\top (x_i^k - \frac{1}{q_i^1} y_j^k \gamma_{i,j}^{\theta_l})^{p-1} \qquad (40)$$

where $\gamma^{\theta_l}$ is the optimal transportation plan for 1D problem $W_p^p((\theta_l)_{\#}\hat{\mu}, (\theta_l)_{\#}\hat{\nu}) = W_p^p((\theta_l^k)_{\#}\hat{\mu}^k, (\theta_l^k)_{\#}\hat{\nu}^k)$.

Thus,

$$\epsilon_L(x_i) = \nabla_{x_i} SW_p^p(\hat{\mu}_d, \hat{\nu}_d; \sum_{l=1}^{L} \delta_{\theta_l^d}) - \widehat{ESSF}(L) \cdot \nabla_{x_i} SW_p^p(\hat{\mu}_k, \hat{\nu}_k; \sum_{l=1}^{L} \delta_{\theta_l^k})$$

$$= \frac{pq_i^1}{L} U \sum_{l=1}^{L} (\|U^\top \theta_l\|^p - \frac{1}{L} \sum_{l'=1}^{L} \|U^\top \theta_{l'}\|^p) \underbrace{\theta_l^k ((\theta_l^k)^\top (x_i^k - \frac{1}{q_i^1} \sum_{j=1}^{m} y_j^k \gamma_{i,j}^{\theta_l}))^{p-1}}_{A(\theta_l^k)}.$$

where $A(\theta_l^k)$ is a vector function from $\mathbb{S}^{k-1}$ to $\mathbb{R}^k$. By Cauchy Schwatz inequality, and the fact $\|\theta_l^k\| = 1$, we have

$$\|A(\theta_l^k)\| \leq \max_{x_i, y_j} \|x_i^k - y_j^k\|^{p-1} = \max_{x_i, y_j} \|x_i - y_j\|^{p-1}$$

Then we have:

$$\|\epsilon_L(x_i)\| = pq_i^1 \left| \underbrace{\sum_{l=1}^{L} \frac{1}{L} \left( \|U^\top \theta_l\|^p - \frac{1}{L} \sum_{l'=1}^{L} \|U^\top \theta_{l'}\|^p \right)}_{B_L} \right| \|UA(\theta_l^k)\|$$

$$= pq_i^1 \|A(\theta_l^k)\| |B_L|$$

$$\leq pq_i^1 K |B_L|.$$

By law of large number, with probability 1, $B_L \to 0$, thus $\|(\epsilon_L)\| \to 0$, that is $\epsilon_L \to 0_d$.

It remains to bound the convergence rate of $\|\epsilon_L\|$.

By Hoeffding inequality and the fact $\|U^\top \theta_l\|^2 \in [0, 1]$, we have

$$\mathbb{P}(|B_L| \geq \epsilon) \leq 2e^{-\epsilon^2 L}.$$

Replacing $\epsilon$ by $\epsilon/(pq_i^1 K)$, we obtain:

$$\mathbb{P}(\|\epsilon_L(x_i)\| \leq \epsilon) \geq 1 - 2e^{-\epsilon^2 L/(pq_i^1 K)^2}.$$

and we complete the proof.

A.9   DISCUSSION WHEN THE ASSUMPTION 4.1 IS NOT SATISFIED.

In this section, we briefly discuss the context when the assumption 4.1 is not satisfied. In particular, we aim to show the following:

**Proposition A.17.** *Let $U, V_k$ be defined in assumption 4.1, choose $\mu^d, \nu^d \in \mathcal{P}_p(\mathbb{R}^d)$, and let $\mu^k, \nu^k$ be defined by $U_{\#}\mu^d, U_{\#}\nu^d$, we claim the following:*

$$W_2^2(\mu^k, \nu^k) \leq W_2^2(\mu^d, \nu^d) \leq W_2^2(\mu^k, \nu^k) + 2(m_2(U_{\#}^\perp \mu^d) + m_2(U_{\#}^\perp \nu^d, p))56 \tag{41}$$

*where $m_p(U_{\#}^\perp \mu^d)$ the $p - th$ moment of measure $U_{\#}^\perp \mu^d$.*

*Proof.* For each pair $(x, y) \in (\mathbb{R}^d)^2$, we have

$$\|P_U(x) - P_U(y)\|^2$$
$$\leq \|x - y\|^2 \qquad\qquad\qquad \text{By definition of projection} \tag{42}$$
$$= \|P_U(x) - P_U(y)\|^2 + \|P_{U^\perp}(x) - P_{U^\perp}(y)\|^2 \qquad\qquad \text{Pythagorean theorem}$$
$$\leq \|P_U(x) - P_U(y)\|^2 + 2\|(U^\perp)^\top x\|^2 + 2\|(U^\perp)^\top y\|^2 \tag{43}$$

From (42), we have

$$W_2^2((P_U)_{\#}\mu^d, (P_U)_{\#}\nu^d) \leq W_2^2(\mu^d, \nu^d).$$

Combined it with Proposition A.4, we have:

$$W_2^2(\mu^k, \nu^k) = W_2^2((P_U)_{\#}\mu^d, (P_U)_{\#}\nu^d) \leq W_2^2(\mu^d, \nu^d),$$

and we prove the first inequality in (41).

Similarly, let $\gamma \in \Gamma(\mu, \nu)$ be the optimal transportation plan for $W_2^2((P_U)_{\#}\mu^d, (P_U)_{\#}\nu^d)$. From (43), we have:

$$W_2^2(\mu^d, \nu^d)$$
$$\leq \mathbb{E}_{(X,Y)\sim\gamma}[\|X - Y\|^2]$$
$$\leq \mathbb{E}_{(X,Y)\sim\gamma}[\|P_U(X) - P_U(Y)\|^2 + 2\|(U^\perp)^\top X\|^2 + 2\|(U^\perp)^\top X\|^2] \qquad \text{By (43)}$$
$$= W_2^2((P_U)_{\#}\mu, (P_U)_{\#}\nu) + 2(m_2((U^\perp)_{\#}\mu^d) + m_2((U^\perp)_{\#}\nu^d))$$

Thus, we prove the second inequality of (41). $\qquad\qquad\square$

**Proposition A.18.** *Based on the same notations of Proposition A.17, we have:*

$$\frac{k}{d}SW_2^2(\mu^k, \nu^k) \leq SW_2^2(\mu^d, \nu^d) \leq \frac{k}{d}SW_2^2(\mu^k, \nu^k) + 2\frac{d-k}{d}(m_2(U_{\#}^\perp \mu^d) + m_2(U_{\#}^\perp \nu^d)) \tag{44}$$

*Proof.* Pick $\theta \in \mathbb{S}^{d-1}$ and $x, y \in \mathbb{R}^d$. We have:

$$\|\theta^\top P_U(x) - \theta^\top P_U(y)\|^2$$
$$\leq \|\theta^\top x - \theta^\top y\|^2 \tag{45}$$
$$= \|P_U(\theta)^\top P_U(x) - P_U(\theta)^\top P_U(y)\|^2 + \|P_{U^\perp}(\theta)^\top P_{U^\perp}(x) - P_{U^\perp}(\theta)^\top P_{U^\perp}(y)\|^2$$
$$\leq \|\theta^\top P_U(x) - \theta^\top P_U(y)\|^2 + \|(U^\perp)^\top \theta\|^2 \|(U^\perp)^\top x - (U^\perp)^\top y\|^2 \qquad \text{Cauchy Schwatz inequality}$$
$$\leq \|\theta^\top P_U(x) - \theta^\top P_U(y)\|^2 + 2\|(U^\perp)^\top \theta\|^2 (\|(U^\perp)^\top x\|^2 + \|(U^\perp)^\top y\|^2) \tag{46}$$

Choose $\theta \in \mathbb{S}^{d-1}$. From Proposition A.17, we have

$$W_2^2(\theta_\#(P_U)_\#\mu^d, \theta_\#(P_U)_\#\nu^d) \leq W_2^2(\theta_\#\mu^d, \theta_\#\nu^d)$$

Take expected value with respect to $\theta$, we have

$$SW_2^2((P_U)_\#\mu^d, (P_U)_\#\nu^d) \leq SW_2^2(\mu^d, \nu^d)$$

Combine it with Theorem 4.9, we prove the first inequality in (44).

Similarly, from (46), we have

$$W_2^2(\theta_\#\mu^d, \theta_\#\nu^d) \leq W_2^2(\theta_\#(P_U)_\#\mu^d, \theta_\#(P_U)_\#\nu^d) + 2\|(U^\perp)^\top\theta\|^2(m_2(U_\#^\perp\mu^d) + m_2(U_\#^\perp\nu^d)).$$

Take expected value with respect to $\theta$, we obtain:

$$\begin{aligned}
SW_2^2(\mu^d, \nu^d) \\
\leq SW_2^2((P_U)_\#\mu^d, (P_U)_\#\nu^d) + 2\mathbb{E}_\theta[\|U^\perp\theta\|^2](m_2(U_\#^\perp\mu^d) + m_2(U_\#^\perp\nu^d)) \\
= \frac{k}{d}SW_2^2(\mu^k, \nu^k) + 2\frac{d-k}{d}(m_2(U_\#^\perp\mu^d) + m_2(U_\#^\perp\nu^d))
\end{aligned}$$

there the last equality holds from Theorem 4.9 and the fact $\|U^\perp\theta\|^2 \sim \text{Beta}(\frac{d-k}{2}, \frac{k}{2})$.

$\square$

## B  ADDITIONAL DETAILS FOR THE NUMERICAL EXPERIMENTS

### B.1  GRADIENT FLOW

#### B.1.1  BACKGROUND OVERVIEW

Let $\mathcal{P}(\mathbb{R}^d)$ denote the space of probability measures on $\mathbb{R}^d$. For $\mu, \nu \in \mathcal{P}(\mathbb{R}^d)$, the gradient flow of the SW distance in the space of probability measures evolves according to the continuity equation

$$\frac{\partial \mu_t}{\partial t} + \nabla \cdot (v_t \mu_t) = 0, \tag{47}$$

where $\mu_t$ is a time-dependent probability measure and $v_t$ the velocity field $v_t = -\nabla \frac{\delta SW_2^2(\mu_t, \nu)}{\delta \mu_t}$. This describes the transport of measure $\mu_t$ in the Wasserstein space $\mathcal{P}_2(\mathbb{R}^d)$, commonly referred to as the **Wasserstein Gradient Flows** (WGF, Ambrosio et al. (2008))

For numerical simulation in practice, one discretizes this dynamic using a particle approximation. We let $\{x_t^i\}_{i=1}^N$ denote a system of $N$ particles evolving according to the following system of ODEs:

$$\frac{dx_i^t}{dt} = -\nabla_{x_i} SW_2^2(\mu_t^N, \nu), \tag{48}$$

where $\mu_t^N = \frac{1}{L} \sum_{i=1}^L \delta_{x_i^t}$ is the empirical measure based on the particle positions $x_i^t$.

These WGF particle-based approaches preserve key features of continuous systems and have been widely adopted, especially machine learning applications (Peyré et al. (2019).

#### B.1.2  EXPERIMENTS

**On classic synthetic datasets**

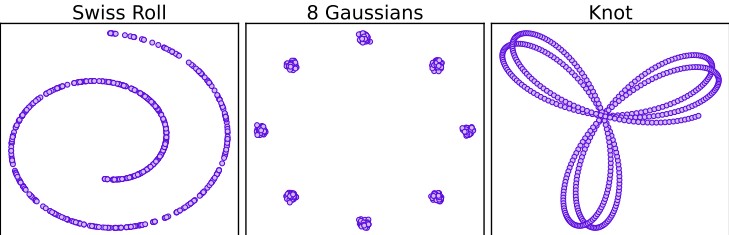

Figure 9: Classic synthetic 2D datasets (shown) embedded in spaces of different target dimensions.

Table 1: Quantitative comparison of the best final converged $W_2(\downarrow)$ and runtime ($\downarrow$) between different variants for Gradient Flow with (embedded) classic synthetic datasets.

| Met. | Swiss | | | 8 Gauss. | | | Knot | | | RT(s)↓ |
|---|---|---|---|---|---|---|---|---|---|---|
| | $d=2$ | $d=50$ | $d=100$ | $d=2$ | $d=50$ | $d=100$ | $d=2$ | $d=50$ | $d=100$ | |
| SW | $0.0001^{\pm 0.0000}$ | $0.0004^{\pm 0.0000}$ | $0.0004^{\pm 0.0000}$ | $0.0002^{\pm 0.0000}$ | $0.0002^{\pm 0.0001}$ | $0.0006^{\pm 0.0001}$ | $0.0002^{\pm 0.0000}$ | $0.0004^{\pm 0.0000}$ | $0.0004^{\pm 0.0000}$ | $8.62^{\pm 0.04}$ |
| MaxSW | $0.0000^{\pm 0.0000}$ | $0.0219^{\pm 0.0051}$ | $0.0342^{\pm 0.0022}$ | $0.0005^{\pm 0.0000}$ | $0.0171^{\pm 0.0004}$ | $0.0385^{\pm 0.0006}$ | $0.0005^{\pm 0.0000}$ | $0.0246^{\pm 0.0009}$ | $0.0303^{\pm 0.0009}$ | $74.02^{\pm 1.61}$ |
| DSW | $0.0002^{\pm 0.0001}$ | $0.0004^{\pm 0.0000}$ | $0.0004^{\pm 0.0000}$ | $0.0002^{\pm 0.0001}$ | $0.0004^{\pm 0.0000}$ | $0.0006^{\pm 0.0001}$ | $0.0003^{\pm 0.0000}$ | $0.0004^{\pm 0.0000}$ | $0.0004^{\pm 0.0000}$ | $162.25^{\pm 0.20}$ |
| MaxKSW | $0.0002^{\pm 0.0000}$ | $0.0124^{\pm 0.0082}$ | $0.0122^{\pm 0.0010}$ | $0.0002^{\pm 0.0000}$ | $0.0154^{\pm 0.0001}$ | $0.0216^{\pm 0.0007}$ | $0.0002^{\pm 0.0001}$ | $0.0165^{\pm 0.0048}$ | $0.0171^{\pm 0.0048}$ | $125.23^{\pm 0.54}$ |
| iMSW | $0.0001^{\pm 0.0000}$ | $0.0021^{\pm 0.0001}$ | $0.0050^{\pm 0.0001}$ | $0.0001^{\pm 0.0000}$ | $0.0021^{\pm 0.0001}$ | $0.0059^{\pm 0.0001}$ | $0.0002^{\pm 0.0001}$ | $0.0034^{\pm 0.0001}$ | $0.0054^{\pm 0.0001}$ | $74.45^{\pm 0.03}$ |
| viMSW | $0.0002^{\pm 0.0001}$ | $0.0003^{\pm 0.0000}$ | $0.0005^{\pm 0.0000}$ | $0.0003^{\pm 0.0001}$ | $0.0003^{\pm 0.0001}$ | $0.0008^{\pm 0.0000}$ | $0.0003^{\pm 0.0001}$ | $0.0005^{\pm 0.0000}$ | $0.0005^{\pm 0.0000}$ | $255.76^{\pm 0.28}$ |
| oMSW | $0.0001^{\pm 0.0000}$ | $0.0002^{\pm 0.0000}$ | $0.0005^{\pm 0.0000}$ | $0.0002^{\pm 0.0001}$ | $0.0002^{\pm 0.0000}$ | $0.0006^{\pm 0.0000}$ | $0.0002^{\pm 0.0001}$ | $0.0004^{\pm 0.0000}$ | $0.0004^{\pm 0.0000}$ | $16.55^{\pm 0.01}$ |
| rMSW | $0.0002^{\pm 0.0001}$ | $0.0003^{\pm 0.0000}$ | $0.0005^{\pm 0.0000}$ | $0.0003^{\pm 0.0001}$ | $0.0003^{\pm 0.0001}$ | $0.0008^{\pm 0.0000}$ | $0.0003^{\pm 0.0000}$ | $0.0006^{\pm 0.0001}$ | $0.0005^{\pm 0.0000}$ | $179.70^{\pm 1.08}$ |
| EBSW | $0.0002^{\pm 0.0001}$ | $0.0002^{\pm 0.0000}$ | $0.0005^{\pm 0.0000}$ | $0.0001^{\pm 0.0000}$ | $0.0002^{\pm 0.0000}$ | $0.0006^{\pm 0.0000}$ | $0.0003^{\pm 0.0001}$ | $0.0004^{\pm 0.0000}$ | $0.0002^{\pm 0.0000}$ | $9.66^{\pm 1.15}$ |
| RPSW | $0.0001^{\pm 0.0000}$ | $0.0001^{\pm 0.0000}$ | $0.0004^{\pm 0.0000}$ | $0.0002^{\pm 0.0000}$ | $0.0001^{\pm 0.0000}$ | $0.0010^{\pm 0.0000}$ | $0.0002^{\pm 0.0000}$ | $0.0001^{\pm 0.0000}$ | $0.0004^{\pm 0.0000}$ | $19.47^{\pm 0.03}$ |
| EBRPSW | $0.0007^{\pm 0.0002}$ | $0.0002^{\pm 0.0001}$ | $0.0003^{\pm 0.0000}$ | $0.0002^{\pm 0.0001}$ | $0.0002^{\pm 0.0001}$ | $0.0006^{\pm 0.0000}$ | $0.0002^{\pm 0.0001}$ | $0.0004^{\pm 0.0000}$ | $0.0002^{\pm 0.0000}$ | $20.30^{\pm 0.05}$ |

**On the MNIST and CelebA images**

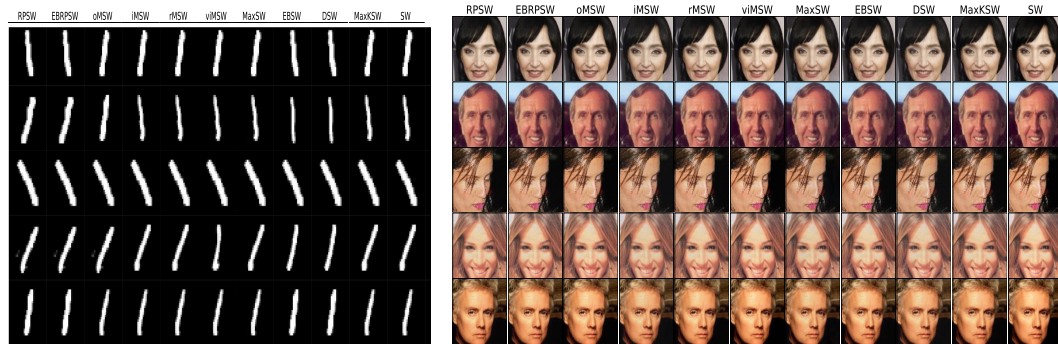

Figure 10: Gradient Flow visualization for images from the MNIST dataset (left) and the CelebA dataset (right).

| Method | MNIST (s) ↓ | CelebA (s)↓ |
|---|---|---|
| DSW | $12500.00^{\pm 0.00}$ | $126054.85^{\pm 745.89}$ |
| EBSW | $686.18^{\pm 45.31}$ | $6694.50^{\pm 148.08}$ |
| RPSW | $800.36^{\pm 5.97}$ | $6038.05^{\pm 86.32}$ |
| EBRPSW | $699.33^{\pm 9.70}$ | $3171.20^{\pm 582.31}$ |
| oMSW | $482.29^{\pm 8.46}$ | $3808.34^{\pm 475.66}$ |
| iMSW | $1359.97^{\pm 10.19}$ | $3601.99^{\pm 11.01}$ |
| rMSW | $1115.98^{\pm 150.49}$ | $100358.26^{\pm 1002.95}$ |
| viMSW | $4161.11^{\pm 16.05}$ | $96007.74^{\pm 937.50}$ |
| MaxSW | $7231.97^{\pm 70.28}$ | $9780.51^{\pm 485.71}$ |
| MaxKSW | $6891.43^{\pm 35.52}$ | $65560.25^{\pm 332.10}$ |
| SW | $441.41^{\pm 36.85}$ | $3335.51^{\pm 76.52}$ |

Table 2: Runtime comparison for all methods in the MNIST/CelebA setups

## B.2 COLOR TRANSFER

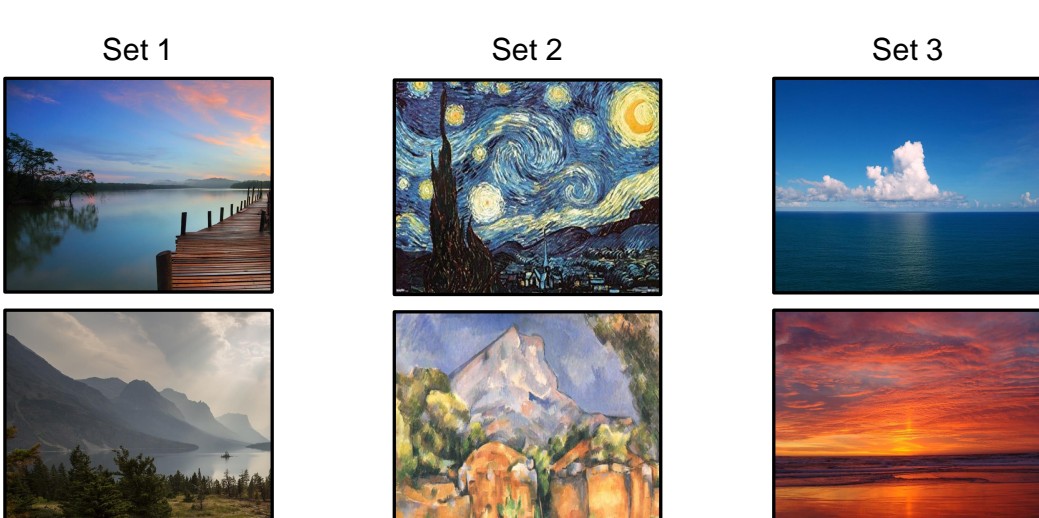

Figure 11: 3 sets of source (top) and target (bottom) images for Color Transfer.

Table 3: Quantitative comparison of the best final converged
$W_2 \downarrow$ and runtime $\downarrow$ between different variants for Color Transfer.

| Method | Best $W_2 \downarrow$ (LR) | | | Runtime(s) $\downarrow$ |
|---|---|---|---|---|
| | Set 1 | Set 2 | Set 3 | |
| SW | $0.01^{\pm 0.00}$ (1e-1) | $0.01^{\pm 0.00}$ (8e-1) | $0.00^{\pm 0.00}$ (1e0) | $8.62^{\pm 0.04}$ |
| MaxSW | $0.03^{\pm 0.00}$ (1e-3) | $0.03^{\pm 0.00}$ (3e-4) | $0.03^{\pm 0.00}$ (3e-4) | $74.02^{\pm 1.61}$ |
| DSW | $0.03^{\pm 0.00}$ (1e-3) | $0.03^{\pm 0.00}$ (1e-3) | $0.03^{\pm 0.00}$ (8e-4) | $162.25^{\pm 0.20}$ |
| MaxKSW | $0.03^{\pm 0.00}$ (5e-4) | $0.03^{\pm 0.00}$ (5e-4) | $0.03^{\pm 0.00}$ (5e-4) | $125.23^{\pm 0.54}$ |
| iMSW | $0.03^{\pm 0.00}$ (1e-3) | $0.03^{\pm 0.00}$ (1e-4) | $0.03^{\pm 0.00}$ (1e-3) | $74.45^{\pm 0.03}$ |
| viMSW | $0.03^{\pm 0.00}$ (1e-3) | $0.03^{\pm 0.00}$ (1e-4) | $0.03^{\pm 0.00}$ (1e-3) | $255.76^{\pm 0.28}$ |
| oMSW | $0.03^{\pm 0.00}$ (1e-3) | $0.03^{\pm 0.00}$ (1e-3) | $0.03^{\pm 0.00}$ (1e-3) | $16.55^{\pm 0.01}$ |
| rMSW | $0.03^{\pm 0.00}$ (1e-3) | $0.03^{\pm 0.00}$ (1e-3) | $0.03^{\pm 0.00}$ (1e-3) | $179.70^{\pm 1.08}$ |
| EBSW | $0.03^{\pm 0.00}$ (1e-3) | $0.03^{\pm 0.00}$ (1e-3) | $0.03^{\pm 0.00}$ (1e-3) | $9.66^{\pm 1.15}$ |
| RPSW | $0.10^{\pm 0.00}$ (3e-3) | $0.10^{\pm 0.00}$ (1e-2) | $0.10^{\pm 0.09}$ (1e-3) | $19.47^{\pm 0.03}$ |
| EBRPSW | $0.03^{\pm 0.00}$ (8e-4) | $0.03^{\pm 0.00}$ (5e-4) | $0.03^{\pm 0.00}$ (5e-4) | $20.30^{\pm 0.05}$ |

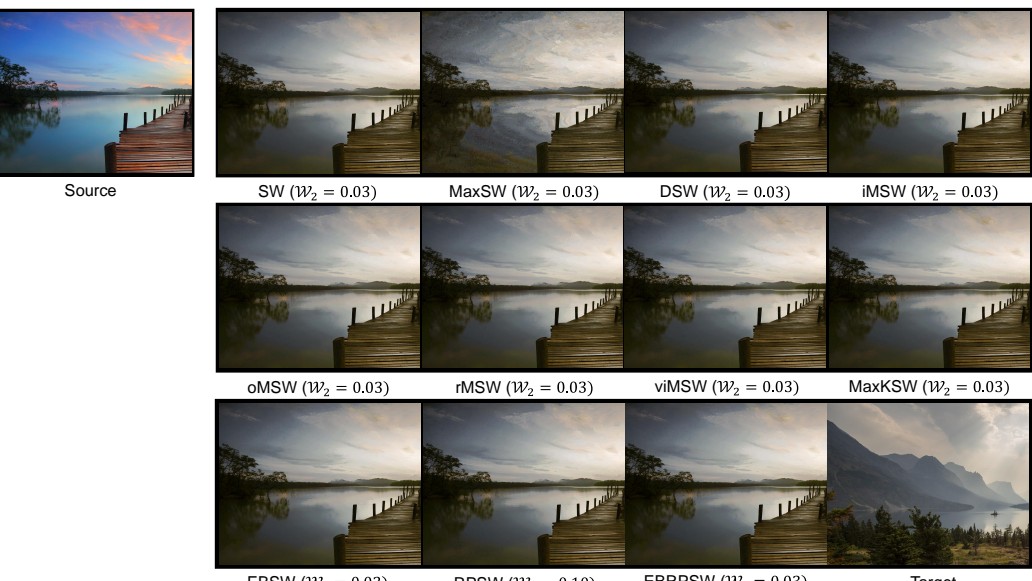

Figure 12: Color Transfer (Set 1)

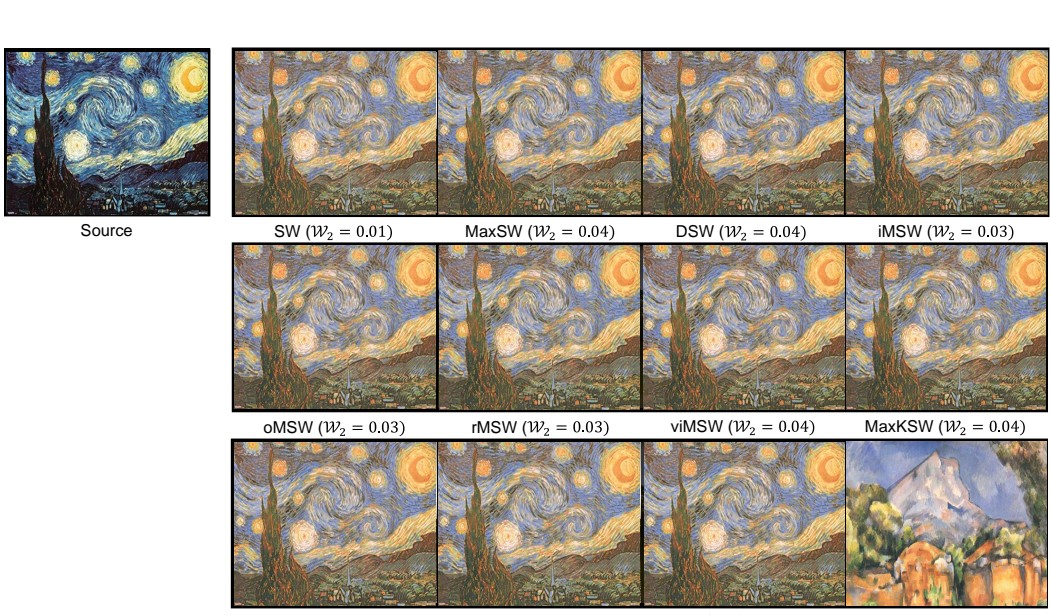

Figure 13: Color Transfer (Set 2)

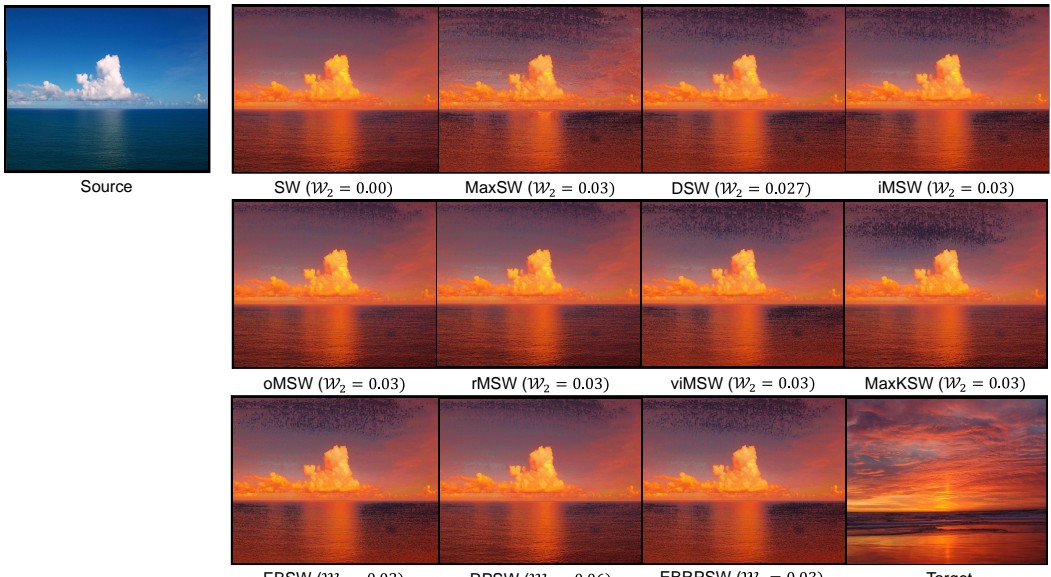

Figure 14: Color Transfer (Set 3)

## B.3 DEEP GENERATIVE MODELING

In this section, we provide additional details on the model architecture and numerical results for the deep generative modeling tasks. We present both qualitative and quantitative results, including the $W_2$ metric, to evaluate the model's performance across different SW variants. The experiments were conducted using the FFHQ dataset.

**Model architecture:**

For unconditional generation:

$$G(z) = \text{FC}_d \circ \text{LeakyReLU}_{0.2} \circ \text{BN} \circ \text{FC}_{1024}$$
$$\circ \text{LeakyReLU}_{0.2} \circ \text{BN} \circ \text{FC}_{512}$$
$$\circ \text{LeakyReLU}_{0.2} \circ \text{BN} \circ \text{FC}_{256}(z), \quad z \in \mathbb{R}^8, G(z) \in \mathbb{R}^{512}$$

For unpaired translation:

$$T(z) = z + \text{FC}_d \circ \text{LeakyReLU}_{0.2} \circ \text{BN} \circ \text{FC}_{1024}$$
$$\circ \text{LeakyReLU}_{0.2} \circ \text{BN} \circ \text{FC}_{1024}(z), \quad z \in \mathbb{R}^{512} \tag{49}$$

Table 4: Quantitative comparison between different variants for Deep Generative Modeling 5.4.

| Method | Unconditional Gen. | | | Unpaired Translation | | | |
|---|---|---|---|---|---|---|---|
| | $W_2 \downarrow$ (LR) | | RT(s) $\downarrow$ | M2F: $W_2 \downarrow$ (LR) | RT(s)$\downarrow$ | A2C: $W_2 \downarrow$ (LR) | RT(s)$\downarrow$ |
| SW | $14.67^{\pm.01}$ (1e0) | | $26.69^{\pm.23}$ | $14.15^{\pm.02}$ (5e-1) | $25.47^{\pm.09}$ | $14.58^{\pm.03}$ (1e0) | $27.94^{\pm.14}$ |
| MaxSW | $13.38^{\pm.17}$ (1e-2) | | $95.83^{\pm.24}$ | $14.01^{\pm.02}$ (5e-3) | $102.68^{\pm.03}$ | $14.52^{\pm.02}$ (3e-3) | $103.29^{\pm2.98}$ |
| DSW | $14.35^{\pm.06}$ (8e-3) | | $197.70^{\pm.14}$ | $14.11^{\pm.02}$ (5e-3) | $198.42^{\pm.38}$ | $14.60^{\pm.04}$ (5e-3) | $198.08^{\pm.32}$ |
| MaxKSW | $15.22^{\pm.03}$ (1e-2) | | $53.38^{\pm3.16}$ | $14.20^{\pm.01}$ (5e-2) | $45.90^{\pm.04}$ | $14.65^{\pm.02}$ (5e-2) | $45.73^{\pm.08}$ |
| iMSW | $14.27^{\pm.01}$ (1e-3) | | $90.27^{\pm.23}$ | $14.06^{\pm.01}$ (3e-3) | $92.70^{\pm.12}$ | $14.59^{\pm.01}$ (1e-3) | $92.65^{\pm.17}$ |
| viMSW | $15.16^{\pm.03}$ (1e-3) | | $49.26^{\pm.05}$ | $14.09^{\pm.01}$ (3e-3) | $271.10^{\pm.42}$ | $14.57^{\pm.01}$ (3e-3) | $271.09^{\pm.84}$ |
| oMSW | $14.81^{\pm.04}$ (5e-2) | | $29.34^{\pm.18}$ | $14.12^{\pm.01}$ (8e-2) | $31.39^{\pm.05}$ | $14.58^{\pm.03}$ (1e-2) | $31.23^{\pm.04}$ |
| rMSW | $14.80^{\pm.07}$ (5e-2) | | $193.77^{\pm.63}$ | $14.16^{\pm.01}$ (8e-2) | $195.07^{\pm.12}$ | $14.60^{\pm.02}$ (8e-3) | $195.81^{\pm.27}$ |
| EBSW | $16.68^{\pm.19}$ (3e-3) | | $22.16^{\pm.07}$ | $14.71^{\pm.02}$ (1e-2) | $26.67^{\pm.00}$ | $15.09^{\pm.04}$ (8e-4) | $26.68^{\pm.02}$ |
| RPSW | $14.46^{\pm.06}$ (3e-2) | | $33.35^{\pm.07}$ | $14.14^{\pm.00}$ (1e-1) | $37.11^{\pm.10}$ | $14.60^{\pm.01}$ (1e-2) | $36.99^{\pm.12}$ |
| EBRPSW | $16.90^{\pm.22}$ (3e-3) | | $34.40^{\pm.07}$ | $14.69^{\pm.02}$ (1e-2) | $38.15^{\pm.05}$ | $15.10^{\pm.05}$ (1e-3) | $38.14^{\pm.11}$ |

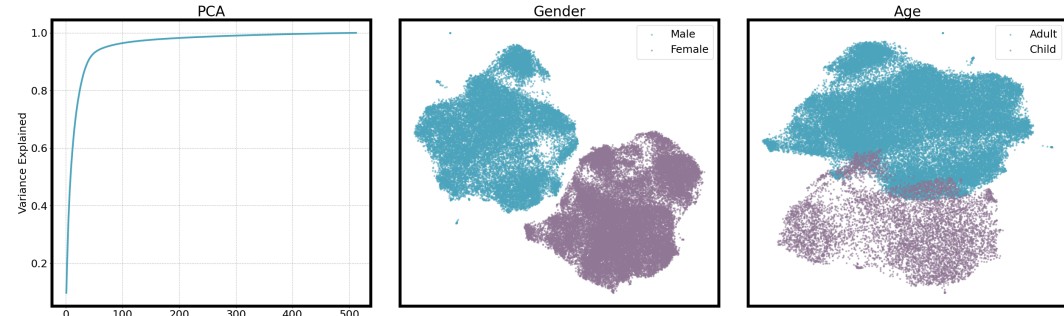

(a) **Left:** Cumulative Explaning Variance plot for the FFHQ latents. **Middle/Right:** UMAP visualization of the Gender and Age splits.

| FFHQ Subset | Train size | Test size |
|---|---|---|
| Adults ($\geq 18$) | 48786 | 8104 |
| Children ($< 10$) | 8345 | 1405 |
| Male | 26732 | 4351 |
| Female | 32816 | 5572 |

(b) Subset size.

Figure 15: FFHQ dataset (Karras et al. (2019))

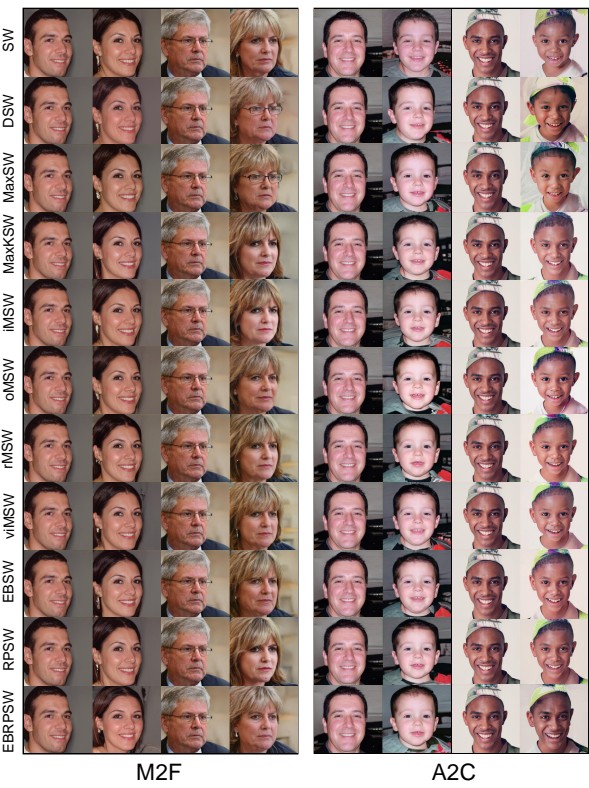

Figure 16: Samples generated by the M2F (left) and (A2C) residual translators.

## B.4 DETAILED NUMERICAL RESULTS

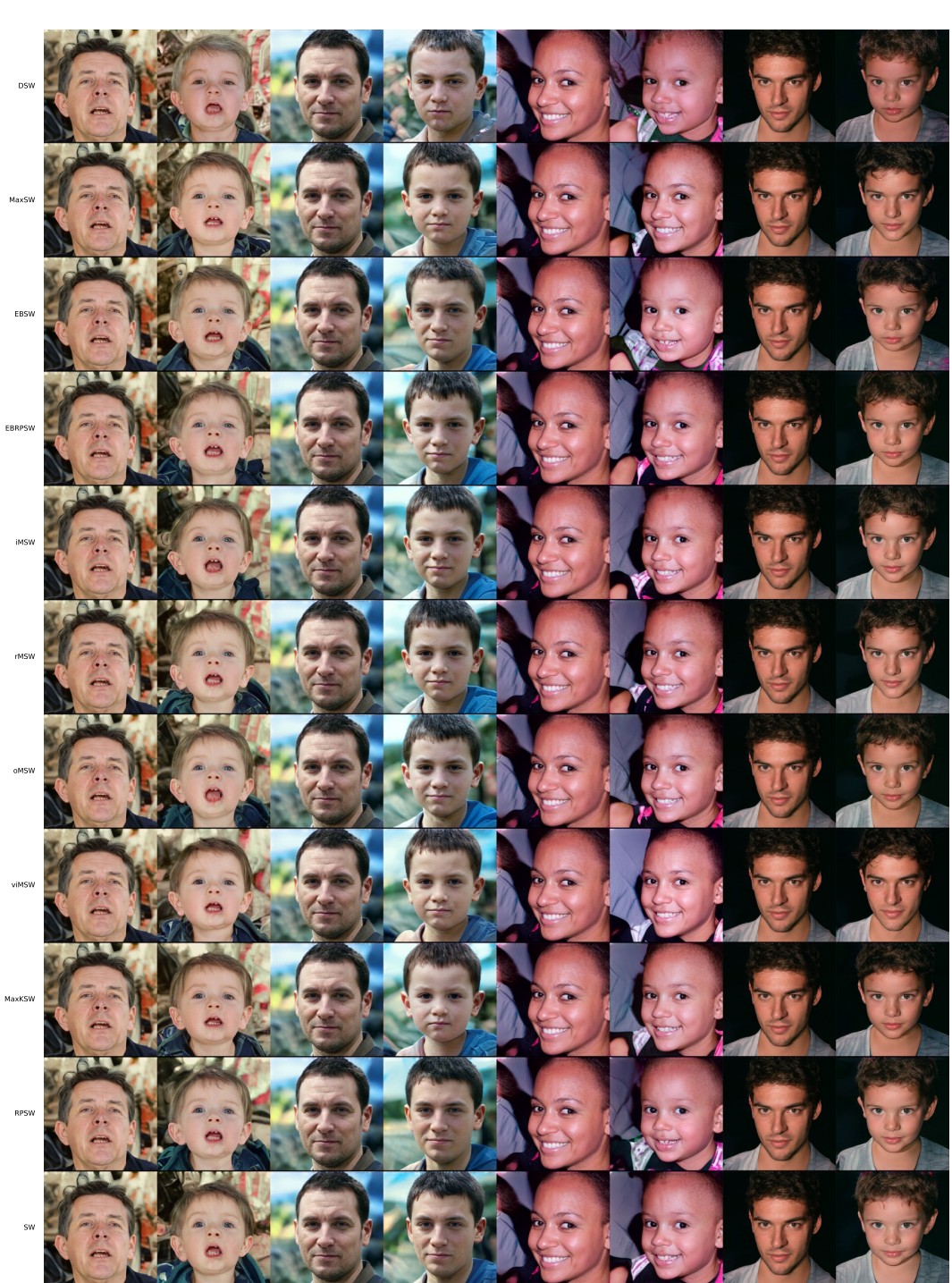

Figure 17: Visualization for the A2C translation task (using the model with the lowest $W_2$ for each method).

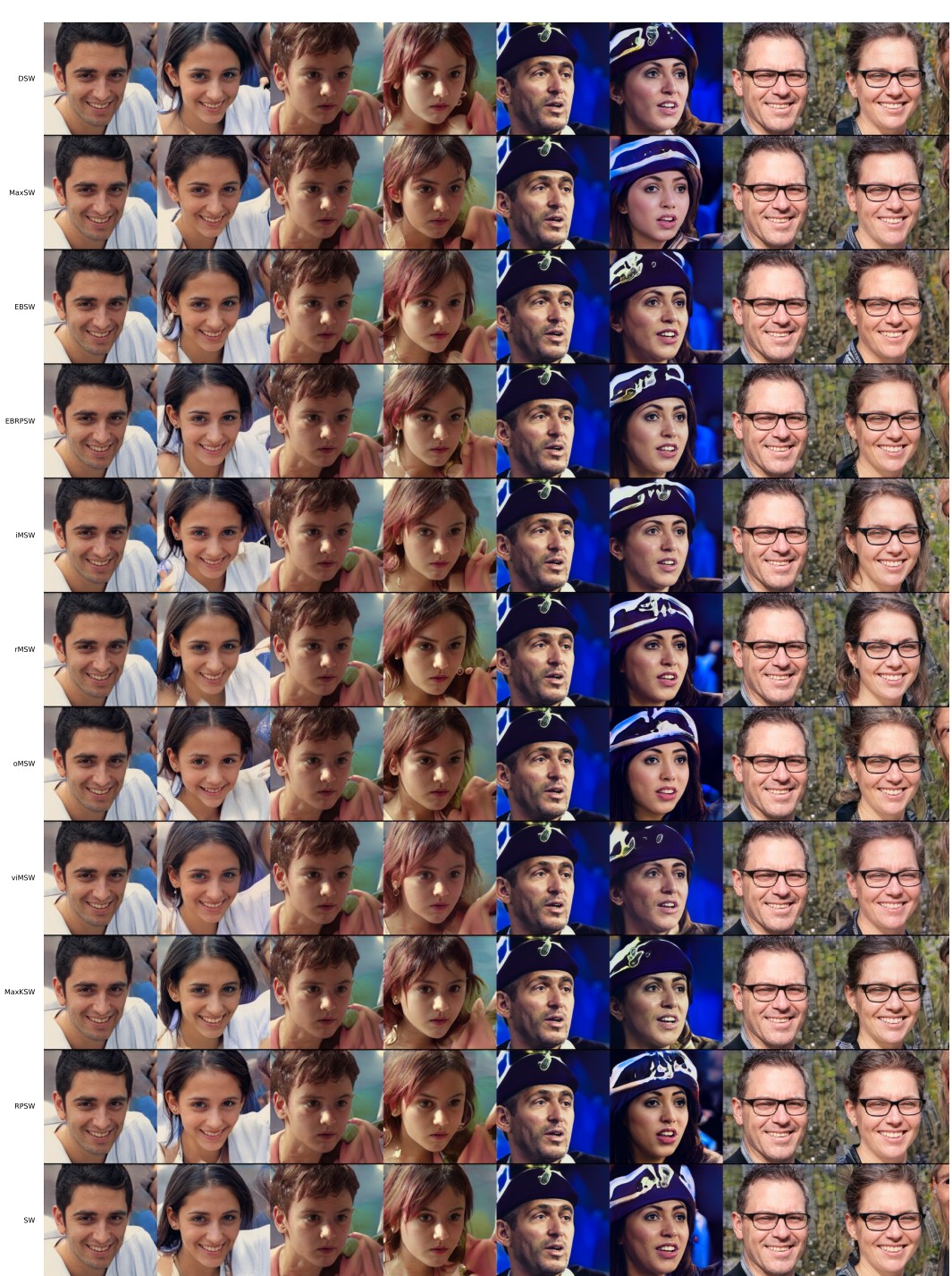

Figure 18: Visualization for the M2F translation task (using the model with the lowest $W_2$ for each method).

| LR | SW | MaxSW | DSW | MaxKSW | iMSW | viMSW | oMSW | rMSW | EBSW | RPSW | EBRPSW |
|---|---|---|---|---|---|---|---|---|---|---|---|
| 100 | 2.3681±0.3508 | 0.5442±0.0195 | 1.8232±1.2533 | 1.2128±0.0685 | 3.8247±0.6643 | 3.1542±0.3530 | 1.2212±0.0565 | 1.8336±0.2363 | 2.9758±0.0588 | 2.7564±0.1211 | 3.0755±0.1628 |
| 80 | 1.7578±0.4630 | 0.5171±0.0098 | 0.9759±0.5058 | 1.3729±0.0088 | 2.3046±0.5586 | 1.8896±0.9034 | 1.3554±0.1909 | 1.8872±0.4225 | 2.4539±0.0358 | 2.1423±0.0344 | 2.2756±0.0813 |
| 50 | 0.8872±0.3438 | 0.9464±0.5872 | 0.9317±0.5872 | 1.6624±0.0214 | 1.2846±0.1253 | 1.4466±0.0949 | 1.4180±0.1613 | 1.1241±0.3024 | 1.4591±0.2815 | 1.1420±0.0967 | 1.4347±0.1352 |
| 30 | 0.6367±0.2107 | 0.9107±0.3422 | 0.5040±0.2023 | 0.9698±0.0011 | 0.7317±0.2280 | 0.7519±0.0310 | 0.9445±0.3024 | 0.9445±0.2815 | 0.3243±0.0414 | 0.7251±0.0681 | 0.9207±0.2912 |
| 10 | 0.2152±0.0336 | 0.4321±0.0154 | 0.2632±0.1356 | 0.1576±0.0045 | 0.2419±0.0488 | 0.1995±0.0240 | 0.2052±0.0648 | 0.2076±0.0414 | 0.1241±0.3024 | 0.2212±0.0339 | 0.3707±0.0406 |
| 8 | 0.2267±0.0385 | 0.3502±0.0581 | 0.1917±0.0742 | 0.2063±0.0025 | 0.2307±0.0693 | 0.1746±0.0061 | 0.1563±0.0275 | 0.1845±0.0750 | 0.1792±0.1090 | 0.1613±0.0202 | 0.3136±0.0649 |
| 5 | 0.1355±0.0168 | 0.2353±0.0056 | 0.1060±0.0565 | 0.1113±0.0150 | 0.1258±0.0381 | 0.1097±0.0052 | 0.1097±0.0381 | 0.1307±0.0414 | 0.1225±0.0701 | 0.1417±0.0255 | 0.2239±0.0023 |
| 3 | 0.0612±0.0111 | 0.1480±0.0087 | 0.0575±0.0198 | 0.0793±0.0036 | 0.0774±0.0143 | 0.0644±0.0023 | 0.0644±0.0143 | 0.0596±0.0094 | 0.1042±0.0027 | 0.0392±0.0146 | 0.1115±0.0030 |
| 1 | 0.0286±0.0079 | 0.0715±0.0020 | 0.0221±0.0119 | 0.0473±0.0016 | 0.0224±0.0038 | 0.0208±0.0004 | 0.0208±0.0038 | 0.0245±0.0041 | 0.0210±0.0021 | 0.0176±0.0076 | 0.0216±0.0022 |
| 8×10⁻¹ | 0.0182±0.0067 | 0.0631±0.0046 | 0.0165±0.0049 | 0.0429±0.0012 | 0.0197±0.0031 | 0.0181±0.0007 | 0.0181±0.0031 | 0.0205±0.0026 | 0.0178±0.0048 | 0.0233±0.0054 | 0.0151±0.0021 |
| 5×10⁻¹ | 0.0144±0.0058 | 0.0492±0.0012 | 0.0100±0.0012 | 0.0353±0.0008 | 0.0123±0.0008 | 0.0118±0.0009 | 0.0118±0.0009 | 0.0121±0.0051 | 0.0109±0.0001 | 0.0117±0.0049 | 0.0091±0.0014 |
| 3×10⁻¹ | 0.0063±0.0025 | 0.0365±0.0014 | 0.0040±0.0026 | 0.0238±0.0014 | 0.0079±0.0014 | 0.0063±0.0003 | 0.0063±0.0003 | 0.0062±0.0020 | 0.0051±0.0010 | 0.0083±0.0052 | 0.0052±0.0019 |
| 1×10⁻¹ | 0.0014±0.0005 | 0.0099±0.0001 | 0.0023±0.0003 | 0.0041±0.0008 | 0.0023±0.0008 | 0.0016±0.0002 | 0.0016±0.0002 | 0.0025±0.0010 | 0.0019±0.0008 | 0.0015±0.0007 | 0.0025±0.0005 |
| 8×10⁻² | 0.0019±0.0005 | 0.0070±0.0005 | 0.0016±0.0008 | 0.0022±0.0000 | 0.0017±0.0001 | 0.0009±0.0002 | 0.0009±0.0002 | 0.0017±0.0008 | 0.0023±0.0002 | 0.0020±0.0003 | 0.0012±0.0001 |
| 5×10⁻² | 0.0013±0.0005 | 0.0036±0.0006 | 0.0016±0.0006 | 0.0013±0.0001 | 0.0015±0.0002 | 0.0009±0.0002 | 0.0009±0.0002 | 0.0014±0.0008 | 0.0012±0.0002 | 0.0013±0.0003 | 0.0012±0.0002 |
| 3×10⁻² | 0.0005±0.0001 | 0.0014±0.0001 | 0.0007±0.0004 | 0.0006±0.0001 | 0.0006±0.0002 | 0.0007±0.0000 | 0.0007±0.0000 | 0.0006±0.0001 | 0.0004±0.0003 | 0.0009±0.0001 | 0.0007±0.0002 |
| 1×10⁻² | 0.0002±0.0001 | 0.0005±0.0000 | 0.0003±0.0000 | 0.0002±0.0000 | 0.0002±0.0000 | 0.0002±0.0000 | 0.0002±0.0000 | 0.0003±0.0000 | 0.0003±0.0001 | 0.0002±0.0000 | 0.0002±0.0001 |
| 8×10⁻³ | 0.0002±0.0001 | 0.0004±0.0001 | 0.0002±0.0001 | 0.0002±0.0000 | 0.0002±0.0000 | 0.0001±0.0000 | 0.0001±0.0000 | 0.0003±0.0000 | 0.0002±0.0000 | 0.0002±0.0000 | 0.0002±0.0001 |
| 5×10⁻³ | 0.0046±0.0063 | 0.0145±0.0083 | 0.0001±0.0000 | 0.0056±0.0045 | 0.0176±0.0004 | 0.0108±0.0023 | 0.0108±0.0023 | 0.0001±0.0000 | 0.0001±0.0023 | 0.0139±0.0109 | 0.0040±0.0045 |
| 3×10⁻³ | 0.0744±0.0031 | 0.0547±0.0049 | 0.0584±0.0035 | 0.0675±0.0070 | 0.0652±0.0027 | 0.0693±0.0032 | 0.0693±0.0032 | 0.0571±0.0490 | 0.0490±0.0050 | 0.0720±0.0118 | 0.0430±0.0022 |
| 1×10⁻³ | 0.4724±0.0245 | 0.4962±0.0489 | 0.3954±0.0311 | 0.4584±0.0498 | 0.4294±0.0182 | 0.4875±0.0095 | 0.4875±0.0300 | 0.4454±0.0413 | 0.4598±0.0300 | 0.4550±0.0257 | 0.4569±0.0212 |
| 8×10⁻⁴ | 0.5596±0.0618 | 0.5197±0.0412 | 0.5045±0.0217 | 0.5379±0.0168 | 0.5398±0.0372 | 0.5704±0.0290 | 0.5704±0.0322 | 0.4655±0.0322 | 0.5102±0.0100 | 0.5269±0.0288 | 0.5842±0.0629 |
| 5×10⁻⁴ | 0.6497±0.0230 | 0.6422±0.0430 | 0.5670±0.0304 | 0.6304±0.0142 | 0.6935±0.0272 | 0.6689±0.0122 | 0.6689±0.0122 | 0.6006±0.0077 | 0.6022±0.0140 | 0.6490±0.0217 | 0.6022±0.0103 |
| 3×10⁻⁴ | 0.7475±0.0404 | 0.7277±0.0425 | 0.7061±0.0107 | 0.7534±0.0072 | 0.7213±0.0118 | 0.7026±0.0273 | 0.7026±0.0120 | 0.6609±0.0067 | 0.7100±0.0204 | 0.7137±0.0168 | 0.7379±0.0714 |
| 1×10⁻⁴ | 0.8183±0.0169 | 0.8544±0.0166 | 0.7573±0.0481 | 0.7824±0.0234 | 0.7953±0.0529 | 0.8008±0.0233 | 0.8008±0.0233 | 0.7258±0.0020 | 0.8298±0.0434 | 0.8379±0.0426 | 0.8268±0.0378 |
| 8×10⁻⁵ | 0.8061±0.0603 | 0.8473±0.0326 | 0.7661±0.0200 | 0.8084±0.0587 | 0.8328±0.0445 | 0.8242±0.0079 | 0.8242±0.0773 | 0.7232±0.0083 | 0.8122±0.0083 | 0.8063±0.0226 | 0.7910±0.0183 |
| 5×10⁻⁵ | 0.8857±0.0193 | 0.7754±0.0065 | 0.7568±0.0355 | 0.8341±0.0581 | 0.7777±0.0325 | 0.8440±0.0466 | 0.8440±0.0466 | 0.7539±0.0457 | 0.8688±0.0086 | 0.8262±0.0200 | 0.8144±0.0535 |
| 3×10⁻⁵ | 0.8197±0.0106 | 0.8329±0.0499 | 0.7885±0.0524 | 0.8745±0.0060 | 0.8467±0.0079 | 0.8248±0.0403 | 0.8248±0.0234 | 0.7488±0.0444 | 0.8240±0.0296 | 0.8314±0.0190 | 0.8560±0.0302 |
| 1×10⁻⁵ | 0.8319±0.0335 | 0.8280±0.0646 | 0.7332±0.0098 | 0.8436±0.0053 | 0.8606±0.0325 | 0.7989±0.0125 | 0.7989±0.0207 | 0.8057±0.0343 | 0.8514±0.0213 | 0.8401±0.0153 | 0.8115±0.0153 |
| 8×10⁻⁶ | 0.7844±0.0210 | 0.8536±0.0094 | 0.7587±0.0282 | 0.7918±0.0159 | 0.8102±0.0273 | 0.8531±0.0217 | 0.8531±0.0241 | 0.8103±0.0076 | 0.8061±0.0339 | 0.8325±0.0247 | 0.8034±0.0514 |
| 5×10⁻⁶ | 0.8850±0.0634 | 0.8597±0.0217 | 0.7783±0.0245 | 0.8346±0.0281 | 0.8473±0.0400 | 0.8505±0.0519 | 0.8505±0.0526 | 0.8617±0.0261 | 0.8431±0.0268 | 0.8187±0.0314 | 0.8304±0.0387 |
| 3×10⁻⁶ | 0.8508±0.0199 | 0.8491±0.0034 | 0.7657±0.0149 | 0.8293±0.0317 | 0.8062±0.0639 | 0.8537±0.0494 | 0.8537±0.0131 | 0.8266±0.0428 | 0.8254±0.0214 | 0.8292±0.0522 | 0.8453±0.0027 |
| 1×10⁻⁶ | 0.8197±0.0568 | 0.8836±0.0216 | 0.8065±0.0506 | 0.8372±0.0384 | 0.8665±0.0218 | 0.8543±0.0447 | 0.8868±0.0218 | 0.8319±0.0460 | 0.8134±0.0318 | 0.8402±0.0557 | 0.8057±0.0303 |

Table 5: Numerical results for Gradient Flow with the Knot dataset ($d = 2$)

| LR | SW | MaxSW | DSW | MaxKSW | iMSW | viMSW | oMSW | rMSW | EBSW | RPSW | EBRPSW |
|---|---|---|---|---|---|---|---|---|---|---|---|
| 100 | 2.7116±0.6616 | 0.6408±0.0079 | 3.2280±0.5094 | 1.3326±0.0411 | 2.2337±0.8727 | 2.5000±0.8335 | 1.3862±0.2603 | 2.5671±0.8620 | 2.8073±0.1416 | 2.6577±0.0282 | 2.7618±0.1329 |
| 80 | 2.1367±0.6087 | 0.5710±0.0041 | 1.0384±0.3640 | 1.3753±0.1106 | 1.7140±0.1703 | 1.8681±0.6268 | 1.5932±0.1041 | 2.5953±0.6861 | 2.3348±0.0828 | 2.2001±0.0631 | 2.2090±0.1610 |
| 50 | 0.6877±0.0723 | 1.4925±0.8598 | 1.4048±0.1910 | 1.5581±0.0122 | 1.1232±0.1551 | 0.8977±0.2941 | 1.2540±0.0568 | 1.4821±0.1792 | 1.4024±0.1124 | 1.1371±0.0850 | 1.4613±0.0400 |
| 30 | 0.5389±0.2452 | 0.8486±0.0496 | 0.8765±0.1254 | 0.9592±0.0019 | 0.8816±0.0522 | 0.6488±0.3089 | 0.8072±0.0214 | 0.9189±0.1387 | 0.9178±0.3697 | 0.6777±0.0658 | 0.9473±0.1249 |
| 10 | 0.1721±0.0402 | 0.4116±0.0909 | 0.1965±0.0915 | 0.1744±0.0095 | 0.2179±0.1150 | 0.3502±0.0193 | 0.1614±0.0207 | 0.1752±0.0400 | 0.3453±0.0457 | 0.1866±0.1160 | 0.3806±0.0842 |
| 8 | 0.1892±0.0560 | 0.4166±0.0334 | 0.2441±0.0545 | 0.2037±0.0043 | 0.1355±0.0865 | 0.1707±0.0453 | 0.1800±0.0074 | 0.2334±0.0587 | 0.2948±0.1139 | 0.1311±0.0399 | 0.2549±0.1194 |
| 5 | 0.0907±0.0390 | 0.2481±0.0296 | 0.1453±0.0272 | 0.1270±0.0113 | 0.0916±0.0514 | 0.1663±0.0083 | 0.0956±0.0204 | 0.0990±0.0220 | 0.1724±0.0669 | 0.1171±0.0207 | 0.1779±0.0662 |
| 3 | 0.0738±0.0312 | 0.1655±0.0088 | 0.0793±0.0224 | 0.1074±0.0030 | 0.0767±0.0161 | 0.0300±0.0172 | 0.0645±0.0015 | 0.0593±0.0474 | 0.0931±0.0083 | 0.0656±0.0181 | 0.0988±0.0052 |
| 1 | 0.0167±0.0020 | 0.0788±0.0019 | 0.0264±0.0057 | 0.0489±0.0069 | 0.0205±0.0067 | 0.0167±0.0011 | 0.0191±0.0021 | 0.0148±0.0076 | 0.0234±0.0030 | 0.0230±0.0114 | 0.0208±0.0013 |
| $8 \times 10^{-1}$ | 0.0225±0.0040 | 0.0648±0.0041 | 0.0244±0.0036 | 0.0404±0.0040 | 0.0166±0.0036 | 0.0094±0.0042 | 0.0159±0.0007 | 0.0277±0.0018 | 0.0176±0.0034 | 0.0178±0.0031 | 0.0189±0.0066 |
| $5 \times 10^{-1}$ | 0.0090±0.0039 | 0.0482±0.0017 | 0.0147±0.0016 | 0.0251±0.0038 | 0.0078±0.0023 | 0.0125±0.0050 | 0.0106±0.0006 | 0.0099±0.0044 | 0.0095±0.0022 | 0.0101±0.0049 | 0.0086±0.0035 |
| $3 \times 10^{-1}$ | 0.0057±0.0014 | 0.0405±0.0016 | 0.0134±0.0078 | 0.0131±0.0008 | 0.0060±0.0010 | 0.0092±0.0013 | 0.0066±0.0003 | 0.0039±0.0021 | 0.0062±0.0012 | 0.0047±0.0011 | 0.0055±0.0015 |
| $1 \times 10^{-1}$ | 0.0018±0.0009 | 0.0138±0.0012 | 0.0019±0.0005 | 0.0031±0.0019 | 0.0078±0.0088 | 0.0021±0.0011 | 0.0031±0.0011 | 0.0013±0.0003 | 0.0018±0.0004 | 0.0020±0.0003 | 0.0019±0.0002 |
| $8 \times 10^{-2}$ | 0.0018±0.0004 | 0.0076±0.0007 | 0.0025±0.0001 | 0.0020±0.0002 | 0.0088±0.0099 | 0.0012±0.0002 | 0.0013±0.0001 | 0.0020±0.0001 | 0.0018±0.0002 | 0.0016±0.0005 | 0.0015±0.0004 |
| $5 \times 10^{-2}$ | 0.0010±0.0001 | 0.0038±0.0004 | 0.0017±0.0005 | 0.0016±0.0003 | 0.0015±0.0004 | 0.0009±0.0004 | 0.0018±0.0000 | 0.0010±0.0000 | 0.0009±0.0001 | 0.0010±0.0001 | 0.0007±0.0002 |
| $3 \times 10^{-2}$ | 0.0007±0.0004 | 0.0020±0.0003 | 0.0081±0.0104 | 0.0008±0.0000 | 0.0008±0.0001 | 0.0009±0.0002 | 0.0010±0.0000 | 0.0005±0.0001 | 0.0031±0.0030 | 0.0005±0.0001 | 0.0007±0.0002 |
| $1 \times 10^{-2}$ | 0.0002±0.0001 | 0.0004±0.0001 | 0.0003±0.0001 | 0.0002±0.0001 | 0.0001±0.0000 | 0.0071±0.0098 | 0.0005±0.0000 | 0.0002±0.0000 | 0.0002±0.0000 | 0.0002±0.0000 | 0.0003±0.0001 |
| $8 \times 10^{-3}$ | 0.0002±0.0001 | 0.0000±0.0000 | 0.0002±0.0000 | 0.0002±0.0000 | 0.0002±0.0000 | 0.0002±0.0000 | 0.0001±0.0000 | 0.0002±0.0001 | 0.0003±0.0001 | 0.0001±0.0000 | 0.0028±0.0037 |
| $5 \times 10^{-3}$ | 0.0001±0.0000 | 0.0113±0.0080 | 0.0023±0.0031 | 0.0006±0.0007 | 0.0074±0.0104 | 0.0014±0.0130 | 0.0018±0.0016 | 0.0012±0.0016 | 0.0018±0.0024 | 0.0001±0.0000 | 0.0101±0.0102 |
| $3 \times 10^{-3}$ | 0.0462±0.0025 | 0.0576±0.0029 | 0.0463±0.0037 | 0.0524±0.0029 | 0.0520±0.0014 | 0.0573±0.0045 | 0.0440±0.0063 | 0.0443±0.0036 | 0.0388±0.0011 | 0.0537±0.0011 | 0.0461±0.0020 |
| $1 \times 10^{-3}$ | 0.4562±0.0128 | 0.4431±0.0294 | 0.4015±0.0061 | 0.4037±0.0245 | 0.4144±0.0502 | 0.4502±0.0148 | 0.4378±0.0449 | 0.3841±0.0371 | 0.4561±0.0102 | 0.4135±0.0113 | 0.4053±0.0156 |
| $8 \times 10^{-4}$ | 0.5542±0.0443 | 0.5135±0.0036 | 0.4889±0.0376 | 0.5049±0.0621 | 0.4779±0.0132 | 0.4414±0.0209 | 0.5153±0.0144 | 0.4962±0.0564 | 0.5377±0.0322 | 0.5181±0.0430 | 0.4708±0.0153 |
| $5 \times 10^{-4}$ | 0.6267±0.0115 | 0.6978±0.0145 | 0.6018±0.0523 | 0.5962±0.0185 | 0.6326±0.0183 | 0.6380±0.0633 | 0.5696±0.0220 | 0.6054±0.0356 | 0.5978±0.0170 | 0.6307±0.0299 | 0.6033±0.0117 |
| $3 \times 10^{-4}$ | 0.6768±0.0396 | 0.7378±0.0884 | 0.6948±0.0818 | 0.6501±0.0650 | 0.7257±0.0189 | 0.6678±0.0099 | 0.7080±0.0066 | 0.6992±0.0195 | 0.6705±0.0717 | 0.7056±0.0626 | 0.6892±0.0496 |
| $1 \times 10^{-4}$ | 0.8010±0.0609 | 0.8032±0.0078 | 0.7756±0.0116 | 0.8050±0.0386 | 0.7569±0.0699 | 0.7831±0.0213 | 0.7689±0.0297 | 0.7545±0.0447 | 0.7353±0.0105 | 0.8001±0.0219 | 0.7609±0.0142 |
| $8 \times 10^{-5}$ | 0.7912±0.0136 | 0.8318±0.0096 | 0.7750±0.0291 | 0.7513±0.0187 | 0.7757±0.0432 | 0.7533±0.0463 | 0.7707±0.0304 | 0.7996±0.0452 | 0.7798±0.0275 | 0.8173±0.0162 | 0.7751±0.0186 |
| $5 \times 10^{-5}$ | 0.7931±0.0177 | 0.8609±0.0384 | 0.7861±0.0270 | 0.7883±0.0083 | 0.7286±0.0474 | 0.7751±0.0454 | 0.7664±0.0712 | 0.7524±0.0513 | 0.7911±0.0444 | 0.7857±0.0189 | 0.7777±0.0174 |
| $3 \times 10^{-5}$ | 0.7866±0.0157 | 0.8463±0.0257 | 0.8185±0.0219 | 0.8197±0.0474 | 0.8045±0.0219 | 0.7678±0.0227 | 0.8187±0.0423 | 0.7756±0.0405 | 0.8055±0.0188 | 0.7912±0.0439 | 0.7578±0.0359 |
| $1 \times 10^{-5}$ | 0.7728±0.0195 | 0.8348±0.0121 | 0.8120±0.0219 | 0.7983±0.0566 | 0.7745±0.0089 | 0.7827±0.0296 | 0.8054±0.0260 | 0.8240±0.0307 | 0.8172±0.0239 | 0.7912±0.0288 | 0.7982±0.0576 |
| $8 \times 10^{-6}$ | 0.7996±0.0623 | 0.8318±0.0384 | 0.7549±0.0266 | 0.8504±0.0217 | 0.8288±0.0243 | 0.8287±0.0260 | 0.8424±0.0903 | 0.8012±0.0424 | 0.7854±0.0054 | 0.8035±0.0243 | 0.7890±0.0228 |
| $5 \times 10^{-6}$ | 0.8437±0.0315 | 0.8609±0.0384 | 0.8357±0.0442 | 0.8353±0.0145 | 0.7918±0.0215 | 0.7905±0.0248 | 0.7750±0.0088 | 0.8025±0.0390 | 0.8363±0.0332 | 0.7982±0.0254 | 0.8205±0.0078 |
| $3 \times 10^{-6}$ | 0.7935±0.0268 | 0.8463±0.0257 | 0.8084±0.0411 | 0.8457±0.0174 | 0.7783±0.0195 | 0.7867±0.0324 | 0.8515±0.0514 | 0.8093±0.0442 | 0.8079±0.0208 | 0.8266±0.0051 | 0.8228±0.0168 |
| $1 \times 10^{-6}$ | 0.8332±0.0122 | 0.8348±0.0121 | 0.8042±0.0282 | 0.8091±0.0190 | 0.7770±0.0248 | 0.7748±0.0164 | 0.7932±0.0170 | 0.8373±0.0212 | 0.8203±0.0194 | 0.8121±0.0115 | 0.8057±0.0246 |

Table 6: Numerical results for Gradient Flow with the Swiss dataset ($d = 2$)

| LR | SW | MaxSW | DSW | MaxKSW | iMSW | viMSW | oMSW | rMSW | EBSW | RPSW | EBRPSW |
|---|---|---|---|---|---|---|---|---|---|---|---|
| 100 | 1.8754±1.2183 | 3.6719±1.6669 | 2.2167±1.3910 | 1.1766±0.0901 | 2.0996±0.5058 | 2.3956±0.7993 | 3.0252±0.1706 | 2.6436±1.0882 | 3.0252±0.1706 | 2.6426±0.0669 | 2.8145±0.1491 |
| 80 | 1.6682±0.9727 | 1.8203±0.9088 | 2.2781±0.4102 | 1.1566±0.0318 | 2.0056±0.7764 | 1.8520±0.5824 | 2.2930±0.0903 | 1.8364±1.0702 | 2.2930±0.0903 | 2.0552±0.1070 | 2.3738±0.0394 |
| 50 | 0.7122±0.1076 | 1.5041±0.2527 | 0.7793±0.4214 | 0.9272±0.0260 | 1.1239±0.3555 | 1.3310±0.4241 | 1.7172±0.1023 | 1.4756±0.3155 | 1.7172±0.1023 | 1.1673±0.1447 | 1.4905±0.0416 |
| 30 | 0.6045±0.3404 | 1.0704±0.0587 | 0.6909±0.3766 | 0.7364±0.0218 | 0.7664±0.0791 | 0.4753±0.1999 | 1.0015±0.0625 | 0.8430±0.2975 | 1.0015±0.0625 | 0.6435±0.3303 | 0.9820±0.1278 |
| 10 | 0.1544±0.0464 | 0.4782±0.0479 | 0.2348±0.0887 | 0.1933±0.0312 | 0.1795±0.0326 | 0.3404±0.0127 | 0.2363±0.0245 | 0.2242±0.0245 | 0.2363±0.0112 | 0.2729±0.0176 | 0.3536±0.0355 |
| 8 | 0.2014±0.0251 | 0.3462±0.0410 | 0.2254±0.0901 | 0.1854±0.0010 | 0.1386±0.0738 | 0.1293±0.0563 | 0.3805±0.0112 | 0.1638±0.0318 | 0.3805±0.0013 | 0.2027±0.0718 | 0.0854±0.0351 |
| 5 | 0.0995±0.0458 | 0.2004±0.0136 | 0.0772±0.0469 | 0.0998±0.0078 | 0.1063±0.0017 | 0.1894±0.0047 | 0.0780±0.0013 | 0.2197±0.0233 | 0.0780±0.0058 | 0.1016±0.0034 | 0.1692±0.0604 |
| 3 | 0.0868±0.0193 | 0.1177±0.0058 | 0.0640±0.0260 | 0.0613±0.0033 | 0.0885±0.0218 | 0.4035±0.0251 | 0.1012±0.0058 | 0.4207±0.0243 | 0.1012±0.0014 | 0.0676±0.0091 | 0.0958±0.0013 |
| 1 | 0.0230±0.0086 | 0.0436±0.0010 | 0.0245±0.0084 | 0.0230±0.0008 | 0.0183±0.0091 | 0.7765±0.0445 | 0.0187±0.0014 | 0.8306±0.0391 | 0.0171±0.0050 | 0.0250±0.0009 | 0.0209±0.0019 |
| $8 \times 10^{-1}$ | 0.0246±0.0058 | 0.0342±0.0012 | 0.0168±0.0050 | 0.0189±0.0005 | 0.0171±0.0011 | 0.8516±0.0207 | 0.0139±0.0012 | 0.8743±0.0473 | 0.0177±0.0177 | 0.0217±0.0016 | 0.0151±0.0036 |
| $5 \times 10^{-1}$ | 0.0091±0.0027 | 0.0233±0.0006 | 0.0103±0.0042 | 0.0146±0.0006 | 0.0096±0.0027 | 0.9355±0.0290 | 0.0090±0.0007 | 0.9269±0.0258 | 0.0092±0.0092 | 0.0142±0.0033 | 0.0099±0.0037 |
| $3 \times 10^{-1}$ | 0.0070±0.0006 | 0.0158±0.0001 | 0.0064±0.0027 | 0.0119±0.0005 | 0.0078±0.0019 | 0.9861±0.0259 | 0.0062±0.0002 | 1.0561±0.0192 | 0.0041±0.0041 | 0.0059±0.0002 | 0.0017±0.0007 |
| $1 \times 10^{-1}$ | 0.0020±0.0002 | 0.0097±0.0002 | 0.0018±0.0010 | 0.0059±0.0009 | 0.0025±0.0007 | 1.0240±0.0293 | 0.0001±0.0001 | 1.0638±0.0350 | 0.0015±0.0015 | 0.0027±0.0008 | 0.0021±0.0007 |
| $8 \times 10^{-2}$ | 0.0019±0.0003 | 0.0076±0.0002 | 0.0021±0.0003 | 0.0040±0.0002 | 0.0020±0.0001 | 1.0657±0.0399 | 0.0017±0.0001 | 1.0979±0.0350 | 0.0012±0.0018 | 0.0019±0.0003 | 0.0013±0.0001 |
| $5 \times 10^{-2}$ | 0.0010±0.0003 | 0.0061±0.0000 | 0.0006±0.0002 | 0.0031±0.0004 | 0.0013±0.0002 | 1.0926±0.0206 | 0.0010±0.0001 | 1.1166±0.0107 | 0.0012±0.0320 | 0.0014±0.0005 | 0.0006±0.0004 |
| $3 \times 10^{-2}$ | 0.0008±0.0002 | 0.0031±0.0000 | 0.0010±0.0001 | 0.0018±0.0002 | 0.0008±0.0002 | 1.1155±0.0033 | 0.0006±0.0001 | 1.1021±0.0320 | 0.0009±0.0313 | 0.0007±0.0002 | 0.0007±0.0002 |
| $1 \times 10^{-2}$ | 0.0002±0.0001 | 0.1316±0.0190 | 0.0002±0.0001 | 0.0384±0.0497 | 0.0003±0.0000 | 0.0002±0.0000 | 0.0002±0.0000 | 0.0003±0.0313 | 0.0001±0.0000 | 0.0002±0.0000 | 0.0009±0.0001 |
| $8 \times 10^{-3}$ | 0.0007±0.0009 | 0.1939±0.0405 | 0.0001±0.0149 | 0.1640±0.0036 | 0.0002±0.0215 | 0.0002±0.0000 | 0.0002±0.0000 | 0.0002±0.0211 | 0.1260±0.0075 | 0.0014±0.0009 | 0.1106±0.0398 |
| $5 \times 10^{-3}$ | 0.1965±0.0290 | 0.3497±0.0149 | 0.2026±0.0270 | 0.3047±0.0215 | 0.2563±0.0217 | 0.1894±0.4035 | 0.2271±0.0173 | 0.2197±0.0233 | 0.3223±0.0112 | 0.1987±0.0204 | 0.2964±0.0164 |
| $3 \times 10^{-3}$ | 0.4310±0.0204 | 0.4287±0.0222 | 0.4433±0.0227 | 0.4484±0.0108 | 0.4080±0.0145 | 0.4035±0.0251 | 0.4203±0.0266 | 0.4207±0.0452 | 0.4304±0.0058 | 0.4490±0.0226 | 0.4122±0.0402 |
| $1 \times 10^{-3}$ | 0.8561±0.0290 | 0.8367±0.0442 | 0.7788±0.0083 | 0.8168±0.0181 | 0.7574±0.0171 | 0.7765±0.0445 | 0.8061±0.0445 | 0.8306±0.0391 | 0.7096±0.0249 | 0.8049±0.0369 | 0.7294±0.0128 |
| $8 \times 10^{-4}$ | 0.8471±0.0388 | 0.8914±0.0049 | 0.8349±0.0178 | 0.8478±0.0113 | 0.8654±0.0206 | 0.8516±0.0207 | 0.8791±0.0156 | 0.8743±0.0473 | 0.8134±0.0245 | 0.8864±0.0115 | 0.7763±0.0230 |
| $5 \times 10^{-4}$ | 0.9638±0.0207 | 0.9201±0.0121 | 0.9686±0.0407 | 0.9992±0.0064 | 0.9837±0.0163 | 0.9355±0.0290 | 0.9339±0.0409 | 0.9269±0.0258 | 0.9093±0.0198 | 0.9818±0.0177 | 0.9462±0.0092 |
| $3 \times 10^{-4}$ | 1.0198±0.0345 | 0.9876±0.0126 | 0.9971±0.0412 | 0.9834±0.0111 | 1.0509±0.0250 | 0.9861±0.0250 | 0.9586±0.0182 | 1.0561±0.0192 | 0.9803±0.0350 | 1.0486±0.0266 | 0.9542±0.0045 |
| $1 \times 10^{-4}$ | 1.0491±0.0545 | 1.0148±0.0181 | 1.0628±0.0181 | 1.0362±0.0423 | 1.0967±0.0304 | 1.0240±0.0192 | 1.0540±0.0293 | 1.0638±0.0370 | 1.0043±0.0284 | 1.0805±0.0667 | 1.0699±0.0781 |
| $8 \times 10^{-5}$ | 1.1368±0.0152 | 1.0651±0.0629 | 1.0756±0.0461 | 1.0883±0.0461 | 1.0514±0.0437 | 1.0657±0.0400 | 1.1419±0.0481 | 1.0979±0.0481 | 1.1419±0.0350 | 1.1368±0.0152 | 1.0124±0.0534 |
| $5 \times 10^{-5}$ | 1.1340±0.0280 | 1.0858±0.0165 | 0.9856±0.0501 | 1.1182±0.0655 | 1.0685±0.0534 | 1.0926±0.0206 | 1.0581±0.0206 | 1.1166±0.0107 | 1.0581±0.0062 | 1.1340±0.0280 | 1.1027±0.0240 |
| $3 \times 10^{-5}$ | 1.1742±0.0390 | 1.0918±0.0456 | 1.1101±0.0523 | 1.1371±0.0335 | 1.1516±0.0201 | 1.1155±0.0033 | 1.0978±0.0461 | 1.1021±0.0320 | 1.0978±0.0461 | 1.1742±0.0390 | 1.1002±0.0187 |
| $1 \times 10^{-5}$ | 1.1502±0.0119 | 1.1172±0.0257 | 1.1101±0.0553 | 1.0822±0.0282 | 1.1097±0.0482 | 1.1363±0.0244 | 1.1149±0.0311 | 1.0607±0.0759 | 1.1149±0.0311 | 1.1284±0.0389 | 1.1040±0.0386 |
| $8 \times 10^{-6}$ | 1.1234±0.0241 | 1.1757±0.0273 | 1.1211±0.0590 | 1.1268±0.0387 | 1.1028±0.0081 | 1.1288±0.0206 | 1.1419±0.0481 | 1.1154±0.0252 | 1.1419±0.0481 | 1.1368±0.0152 | 1.1397±0.0270 |
| $5 \times 10^{-6}$ | 1.0738±0.0273 | 1.0898±0.0584 | 1.1256±0.0493 | 1.1105±0.0139 | 1.1040±0.0153 | 1.1205±0.0187 | 1.0581±0.0194 | 1.1064±0.0194 | 1.0581±0.0062 | 1.1340±0.0280 | 1.1027±0.0240 |
| $3 \times 10^{-6}$ | 1.1109±0.0355 | 1.1249±0.0357 | 1.0902±0.0320 | 1.1371±0.0335 | 1.1516±0.0201 | 1.1130±0.0274 | 1.0978±0.0528 | 1.0785±0.0528 | 1.0978±0.0461 | 1.1742±0.0390 | 1.1002±0.0187 |
| $1 \times 10^{-6}$ | 1.1502±0.0119 | 1.1172±0.0257 | 1.1101±0.0553 | 1.0822±0.0282 | 1.1097±0.0482 | 1.1694±0.0198 | 1.1149±0.0311 | 1.0951±0.0311 | 1.1149±0.0311 | 1.1284±0.0389 | 1.1040±0.0386 |

Table 7: Numerical results for Gradient Flow with the 8 Gaussians dataset ($d = 2$)

| LR | SW | MaxSW | DSW | MaxKSW | iMSW | viMSW | oMSW | rMSW | EBSW | RPSW | EBRPSW |
|---|---|---|---|---|---|---|---|---|---|---|---|
| 100 | 0.7900±0.0704 | 4.4368±0.0966 | 3.0902±0.1026 | 2.9073±0.3947 | 2.2985±0.1220 | 1.2282±0.0578 | 0.8969±0.0500 | 1.2429±0.1167 | 10.5129±0.2240 | 1.2812±0.0165 | 11.0450±0.8609 |
| 80 | 0.6047±0.0850 | 3.4799±0.0544 | 2.4792±0.2602 | 2.2292±0.2711 | 1.7091±0.0921 | 0.9662±0.0284 | 0.6897±0.0442 | 0.9003±0.0392 | 8.5668±0.3944 | 1.0411±0.0586 | 7.8665±0.8317 |
| 50 | 0.3459±0.0234 | 2.7423±0.5278 | 1.0806±0.3335 | 1.7169±0.1028 | 1.0745±0.0742 | 0.5512±0.0653 | 0.3995±0.0698 | 0.5736±0.0181 | 4.3396±0.3767 | 0.6150±0.0368 | 4.0199±0.0448 |
| 30 | 0.2229±0.0178 | 1.8470±0.1703 | 0.8920±0.0499 | 1.5803±0.1051 | 0.7473±0.0193 | 0.3307±0.0363 | 0.2335±0.0071 | 0.3464±0.0307 | 2.3719±0.0339 | 0.3622±0.0363 | 2.7110±0.3379 |
| 10 | 0.0819±0.0056 | 1.2519±0.0267 | 0.1853±0.0391 | 0.9043±0.0135 | 0.5119±0.0033 | 0.1162±0.0097 | 0.0825±0.0035 | 0.1102±0.0206 | 1.0896±0.0531 | 0.1307±0.0169 | 1.0587±0.0079 |
| 8 | 0.0592±0.0014 | 1.0781±0.0115 | 0.1659±0.0566 | 0.7843±0.0256 | 0.4259±0.0210 | 0.0821±0.0011 | 0.0590±0.0082 | 0.0835±0.0041 | 1.0515±0.0428 | 0.1071±0.0068 | 0.9913±0.0529 |
| 5 | 0.0402±0.0026 | 0.7808±0.0114 | 0.0734±0.0121 | 0.5545±0.0083 | 0.3104±0.0177 | 0.0649±0.0070 | 0.0410±0.0007 | 0.0560±0.0027 | 0.0412±0.0027 | 0.0693±0.0138 | 0.0429±0.0037 |
| 3 | 0.0234±0.0026 | 0.5466±0.0103 | 0.0374±0.0089 | 0.4019±0.0028 | 0.2041±0.0022 | 0.0386±0.0027 | 0.0265±0.0017 | 0.0348±0.0009 | 0.0258±0.0015 | 0.0430±0.0029 | 0.0217±0.0019 |
| 1 | 0.0073±0.0002 | 0.2788±0.0092 | 0.0125±0.0025 | 0.1792±0.0007 | 0.0912±0.0016 | 0.0113±0.0013 | 0.0070±0.0009 | 0.0115±0.0005 | 0.0081±0.0008 | 0.0144±0.0009 | 0.0089±0.0009 |
| $8 \times 10^{-1}$ | 0.0067±0.0007 | 0.2448±0.0066 | 0.0099±0.0006 | 0.1514±0.0017 | 0.0805±0.0013 | 0.0082±0.0013 | 0.0063±0.0003 | 0.0094±0.0010 | 0.0063±0.0006 | 0.0117±0.0003 | 0.0062±0.0004 |
| $5 \times 10^{-1}$ | 0.0038±0.0004 | 0.1762±0.0045 | 0.0064±0.0006 | 0.1104±0.0031 | 0.0573±0.0011 | 0.0057±0.0011 | 0.0036±0.0003 | 0.0057±0.0003 | 0.0042±0.0002 | 0.0074±0.0002 | 0.0043±0.0003 |
| $3 \times 10^{-1}$ | 0.0026±0.0002 | 0.1282±0.0012 | 0.0036±0.0001 | 0.0788±0.0018 | 0.0381±0.0004 | 0.0034±0.0004 | 0.0023±0.0001 | 0.0035±0.0001 | 0.0022±0.0003 | 0.0041±0.0003 | 0.0023±0.0001 |
| $1 \times 10^{-1}$ | 0.0008±0.0000 | 0.0597±0.0009 | 0.0015±0.0000 | 0.0342±0.0005 | 0.0107±0.0002 | 0.0012±0.0001 | 0.0008±0.0000 | 0.0012±0.0001 | 0.0008±0.0001 | 0.0015±0.0001 | 0.0008±0.0000 |
| $8 \times 10^{-2}$ | 0.0006±0.0000 | 0.0499±0.0003 | 0.0010±0.0000 | 0.0282±0.0004 | 0.0086±0.0000 | 0.0010±0.0000 | 0.0006±0.0000 | 0.0009±0.0000 | 0.0006±0.0000 | 0.0012±0.0001 | 0.0006±0.0000 |
| $5 \times 10^{-2}$ | 0.0004±0.0000 | 0.0342±0.0003 | 0.0007±0.0000 | 0.0171±0.0003 | 0.0054±0.0000 | 0.0006±0.0000 | 0.0003±0.0000 | 0.0006±0.0000 | 0.0004±0.0000 | 0.0008±0.0000 | 0.0004±0.0000 |
| $3 \times 10^{-2}$ | 0.0002±0.0000 | 0.0272±0.0072 | 0.0004±0.0000 | 0.0080±0.0000 | 0.0031±0.0000 | 0.0004±0.0000 | 0.0002±0.0000 | 0.0003±0.0000 | 0.0003±0.0000 | 0.0005±0.0001 | 0.0002±0.0000 |
| $1 \times 10^{-2}$ | 0.1699±0.0262 | 0.0530±0.0060 | 0.0395±0.0020 | 0.0639±0.0039 | 0.1088±0.0063 | 0.1371±0.0141 | 0.1715±0.0089 | 0.1507±0.0100 | 0.0535±0.0008 | 0.0056±0.0077 | 0.0641±0.0027 |
| $8 \times 10^{-3}$ | 0.2678±0.0292 | 0.0706±0.0028 | 0.0564±0.0037 | 0.0922±0.0046 | 0.1784±0.0108 | 0.2101±0.0076 | 0.2381±0.0090 | 0.2667±0.0288 | 0.0968±0.0027 | 0.0010±0.0012 | 0.0925±0.0011 |
| $5 \times 10^{-3}$ | 0.4264±0.0227 | 0.1373±0.0006 | 0.1307±0.0060 | 0.2096±0.0176 | 0.3224±0.0325 | 0.3775±0.0155 | 0.4109±0.0132 | 0.3907±0.0216 | 0.1715±0.0034 | 0.0770±0.0092 | 0.1853±0.0080 |
| $3 \times 10^{-3}$ | 0.5863±0.0435 | 0.2441±0.0004 | 0.2907±0.0178 | 0.3481±0.0071 | 0.4836±0.0280 | 0.5586±0.0159 | 0.5553±0.0304 | 0.5572±0.0043 | 0.2927±0.0148 | 0.2330±0.0223 | 0.3111±0.0292 |
| $1 \times 10^{-3}$ | 0.7987±0.0118 | 0.7585±0.0052 | 0.7517±0.0287 | 0.7607±0.0420 | 0.7778±0.0164 | 0.7404±0.0106 | 0.7858±0.0109 | 0.7806±0.0178 | 0.7751±0.0066 | 0.7427±0.0242 | 0.7704±0.0552 |
| $8 \times 10^{-4}$ | 0.7785±0.0360 | 0.7701±0.0215 | 0.7445±0.0253 | 0.7571±0.0562 | 0.7751±0.0262 | 0.7760±0.0339 | 0.7826±0.0458 | 0.7852±0.0142 | 0.7615±0.0246 | 0.7313±0.0053 | 0.7520±0.0261 |
| $5 \times 10^{-4}$ | 0.7931±0.0415 | 0.7580±0.0470 | 0.7146±0.0470 | 0.7738±0.0153 | 0.7709±0.0115 | 0.7612±0.0216 | 0.7837±0.0457 | 0.8190±0.0229 | 0.7759±0.0414 | 0.7704±0.0147 | 0.7323±0.0210 |
| $3 \times 10^{-4}$ | 0.7726±0.0087 | 0.7806±0.0640 | 0.7752±0.0142 | 0.7582±0.0184 | 0.7892±0.0316 | 0.7936±0.0294 | 0.7779±0.0213 | 0.7843±0.0390 | 0.7901±0.0189 | 0.7528±0.0385 | 0.7730±0.0263 |
| $1 \times 10^{-4}$ | 0.7724±0.0251 | 0.7373±0.0580 | 0.7380±0.0290 | 0.7984±0.0495 | 0.7581±0.0325 | 0.8025±0.0279 | 0.7750±0.0102 | 0.7991±0.0153 | 0.7713±0.0014 | 0.7643±0.0285 | 0.7380±0.0197 |
| $8 \times 10^{-5}$ | 0.7802±0.0147 | 0.7643±0.0223 | 0.7921±0.0231 | 0.7104±0.0534 | 0.7957±0.0327 | 0.7725±0.0192 | 0.7747±0.0062 | 0.7413±0.0228 | 0.7748±0.0297 | 0.7585±0.0187 | 0.7867±0.0122 |
| $5 \times 10^{-5}$ | 0.7682±0.0403 | 0.7974±0.0291 | 0.8113±0.0685 | 0.7807±0.0192 | 0.7793±0.0381 | 0.8181±0.0332 | 0.8045±0.0346 | 0.7838±0.0321 | 0.7908±0.0161 | 0.7631±0.0420 | 0.7142±0.0102 |
| $3 \times 10^{-5}$ | 0.7980±0.0513 | 0.7476±0.0426 | 0.7775±0.0282 | 0.7561±0.0425 | 0.7576±0.0305 | 0.7579±0.0212 | 0.7556±0.0193 | 0.7656±0.0462 | 0.7890±0.0293 | 0.8213±0.0123 | 0.7827±0.0255 |
| $1 \times 10^{-5}$ | 0.7860±0.0132 | 0.7859±0.0271 | 0.7955±0.0397 | 0.7627±0.0285 | 0.7747±0.0025 | 0.7576±0.0395 | 0.7869±0.0353 | 0.7682±0.0255 | 0.7753±0.0191 | 0.7573±0.0311 | 0.7831±0.0199 |
| $8 \times 10^{-6}$ | 0.7860±0.0132 | 0.7859±0.0271 | 0.7955±0.0397 | 0.7627±0.0285 | 0.7747±0.0025 | 0.7576±0.0395 | 0.7869±0.0353 | 0.7682±0.0255 | 0.7753±0.0191 | 0.7573±0.0311 | 0.7831±0.0199 |
| $5 \times 10^{-6}$ | 0.7860±0.0132 | 0.7859±0.0271 | 0.7955±0.0397 | 0.7561±0.0425 | 0.7576±0.0305 | 0.7579±0.0212 | 0.7556±0.0193 | 0.7682±0.0255 | 0.7753±0.0191 | 0.7573±0.0311 | 0.7831±0.0199 |
| $3 \times 10^{-6}$ | 0.7980±0.0513 | 0.7476±0.0426 | 0.7775±0.0282 | 0.7561±0.0425 | 0.7576±0.0305 | 0.7579±0.0212 | 0.7556±0.0193 | 0.7656±0.0462 | 0.7890±0.0293 | 0.8213±0.0123 | 0.7827±0.0255 |
| $1 \times 10^{-6}$ | 0.7860±0.0132 | 0.7859±0.0271 | 0.7955±0.0397 | 0.7627±0.0285 | 0.7747±0.0025 | 0.7576±0.0395 | 0.7869±0.0353 | 0.7682±0.0255 | 0.7753±0.0191 | 0.7573±0.0311 | 0.7831±0.0199 |

Table 8: Numerical results for Gradient Flow with the Knot dataset ($d = 50$)

| LR | SW | MaxSW | DSW | MaxKSW | iMSW | viMSW | oMSW | rMSW | EBSW | RPSW | EBRPSW |
|---|---|---|---|---|---|---|---|---|---|---|---|
| 100 | 0.7547±0.0159 | 5.2763±0.0581 | 2.7308±0.2356 | 3.2047±0.1553 | 2.2315±0.0152 | 1.1926±0.1009 | 0.7655±0.0725 | 1.0225±0.0259 | 8.2665±0.6763 | 1.2582±0.1625 | 8.2852±0.4012 |
| 80 | 0.7061±0.0588 | 4.4078±0.8741 | 2.1265±0.5707 | 2.6758±0.2432 | 1.5470±0.0262 | 0.9951±0.0213 | 0.6729±0.050 | 0.8837±0.0816 | 6.7290±0.0731 | 1.1278±0.1272 | 6.6422±0.3327 |
| 50 | 0.4126±0.0458 | 3.1516±0.4662 | 1.5598±0.1044 | 2.6188±0.1909 | 1.0823±0.0751 | 0.6173±0.0160 | 0.4052±0.0346 | 0.5515±0.0068 | 4.5307±0.1181 | 0.6737±0.0310 | 4.1734±0.1881 |
| 30 | 0.2164±0.0158 | 2.5630±0.0389 | 0.8967±0.0765 | 2.0890±0.0750 | 1.2381±0.0289 | 0.3187±0.0404 | 0.2560±0.0072 | 0.3502±0.0163 | 2.8793±0.0917 | 0.4439±0.0344 | 2.8941±0.1512 |
| 10 | 0.0798±0.0005 | 1.4811±0.0137 | 0.1845±0.0386 | 1.1085±0.0246 | 0.6727±0.0053 | 0.1076±0.0092 | 0.0733±0.0010 | 0.1136±0.0098 | 1.6626±0.0213 | 0.1442±0.0078 | 1.5850±0.0564 |
| 8 | 0.0701±0.0045 | 1.2975±0.0386 | 0.1219±0.0369 | 0.9595±0.0109 | 0.5404±0.0106 | 0.0885±0.0099 | 0.0584±0.0057 | 0.0884±0.0097 | 1.3687±0.0730 | 0.1110±0.0185 | 1.4399±0.0461 |
| 5 | 0.0387±0.0020 | 1.0414±0.0267 | 0.0892±0.0231 | 0.7089±0.0243 | 0.3520±0.0050 | 0.0595±0.0041 | 0.0373±0.0030 | 0.0508±0.0028 | 0.0395±0.0016 | 0.0723±0.0041 | 0.0426±0.0054 |
| 3 | 0.0253±0.0009 | 0.7394±0.0176 | 0.0374±0.0112 | 0.4655±0.0047 | 0.1601±0.0228 | 0.0301±0.0002 | 0.0235±0.0011 | 0.0314±0.0018 | 0.0226±0.0022 | 0.0438±0.0020 | 0.0239±0.0020 |
| 1 | 0.0079±0.0003 | 0.2967±0.0070 | 0.0146±0.0022 | 0.1618±0.0059 | 0.0558±0.0016 | 0.0112±0.0014 | 0.0076±0.0004 | 0.0120±0.0013 | 0.0078±0.0009 | 0.0150±0.0005 | 0.0089±0.0003 |
| $8 \times 10^{-1}$ | 0.0065±0.0005 | 0.2347±0.0065 | 0.0106±0.0018 | 0.0932±0.0034 | 0.0469±0.0011 | 0.0089±0.0002 | 0.0059±0.0003 | 0.0091±0.0008 | 0.0068±0.0004 | 0.0126±0.0006 | 0.0059±0.0003 |
| $5 \times 10^{-1}$ | 0.0039±0.0002 | 0.1408±0.0040 | 0.0072±0.0003 | 0.0627±0.0017 | 0.0350±0.0005 | 0.0053±0.0004 | 0.0043±0.0003 | 0.0056±0.0006 | 0.0038±0.0002 | 0.0081±0.0007 | 0.0039±0.0003 |
| $3 \times 10^{-1}$ | 0.0025±0.0001 | 0.0740±0.0014 | 0.0038±0.0004 | 0.0448±0.0006 | 0.0256±0.0004 | 0.0033±0.0003 | 0.0024±0.0003 | 0.0035±0.0002 | 0.0024±0.0004 | 0.0052±0.0002 | 0.0024±0.0002 |
| $1 \times 10^{-1}$ | 0.0009±0.0001 | 0.0338±0.0004 | 0.0012±0.0001 | 0.0250±0.0009 | 0.0123±0.0001 | 0.0013±0.0001 | 0.0008±0.0001 | 0.0011±0.0002 | 0.0008±0.0001 | 0.0017±0.0001 | 0.0009±0.0000 |
| $8 \times 10^{-2}$ | 0.0006±0.0001 | 0.0303±0.0003 | 0.0009±0.0002 | 0.0216±0.0007 | 0.0102±0.0004 | 0.0008±0.0001 | 0.0006±0.0000 | 0.0010±0.0000 | 0.0007±0.0000 | 0.0013±0.0001 | 0.0006±0.0000 |
| $5 \times 10^{-2}$ | 0.0004±0.0000 | 0.0246±0.0009 | 0.0007±0.0001 | 0.0154±0.0001 | 0.0065±0.0001 | 0.0006±0.0001 | 0.0004±0.0000 | 0.0006±0.0000 | 0.0004±0.0000 | 0.0009±0.0000 | 0.0004±0.0000 |
| $3 \times 10^{-2}$ | 0.1207±0.0565 | 0.0171±0.0063 | 0.0004±0.0001 | 0.1128±0.0265 | 0.1479±0.0204 | 0.0808±0.0513 | 0.0953±0.0630 | 0.1041±0.0235 | 0.0002±0.0000 | 0.0005±0.0000 | 0.0002±0.0000 |
| $1 \times 10^{-2}$ | 0.5471±0.0314 | 0.3074±0.0314 | 0.2104±0.0210 | 0.3963±0.0099 | 0.4644±0.0191 | 0.5370±0.0153 | 0.5546±0.0361 | 0.5783±0.0261 | 0.3110±0.0059 | 0.1862±0.0244 | 0.3056±0.0195 |
| $8 \times 10^{-3}$ | 0.6607±0.0252 | 0.3471±0.0367 | 0.2931±0.0161 | 0.4406±0.0058 | 0.5575±0.0223 | 0.6040±0.0050 | 0.6462±0.0206 | 0.6438±0.0218 | 0.3705±0.0090 | 0.2789±0.0171 | 0.3714±0.0098 |
| $5 \times 10^{-3}$ | 0.8309±0.0754 | 0.4564±0.0277 | 0.4910±0.0170 | 0.5249±0.0144 | 0.6980±0.0091 | 0.7606±0.0329 | 0.7976±0.0299 | 0.8461±0.0325 | 0.5150±0.0186 | 0.4749±0.0196 | 0.4967±0.0190 |
| $3 \times 10^{-3}$ | 0.9346±0.0160 | 0.5658±0.0233 | 0.6897±0.0284 | 0.6755±0.0063 | 0.7964±0.0128 | 0.8806±0.0164 | 0.9133±0.0362 | 0.9116±0.0411 | 0.6939±0.0332 | 0.6552±0.0181 | 0.7147±0.0048 |
| $1 \times 10^{-3}$ | 1.0732±0.0251 | 0.8076±0.0251 | 0.9769±0.0476 | 0.9318±0.0156 | 1.0055±0.0251 | 1.0421±0.0599 | 1.0351±0.0351 | 1.0009±0.0105 | 0.9582±0.0105 | 0.9311±0.0464 | 0.9782±0.0298 |
| $8 \times 10^{-4}$ | 1.0660±0.0099 | 0.8543±0.0055 | 0.9574±0.0055 | 0.9526±0.0065 | 1.0176±0.0510 | 1.0811±0.0094 | 1.0274±0.0524 | 1.0162±0.0435 | 1.0022±0.0106 | 1.0049±0.0174 | 0.9958±0.0305 |
| $5 \times 10^{-4}$ | 1.0721±0.0250 | 0.9709±0.0153 | 0.9846±0.0195 | 0.9930±0.0107 | 1.0416±0.0409 | 1.0991±0.0175 | 1.1032±0.0322 | 1.0534±0.0326 | 1.0223±0.0399 | 1.0613±0.0227 | 1.0550±0.0291 |
| $3 \times 10^{-4}$ | 1.0848±0.0382 | 1.0400±0.0263 | 1.0311±0.0734 | 1.0448±0.0222 | 1.0676±0.0230 | 1.0752±0.0319 | 1.0726±0.0208 | 1.1065±0.0231 | 1.0811±0.0291 | 1.0522±0.0462 | 1.0969±0.0233 |
| $1 \times 10^{-4}$ | 1.1030±0.0147 | 1.0385±0.0311 | 1.1083±0.0502 | 1.1083±0.0311 | 1.1079±0.0175 | 1.0896±0.0660 | 1.1286±0.0108 | 1.1286±0.0108 | 1.1013±0.0297 | 1.0749±0.0316 | 1.0884±0.0245 |
| $8 \times 10^{-5}$ | 1.0942±0.0655 | 1.0906±0.0433 | 1.1044±0.0174 | 1.1044±0.0433 | 1.1241±0.0615 | 1.1241±0.0374 | 1.0936±0.0374 | 1.0882±0.0441 | 1.0679±0.0441 | 1.1070±0.0730 | 1.0661±0.0650 |
| $5 \times 10^{-5}$ | 1.1349±0.0215 | 1.0513±0.0381 | 1.0550±0.0381 | 1.1250±0.0374 | 1.0875±0.0187 | 1.0875±0.0187 | 1.0709±0.0248 | 1.1488±0.0098 | 1.0924±0.0221 | 1.0870±0.0166 | 1.0661±0.0188 |
| $3 \times 10^{-5}$ | 1.0990±0.0313 | 1.0985±0.0104 | 1.0675±0.0104 | 1.1064±0.0454 | 1.1231±0.0358 | 1.1231±0.0358 | 1.1014±0.0307 | 1.1131±0.0082 | 1.1001±0.0724 | 1.0806±0.0281 | 1.1312±0.0125 |
| $1 \times 10^{-5}$ | 1.0485±0.0162 | 1.1895±0.0512 | 1.0654±0.0337 | 1.1409±0.0482 | 1.1434±0.0657 | 1.1434±0.0207 | 1.1062±0.0207 | 1.1249±0.0194 | 1.1434±0.0072 | 1.1078±0.0490 | 1.1283±0.0508 |
| $8 \times 10^{-6}$ | 1.1477±0.0352 | 1.1061±0.0364 | 1.0993±0.0411 | 1.1129±0.0411 | 1.1289±0.0148 | 1.0819±0.0535 | 1.0914±0.0029 | 1.0946±0.0192 | 1.1224±0.0029 | 1.0909±0.0091 | 1.0903±0.0345 |
| $5 \times 10^{-6}$ | 1.1039±0.0245 | 1.0986±0.0137 | 1.1386±0.0528 | 1.1225±0.0556 | 1.0472±0.0103 | 1.0801±0.0381 | 1.0740±0.0291 | 1.1074±0.0248 | 1.0940±0.0571 | 1.1170±0.0424 | 1.1369±0.0183 |
| $3 \times 10^{-6}$ | 1.0944±0.0144 | 1.1067±0.0313 | 1.0853±0.0478 | 1.1657±0.0502 | 1.0876±0.0485 | 1.1064±0.0251 | 1.1064±0.0428 | 1.1198±0.0351 | 1.1420±0.0470 | 1.1234±0.0185 | 1.1284±0.0406 |
| $1 \times 10^{-6}$ | 1.0992±0.0299 | 1.0725±0.0256 | 1.1476±0.0510 | 1.1265±0.0194 | 1.1256±0.0139 | 1.1005±0.0335 | 1.0683±0.0558 | 1.0980±0.0285 | 1.1159±0.0161 | 1.0996±0.0567 | 1.0926±0.0170 |

Table 9: Numerical results for Gradient Flow with the 8 Gaussians dataset ($d = 50$)

| LR | SW | MaxSW | DSW | MaxKSW | iMSW | viMSW | oMSW | rMSW | EBSW | RPSW | EBRPSW |
|---|---|---|---|---|---|---|---|---|---|---|---|
| 100 | 0.7762±0.1023 | 4.1470±0.0352 | 2.9805±0.0642 | 2.5629±0.4443 | 2.0662±0.1062 | 1.2985±0.1430 | 0.7226±0.0545 | 1.0492±0.0837 | 8.9485±0.8584 | 1.2748±0.0332 | 8.9753±0.7660 |
| 80 | 0.6647±0.0359 | 3.2381±0.0367 | 2.2192±0.2220 | 2.1054±0.0841 | 1.6892±0.1181 | 0.9497±0.1181 | 0.5743±0.0987 | 0.9389±0.05567 | 9.5420±0.9074 | 1.1116±0.07587 | 7.5470±0.5091 |
| 50 | 0.3936±0.0427 | 2.3955±0.5774 | 2.1054±0.1405 | 1.5428±0.0838 | 1.1088±0.0616 | 0.5304±0.0832 | 0.3697±0.0686 | 0.6062±0.0722 | 4.2090±0.0792 | 0.6496±0.0087 | 4.4983±0.4854 |
| 30 | 0.2592±0.0305 | 1.7158±0.0576 | 0.9065±0.0385 | 1.3463±0.0646 | 0.7396±0.0385 | 0.3412±0.0455 | 0.2242±0.0245 | 0.3475±0.0257 | 2.6767±0.0900 | 0.3805±0.0476 | 2.5791±0.0469 |
| 10 | 0.0843±0.0069 | 1.2179±0.0184 | 0.2198±0.0271 | 0.8623±0.0304 | 0.5016±0.0059 | 0.1124±0.0121 | 0.0718±0.0044 | 0.1114±0.0134 | 1.0026±0.0296 | 0.1323±0.0029 | 0.9923±0.0272 |
| 8 | 0.0563±0.0017 | 1.0353±0.0213 | 0.2052±0.0283 | 0.7273±0.0190 | 0.4534±0.0015 | 0.0943±0.0109 | 0.0583±0.0041 | 0.0887±0.0075 | 0.9543±0.0348 | 0.1118±0.0083 | 0.9426±0.0210 |
| 5 | 0.0399±0.0046 | 0.7122±0.0174 | 0.0919±0.0103 | 0.5057±0.0135 | 0.3198±0.0128 | 0.0546±0.0033 | 0.0387±0.0038 | 0.0542±0.0050 | 0.0414±0.0050 | 0.0714±0.0053 | 0.0407±0.0013 |
| 3 | 0.0298±0.0048 | 0.5158±0.0225 | 0.0368±0.0029 | 0.3866±0.0041 | 0.2153±0.0097 | 0.0334±0.0026 | 0.0256±0.0026 | 0.0322±0.0007 | 0.0271±0.0061 | 0.0448±0.0043 | 0.0280±0.0066 |
| 1 | 0.0084±0.0009 | 0.2754±0.0034 | 0.0120±0.0024 | 0.1864±0.0049 | 0.1010±0.0016 | 0.0105±0.0007 | 0.0072±0.0003 | 0.0113±0.0020 | 0.0082±0.0001 | 0.0150±0.0013 | 0.0076±0.0006 |
| $8 \times 10^{-1}$ | 0.0064±0.0006 | 0.2443±0.0031 | 0.0164±0.0065 | 0.1644±0.0012 | 0.0856±0.0006 | 0.0103±0.0005 | 0.0059±0.0009 | 0.0092±0.0001 | 0.0066±0.0007 | 0.0128±0.0010 | 0.0122±0.0084 |
| $5 \times 10^{-1}$ | 0.0039±0.0003 | 0.1865±0.0033 | 0.0061±0.0005 | 0.1229±0.0014 | 0.0555±0.0006 | 0.0058±0.0003 | 0.0044±0.0003 | 0.0056±0.0004 | 0.0042±0.0003 | 0.0075±0.0004 | 0.0040±0.0002 |
| $3 \times 10^{-1}$ | 0.0023±0.0000 | 0.1347±0.0006 | 0.0035±0.0003 | 0.0823±0.0012 | 0.0367±0.0031 | 0.0038±0.0001 | 0.0022±0.0002 | 0.0032±0.0002 | 0.0022±0.0003 | 0.0041±0.0004 | 0.0024±0.0002 |
| $1 \times 10^{-1}$ | 0.0008±0.0001 | 0.0598±0.0005 | 0.0013±0.0000 | 0.0301±0.0000 | 0.0080±0.0001 | 0.0011±0.0002 | 0.0073±0.0092 | 0.0011±0.0000 | 0.0009±0.0001 | 0.0015±0.0000 | 0.0008±0.0000 |
| $8 \times 10^{-2}$ | 0.0006±0.0001 | 0.0479±0.0024 | 0.0011±0.0001 | 0.0232±0.0001 | 0.0059±0.0001 | 0.0010±0.0001 | 0.0006±0.0000 | 0.0009±0.0001 | 0.0007±0.0001 | 0.0011±0.0001 | 0.0007±0.0000 |
| $5 \times 10^{-2}$ | 0.0004±0.0000 | 0.0358±0.0047 | 0.0007±0.0000 | 0.0165±0.0048 | 0.0034±0.0001 | 0.0005±0.0001 | 0.0004±0.0000 | 0.0006±0.0000 | 0.0004±0.0000 | 0.0008±0.0001 | 0.0004±0.0000 |
| $3 \times 10^{-2}$ | 0.0030±0.0040 | 0.0210±0.0051 | 0.0004±0.0000 | 0.0124±0.0082 | 0.0021±0.0001 | 0.0003±0.0000 | 0.0002±0.0000 | 0.0003±0.0000 | 0.0002±0.0000 | 0.0004±0.0000 | 0.0000±0.0000 |
| $1 \times 10^{-2}$ | 0.1316±0.0033 | 0.0501±0.0031 | 0.0302±0.0003 | 0.0598±0.0009 | 0.0906±0.0095 | 0.1090±0.0164 | 0.1320±0.0154 | 0.1244±0.0033 | 0.0472±0.0001 | 0.0001±0.0000 | 0.0073±0.0100 |
| $8 \times 10^{-3}$ | 0.2177±0.0424 | 0.0575±0.0036 | 0.0526±0.0021 | 0.0824±0.0047 | 0.1306±0.0105 | 0.1858±0.0275 | 0.2047±0.0078 | 0.2182±0.0427 | 0.0760±0.0004 | 0.0111±0.0155 | 0.0546±0.0011 |
| $5 \times 10^{-3}$ | 0.3952±0.0247 | 0.1087±0.0024 | 0.1025±0.0098 | 0.1622±0.0081 | 0.3173±0.0275 | 0.3845±0.0282 | 0.4071±0.0187 | 0.4151±0.0329 | 0.1511±0.0098 | 0.0622±0.0067 | 0.1411±0.0066 |
| $3 \times 10^{-3}$ | 0.5586±0.0151 | 0.2257±0.0155 | 0.2471±0.0133 | 0.3464±0.0096 | 0.4916±0.0133 | 0.5395±0.0327 | 0.5485±0.0271 | 0.5645±0.0297 | 0.3053±0.0138 | 0.2302±0.0240 | 0.2800±0.0173 |
| $1 \times 10^{-3}$ | 0.7368±0.0071 | 0.5209±0.0295 | 0.6224±0.0271 | 0.6223±0.0124 | 0.6734±0.0601 | 0.6885±0.0230 | 0.7442±0.0126 | 0.7277±0.0133 | 0.6449±0.0427 | 0.6091±0.0403 | 0.6653±0.0580 |
| $8 \times 10^{-4}$ | 0.7768±0.0225 | 0.6110±0.0293 | 0.6699±0.0478 | 0.6629±0.0105 | 0.7314±0.0266 | 0.7445±0.0226 | 0.7542±0.0457 | 0.7287±0.0117 | 0.6765±0.0522 | 0.6724±0.0188 | 0.6419±0.0300 |
| $5 \times 10^{-4}$ | 0.7490±0.0225 | 0.6869±0.0197 | 0.6954±0.0774 | 0.7295±0.0491 | 0.7891±0.0284 | 0.7690±0.0318 | 0.7695±0.0567 | 0.7668±0.0369 | 0.7451±0.0210 | 0.6965±0.0405 | 0.7216±0.0103 |
| $3 \times 10^{-4}$ | 0.7827±0.0395 | 0.6642±0.0428 | 0.7586±0.0062 | 0.7633±0.0165 | 0.7398±0.0400 | 0.7723±0.0400 | 0.7431±0.0404 | 0.7707±0.0251 | 0.7329±0.0414 | 0.7311±0.0240 | 0.7943±0.0136 |
| $1 \times 10^{-4}$ | 0.8140±0.0313 | 0.7916±0.0414 | 0.7861±0.0208 | 0.7831±0.0390 | 0.8007±0.0424 | 0.8189±0.0443 | 0.7720±0.0456 | 0.7897±0.0402 | 0.8255±0.0249 | 0.7853±0.0086 | 0.7956±0.0433 |
| $8 \times 10^{-5}$ | 0.8251±0.0557 | 0.7742±0.0468 | 0.7564±0.0264 | 0.7817±0.0112 | 0.7730±0.0625 | 0.7962±0.0255 | 0.7882±0.0464 | 0.7486±0.0271 | 0.8021±0.0180 | 0.7926±0.0361 | 0.7754±0.0085 |
| $5 \times 10^{-5}$ | 0.8447±0.0485 | 0.8429±0.0513 | 0.7968±0.0078 | 0.7805±0.0763 | 0.7909±0.0351 | 0.7495±0.0394 | 0.7897±0.0347 | 0.8281±0.0520 | 0.8203±0.0164 | 0.7809±0.0271 | 0.8060±0.0349 |
| $3 \times 10^{-5}$ | 0.8487±0.0216 | 0.8205±0.0185 | 0.8784±0.0763 | 0.8266±0.0327 | 0.7722±0.0272 | 0.7971±0.0129 | 0.8065±0.0415 | 0.7762±0.0201 | 0.7999±0.0260 | 0.7997±0.0335 | 0.8139±0.0303 |
| $1 \times 10^{-5}$ | 0.8185±0.0507 | 0.8229±0.0235 | 0.8475±0.0495 | 0.8096±0.0189 | 0.8023±0.0096 | 0.7732±0.0190 | 0.7889±0.0274 | 0.8114±0.0154 | 0.8616±0.0231 | 0.8170±0.0166 | 0.8138±0.0658 |
| $8 \times 10^{-6}$ | 0.7791±0.0343 | 0.8024±0.0081 | 0.8053±0.0170 | 0.7963±0.0017 | 0.8017±0.0222 | 0.7826±0.0209 | 0.8119±0.0277 | 0.8194±0.0162 | 0.8093±0.0391 | 0.7836±0.0407 | 0.8051±0.0071 |
| $5 \times 10^{-6}$ | 0.8004±0.0436 | 0.8115±0.0065 | 0.7933±0.0083 | 0.8196±0.0535 | 0.8087±0.0554 | 0.8442±0.0464 | 0.8336±0.0498 | 0.7760±0.0264 | 0.7825±0.0188 | 0.8363±0.0375 | 0.8199±0.0169 |
| $3 \times 10^{-6}$ | 0.8095±0.0577 | 0.7820±0.0421 | 0.8062±0.0289 | 0.7960±0.0257 | 0.8325±0.0368 | 0.8312±0.0423 | 0.8435±0.0051 | 0.8024±0.0159 | 0.7826±0.0256 | 0.8178±0.0533 | 0.7688±0.0116 |
| $1 \times 10^{-6}$ | 0.8090±0.0194 | 0.8014±0.0267 | 0.7747±0.0278 | 0.7973±0.0361 | 0.8236±0.0444 | 0.8056±0.0109 | 0.8253±0.0041 | 0.8288±0.0236 | 0.8208±0.0290 | 0.8208±0.0422 | 0.8211±0.0454 |

Table 10: Numerical results for Gradient Flow with the Swiss dataset ($d = 50$)

| LR | SW | MaxSW | DSW | MaxKSW | iMSW | viMSW | oMSW | rMSW | EBSW | RPSW | EBRPSW |
|---|---|---|---|---|---|---|---|---|---|---|---|
| 100 | 0.8643±0.0480 | 2.4103±0.0243 | 3.0249±0.0992 | 2.8531±0.3598 | 1.9101±0.0904 | 0.9712±0.0227 | 0.8463±0.0478 | 0.9859±0.0645 | 16.7324±0.3058 | 0.8955±0.0380 | 16.3655±0.8700 |
| 80 | 0.6125±0.0264 | 1.7029±0.0444 | 2.2926±0.1929 | 2.2264±0.1102 | 1.5574±0.1757 | 0.7954±0.0465 | 0.6772±0.0312 | 0.7505±0.0809 | 10.8613±0.1851 | 0.7372±0.0058 | 12.4947±0.8208 |
| 50 | 0.4262±0.0147 | 2.0519±0.0738 | 1.5044±0.0269 | 1.7652±0.2612 | 1.1316±0.0373 | 0.4388±0.0366 | 0.4079±0.0182 | 0.4803±0.0080 | 7.0184±0.8989 | 0.4605±0.0160 | 7.5545±0.5874 |
| 30 | 0.2685±0.0022 | 2.0192±0.1526 | 0.9237±0.0327 | 1.3850±0.1726 | 0.7873±0.0145 | 0.2954±0.0180 | 0.2619±0.0018 | 0.3041±0.0170 | 3.7698±0.2624 | 0.2759±0.0187 | 4.3376±0.2378 |
| 10 | 0.0972±0.0068 | 1.3748±0.0235 | 0.2432±0.0166 | 1.1019±0.0105 | 0.6308±0.0093 | 0.1131±0.0041 | 0.0959±0.0036 | 0.1160±0.0059 | 1.3600±0.0370 | 0.1129±0.0066 | 1.3044±0.0268 |
| 8 | 0.0767±0.0032 | 1.2367±0.0192 | 0.1845±0.0193 | 0.9386±0.0170 | 0.5502±0.0083 | 0.0966±0.0035 | 0.0768±0.0016 | 0.0957±0.0045 | 1.2489±0.0797 | 0.0904±0.0078 | 1.2369±0.0945 |
| 5 | 0.0558±0.0024 | 0.9070±0.0095 | 0.0955±0.0141 | 0.6769±0.0079 | 0.3837±0.0111 | 0.0724±0.0017 | 0.0563±0.0020 | 0.0718±0.0008 | 0.0564±0.0016 | 0.0581±0.0027 | 0.0559±0.0017 |
| 3 | 0.0430±0.0009 | 0.6628±0.0086 | 0.0400±0.0069 | 0.4668±0.0040 | 0.2629±0.0053 | 0.0583±0.0009 | 0.0437±0.0001 | 0.0611±0.0002 | 0.0441±0.0005 | 0.0349±0.0015 | 0.0432±0.0005 |
| 1 | 0.0224±0.0001 | 0.3285±0.0080 | 0.0095±0.0010 | 0.2235±0.0048 | 0.1153±0.0003 | 0.0296±0.0004 | 0.0224±0.0002 | 0.0296±0.0004 | 0.0225±0.0003 | 0.0109±0.0006 | 0.0225±0.0003 |
| $8 \times 10^{-1}$ | 0.0165±0.0001 | 0.2840±0.0054 | 0.0088±0.0004 | 0.1902±0.0019 | 0.1007±0.0003 | 0.0164±0.0003 | 0.0166±0.0002 | 0.0163±0.0002 | 0.0165±0.0001 | 0.0104±0.0004 | 0.0167±0.0002 |
| $5 \times 10^{-1}$ | 0.0080±0.0000 | 0.2212±0.0012 | 0.0057±0.0008 | 0.1405±0.0036 | 0.0740±0.0003 | 0.0046±0.0003 | 0.0080±0.0000 | 0.0047±0.0001 | 0.0080±0.0001 | 0.0067±0.0009 | 0.0080±0.0001 |
| $3 \times 10^{-1}$ | 0.0046±0.0001 | 0.1589±0.0011 | 0.0035±0.0006 | 0.1022±0.0014 | 0.0498±0.0002 | 0.0029±0.0001 | 0.0046±0.0001 | 0.0030±0.0000 | 0.0045±0.0000 | 0.0038±0.0002 | 0.0045±0.0001 |
| $1 \times 10^{-1}$ | 0.0013±0.0002 | 0.0787±0.0024 | 0.0012±0.0001 | 0.0449±0.0002 | 0.0149±0.0001 | 0.0010±0.0000 | 0.0015±0.0000 | 0.0010±0.0000 | 0.0013±0.0002 | 0.0012±0.0000 | 0.0013±0.0002 |
| $8 \times 10^{-2}$ | 0.0011±0.0001 | 0.0650±0.0015 | 0.0010±0.0001 | 0.0410±0.0001 | 0.0117±0.0001 | 0.0008±0.0001 | 0.0012±0.0001 | 0.0008±0.0000 | 0.0012±0.0000 | 0.0010±0.0000 | 0.0012±0.0000 |
| $5 \times 10^{-2}$ | 0.0004±0.0000 | 0.0485±0.0026 | 0.0007±0.0001 | 0.0257±0.0003 | 0.0075±0.0003 | 0.0005±0.0000 | 0.0004±0.0000 | 0.0005±0.0000 | 0.0004±0.0000 | 0.0006±0.0001 | 0.0004±0.0000 |
| $3 \times 10^{-2}$ | 0.0045±0.0060 | 0.0390±0.0057 | 0.0004±0.0000 | 0.0242±0.0000 | 0.0115±0.0104 | 0.0257±0.0119 | 0.0105±0.0048 | 0.0343±0.0043 | 0.0003±0.0000 | 0.0004±0.0000 | 0.0003±0.0000 |
| $1 \times 10^{-2}$ | 0.2804±0.0226 | 0.0720±0.0046 | 0.0502±0.0035 | 0.1019±0.0020 | 0.1841±0.0085 | 0.2933±0.0198 | 0.2789±0.0208 | 0.3458±0.0357 | 0.1314±0.0357 | 0.0346±0.0075 | 0.1413±0.0088 |
| $8 \times 10^{-3}$ | 0.3572±0.0416 | 0.0894±0.0071 | 0.0633±0.0024 | 0.1397±0.0037 | 0.2467±0.0085 | 0.4134±0.0194 | 0.3828±0.0295 | 0.4188±0.0111 | 0.1794±0.0026 | 0.0640±0.0079 | 0.1718±0.0033 |
| $5 \times 10^{-3}$ | 0.7284±0.0214 | 0.1556±0.0114 | 0.1280±0.0096 | 0.2343±0.0145 | 0.4257±0.0203 | 0.5644±0.0408 | 0.5199±0.0729 | 0.5625±0.0292 | 0.2970±0.0092 | 0.1951±0.0367 | 0.3066±0.0145 |
| $3 \times 10^{-3}$ | 0.5855±0.0378 | 0.2609±0.0091 | 0.2891±0.0310 | 0.3839±0.0267 | 0.5528±0.0394 | 0.6505±0.0218 | 0.6237±0.0231 | 0.6863±0.0198 | 0.4115±0.0076 | 0.4189±0.0281 | 0.4130±0.0205 |
| $1 \times 10^{-3}$ | 0.7769±0.0223 | 0.5345±0.0245 | 0.6896±0.0436 | 0.6344±0.0232 | 0.6984±0.0159 | 0.7895±0.0378 | 0.7531±0.0458 | 0.7658±0.0333 | 0.6480±0.0414 | 0.5981±0.0180 | 0.6073±0.0028 |
| $8 \times 10^{-4}$ | 0.7223±0.0554 | 0.5782±0.0436 | 0.6573±0.0227 | 0.6719±0.0370 | 0.6832±0.0032 | 0.8203±0.0277 | 0.7193±0.0335 | 0.7771±0.0393 | 0.6816±0.0188 | 0.6958±0.0269 | 0.6699±0.0186 |
| $5 \times 10^{-4}$ | 0.5284±0.0214 | 0.6497±0.0214 | 0.7132±0.0115 | 0.7223±0.0115 | 0.7746±0.0050 | 0.7589±0.0108 | 0.7435±0.0341 | 0.8034±0.0074 | 0.7169±0.0187 | 0.6618±0.0051 | 0.6806±0.0069 |
| $3 \times 10^{-4}$ | 0.7738±0.0239 | 0.6827±0.0282 | 0.7920±0.0172 | 0.7138±0.0237 | 0.7059±0.0415 | 0.7849±0.0082 | 0.7744±0.0022 | 0.8368±0.0076 | 0.7549±0.0262 | 0.7491±0.0323 | 0.7581±0.0283 |
| $1 \times 10^{-4}$ | 0.7769±0.0223 | 0.7339±0.0433 | 0.8135±0.0433 | 0.7113±0.0696 | 0.7601±0.0250 | 0.8799±0.0589 | 0.7841±0.0240 | 0.8516±0.0175 | 0.7906±0.0125 | 0.7495±0.0050 | 0.7495±0.0843 |
| $8 \times 10^{-5}$ | 0.7693±0.0421 | 0.7755±0.0336 | 0.8598±0.0336 | 0.7652±0.0077 | 0.7680±0.0212 | 0.8133±0.0309 | 0.8206±0.0287 | 0.8416±0.0336 | 0.7758±0.0432 | 0.7305±0.0133 | 0.7520±0.0274 |
| $5 \times 10^{-5}$ | 0.7831±0.0237 | 0.7964±0.0122 | 0.7991±0.0122 | 0.7612±0.0563 | 0.7013±0.0231 | 0.8473±0.0435 | 0.8103±0.0092 | 0.8216±0.0423 | 0.7741±0.0211 | 0.7610±0.0130 | 0.8096±0.0157 |
| $3 \times 10^{-5}$ | 0.7928±0.0350 | 0.7748±0.0484 | 0.8389±0.0374 | 0.7534±0.0318 | 0.7539±0.0115 | 0.8650±0.0087 | 0.7570±0.0352 | 0.8469±0.0187 | 0.7786±0.0070 | 0.7785±0.0239 | 0.7547±0.0152 |
| $1 \times 10^{-5}$ | 0.7588±0.0398 | 0.7469±0.0374 | 0.8472±0.0336 | 0.7930±0.0286 | 0.7672±0.0217 | 0.8318±0.0185 | 0.7799±0.0125 | 0.8329±0.0328 | 0.7835±0.0145 | 0.7804±0.0246 | 0.7885±0.0138 |
| $8 \times 10^{-6}$ | 0.7720±0.0243 | 0.7906±0.0141 | 0.8344±0.0208 | 0.7622±0.0255 | 0.7477±0.0283 | 0.8835±0.0241 | 0.7494±0.0229 | 0.8103±0.0076 | 0.7818±0.0347 | 0.7729±0.0175 | 0.7692±0.0128 |
| $5 \times 10^{-6}$ | 0.7414±0.0346 | 0.7499±0.0495 | 0.8574±0.0395 | 0.8278±0.0333 | 0.7953±0.0533 | 0.8492±0.0526 | 0.7898±0.0170 | 0.8617±0.0261 | 0.7642±0.0455 | 0.7480±0.0184 | 0.7467±0.0161 |
| $3 \times 10^{-6}$ | 0.8039±0.0412 | 0.7761±0.0199 | 0.8475±0.0209 | 0.8175±0.0604 | 0.7411±0.0401 | 0.8466±0.0131 | 0.7385±0.0290 | 0.8266±0.0428 | 0.7709±0.0121 | 0.8049±0.0306 | 0.7930±0.0426 |
| $1 \times 10^{-6}$ | 0.7711±0.0045 | 0.7751±0.0052 | 0.8456±0.0207 | 0.8004±0.0423 | 0.7304±0.0497 | 0.8543±0.0218 | 0.7518±0.0316 | 0.8319±0.0460 | 0.7475±0.0265 | 0.7541±0.0503 | 0.7424±0.0243 |

Table 11: Numerical results for Gradient Flow with the Knot dataset ($d = 100$)

| LR | SW | MaxSW | DSW | MaxKSW | iMSW | viMSW | oMSW | rMSW | EBSW | RPSW | EBRPSW |
|---|---|---|---|---|---|---|---|---|---|---|---|
| 100 | 0.8999±0.0210 | 2.7462±0.0473 | 2.8735±0.1213 | 3.2882±0.2012 | 2.0599±0.0665 | 0.9238±0.0107 | 0.8228±0.0550 | 1.0054±0.0460 | 11.1502±0.5452 | 0.9537±0.0065 | 11.5940±0.2757 |
| 80 | 0.6750±0.0370 | 2.2929±0.0425 | 2.3658±0.1707 | 2.8335±0.0114 | 1.6259±0.0782 | 0.7803±0.0317 | 0.6715±0.0205 | 0.8045±0.0388 | 9.5332±0.1474 | 0.7804±0.0251 | 9.4759±0.1591 |
| 50 | 0.3877±0.0209 | 3.0593±0.5077 | 1.5513±0.0461 | 2.5637±0.1616 | 1.1430±0.0018 | 0.4871±0.0066 | 0.4036±0.0349 | 0.4982±0.0160 | 5.9085±0.0718 | 0.5273±0.0293 | 6.1440±0.0923 |
| 30 | 0.2629±0.0154 | 2.4795±0.3492 | 0.9380±0.0379 | 2.3604±0.1002 | 1.3570±0.0652 | 0.2752±0.0008 | 0.2549±0.0194 | 0.2956±0.0055 | 3.7218±0.1138 | 0.3304±0.0195 | 3.6760±0.0457 |
| 10 | 0.0815±0.0022 | 1.6234±0.0147 | 0.2669±0.0078 | 1.2923±0.0119 | 0.8413±0.0116 | 0.0996±0.0031 | 0.0862±0.0043 | 0.0859±0.0026 | 1.8692±0.0718 | 0.1037±0.0045 | 1.9169±0.0229 |
| 8 | 0.0676±0.0019 | 1.5109±0.0179 | 0.1902±0.0150 | 1.1465±0.0356 | 0.6969±0.0184 | 0.0778±0.0033 | 0.0684±0.0045 | 0.0781±0.0031 | 1.8447±0.0816 | 0.0903±0.0086 | 1.8146±0.1152 |
| 5 | 0.0446±0.0021 | 1.1718±0.0410 | 0.0936±0.0274 | 0.8477±0.0221 | 0.4585±0.0021 | 0.0440±0.0019 | 0.0412±0.0005 | 0.0473±0.0021 | 0.0454±0.0038 | 0.0558±0.0023 | 0.0435±0.0026 |
| 3 | 0.0279±0.0011 | 0.9264±0.0011 | 0.0375±0.0020 | 0.6073±0.0108 | 0.2748±0.0066 | 0.0288±0.0008 | 0.0288±0.0013 | 0.0292±0.0011 | 0.0280±0.0014 | 0.0360±0.0014 | 0.0282±0.0009 |
| 1 | 0.0133±0.0003 | 0.4048±0.0096 | 0.0106±0.0007 | 0.2178±0.0017 | 0.0716±0.0009 | 0.0124±0.0006 | 0.0140±0.0002 | 0.0119±0.0005 | 0.0137±0.0010 | 0.0134±0.0002 | 0.0133±0.0004 |
| $8 \times 10^{-1}$ | 0.0120±0.0003 | 0.3242±0.0024 | 0.0091±0.0015 | 0.1738±0.0022 | 0.0590±0.0012 | 0.0104±0.0004 | 0.0125±0.0005 | 0.0105±0.0005 | 0.0124±0.0002 | 0.0100±0.0004 | 0.0128±0.0007 |
| $5 \times 10^{-1}$ | 0.0098±0.0001 | 0.1982±0.0021 | 0.0055±0.0004 | 0.0843±0.0015 | 0.0424±0.0009 | 0.0077±0.0004 | 0.0096±0.0001 | 0.0078±0.0002 | 0.0097±0.0002 | 0.0065±0.0005 | 0.0098±0.0001 |
| $3 \times 10^{-1}$ | 0.0057±0.0000 | 0.0981±0.0066 | 0.0035±0.0001 | 0.0578±0.0012 | 0.0312±0.0008 | 0.0058±0.0001 | 0.0059±0.0000 | 0.0058±0.0000 | 0.0057±0.0001 | 0.0038±0.0001 | 0.0058±0.0001 |
| $1 \times 10^{-1}$ | 0.0016±0.0001 | 0.0439±0.0013 | 0.0013±0.0001 | 0.0290±0.0003 | 0.0159±0.0003 | 0.0010±0.0000 | 0.0015±0.0000 | 0.0009±0.0001 | 0.0015±0.0001 | 0.0013±0.0001 | 0.0016±0.0001 |
| $8 \times 10^{-2}$ | 0.0010±0.0000 | 0.0385±0.0006 | 0.0010±0.0001 | 0.0261±0.0002 | 0.0135±0.0002 | 0.0008±0.0000 | 0.0011±0.0002 | 0.0008±0.0000 | 0.0011±0.0000 | 0.0010±0.0000 | 0.0011±0.0001 |
| $5 \times 10^{-2}$ | 0.0421±0.0445 | 0.0297±0.0008 | 0.0006±0.0001 | 0.0200±0.0001 | 0.0462±0.0330 | 0.0350±0.0350 | 0.0308±0.0428 | 0.0279±0.0351 | 0.0005±0.0000 | 0.0007±0.0000 | 0.0004±0.0000 |
| $3 \times 10^{-2}$ | 0.3033±0.0398 | 0.0219±0.0003 | 0.0004±0.0000 | 0.1169±0.0074 | 0.2579±0.0305 | 0.2547±0.0021 | 0.2924±0.0156 | 0.2703±0.0201 | 0.0600±0.0394 | 0.0004±0.0000 | 0.0803±0.0208 |
| $1 \times 10^{-2}$ | 0.6815±0.0581 | 0.3337±0.0280 | 0.2210±0.0083 | 0.4191±0.0139 | 0.5300±0.0373 | 0.6595±0.0196 | 0.6801±0.0112 | 0.6874±0.0216 | 0.4444±0.0156 | 0.3403±0.0255 | 0.4169±0.0118 |
| $8 \times 10^{-3}$ | 0.7851±0.0479 | 0.3731±0.0236 | 0.3234±0.0314 | 0.4651±0.0083 | 0.6044±0.0072 | 0.6893±0.0299 | 0.7789±0.0269 | 0.7410±0.0204 | 0.4848±0.0111 | 0.4244±0.0203 | 0.5039±0.0169 |
| $5 \times 10^{-3}$ | 0.8508±0.0152 | 0.4733±0.0185 | 0.4979±0.0280 | 0.5620±0.0089 | 0.7247±0.0213 | 0.8281±0.0402 | 0.8774±0.0232 | 0.9089±0.0383 | 0.6009±0.0188 | 0.6378±0.0079 | 0.6152±0.0155 |
| $3 \times 10^{-3}$ | 1.0324±0.0447 | 0.5853±0.0110 | 0.6446±0.0219 | 0.6916±0.0141 | 0.8709±0.0107 | 0.9276±0.0026 | 0.9828±0.0431 | 0.9791±0.0088 | 0.7829±0.0191 | 0.8020±0.0098 | 0.7405±0.0263 |
| $1 \times 10^{-3}$ | 1.0512±0.0197 | 0.8431±0.0326 | 0.8869±0.0205 | 0.9131±0.0176 | 1.0226±0.0193 | 1.0683±0.0350 | 1.0648±0.0060 | 1.0571±0.0341 | 1.0056±0.0210 | 1.0151±0.0527 | 1.0046±0.0041 |
| $8 \times 10^{-4}$ | 1.0779±0.0182 | 0.8908±0.0561 | 0.9720±0.0380 | 0.9676±0.0211 | 1.0130±0.0372 | 1.0758±0.0151 | 1.0793±0.0137 | 1.0885±0.0170 | 1.0231±0.0193 | 1.0303±0.0372 | 1.0333±0.0203 |
| $5 \times 10^{-4}$ | 1.0988±0.0184 | 0.9505±0.0212 | 1.0193±0.0193 | 0.9798±0.0270 | 1.0378±0.0249 | 1.1476±0.0273 | 1.1071±0.0083 | 1.0677±0.0578 | 1.0552±0.0253 | 1.0367±0.0548 | 1.0364±0.0551 |
| $3 \times 10^{-4}$ | 1.1348±0.0345 | 1.0047±0.0080 | 1.0176±0.0270 | 1.0377±0.0328 | 1.0758±0.0090 | 1.1305±0.0379 | 1.1396±0.0675 | 1.1136±0.0423 | 1.0734±0.0079 | 1.0659±0.0355 | 1.0757±0.0183 |
| $1 \times 10^{-4}$ | 1.0925±0.0405 | 1.0953±0.0485 | 1.1185±0.0088 | 1.0751±0.0584 | 1.0750±0.0136 | 1.0832±0.0300 | 1.0511±0.0267 | 1.1315±0.0190 | 1.0838±0.0455 | 1.1096±0.0402 | 1.1157±0.0268 |
| $8 \times 10^{-5}$ | 1.1123±0.0250 | 1.0700±0.0271 | 1.0796±0.0224 | 1.0788±0.0533 | 1.1064±0.0140 | 1.1237±0.0328 | 1.0980±0.0204 | 1.1517±0.0253 | 1.0981±0.0303 | 1.0822±0.0578 | 1.0918±0.0191 |
| $5 \times 10^{-5}$ | 1.1325±0.0183 | 1.0869±0.0071 | 1.1188±0.0054 | 1.1100±0.0342 | 1.1137±0.0182 | 1.0774±0.0188 | 1.1139±0.0363 | 1.1536±0.0548 | 1.1520±0.0330 | 1.0898±0.0257 | 1.0977±0.0389 |
| $3 \times 10^{-5}$ | 1.0957±0.0242 | 1.1209±0.0221 | 1.0884±0.0234 | 1.0532±0.0352 | 1.0918±0.0186 | 1.1116±0.0255 | 1.1636±0.0557 | 1.0753±0.0131 | 1.1040±0.0511 | 1.0965±0.0552 | 1.0595±0.0265 |
| $1 \times 10^{-5}$ | 1.1133±0.0149 | 1.1525±0.0283 | 1.0880±0.0347 | 1.1194±0.0511 | 1.1360±0.0050 | 1.0627±0.0312 | 1.1375±0.0438 | 1.0889±0.0484 | 1.1512±0.1043 | 1.1299±0.0701 | 1.0808±0.0393 |
| $8 \times 10^{-6}$ | 1.1463±0.0507 | 1.1160±0.0364 | 1.1042±0.0119 | 1.1087±0.0458 | 1.1419±0.0305 | 1.0656±0.0384 | 1.1161±0.0382 | 1.1148±0.0397 | 1.1235±0.0733 | 1.1152±0.0240 | 1.1092±0.0640 |
| $5 \times 10^{-6}$ | 1.0819±0.0338 | 1.0975±0.0435 | 1.0985±0.0218 | 1.0971±0.0492 | 1.1398±0.0347 | 1.1297±0.0235 | 1.1038±0.0064 | 1.1181±0.0381 | 1.1171±0.0500 | 1.0907±0.0028 | 1.1375±0.0591 |
| $3 \times 10^{-6}$ | 1.0974±0.0341 | 1.1184±0.0258 | 1.0649±0.0178 | 1.1366±0.0448 | 1.0637±0.0139 | 1.1125±0.0327 | 1.0915±0.0324 | 1.1307±0.0267 | 1.1339±0.0241 | 1.1261±0.0402 | 1.1100±0.0351 |
| $1 \times 10^{-6}$ | 1.1113±0.0431 | 1.0936±0.0302 | 1.1580±0.0174 | 1.0748±0.0342 | 1.1037±0.0867 | 1.0442±0.0314 | 1.0524±0.0236 | 1.0905±0.0209 | 1.1355±0.0364 | 1.1215±0.0309 | 1.0954±0.0074 |

Table 12: Numerical results for Gradient Flow with the 8 Gaussians dataset ($d = 100$)

| LR | SW | MaxSW | DSW | MaxKSW | iMSW | viMSW | oMSW | rMSW | EBSW | RPSW | EBRPSW |
|---|---|---|---|---|---|---|---|---|---|---|---|
| 100 | 0.8453±0.0023 | 1.7094±0.0385 | 2.8643±0.2356 | 2.6594±0.1227 | 2.1708±0.0707 | 0.8819±0.0498 | 0.8580±0.0126 | 1.0229±0.0815 | 14.9925±0.9149 | 0.9446±0.0524 | 14.6610±0.9302 |
| 80 | 0.6462±0.0144 | 2.3625±1.6301 | 2.5657±0.0937 | 2.1892±0.1915 | 1.6570±0.0787 | 0.7463±0.0104 | 0.6837±0.0626 | 0.7806±0.0436 | 10.8735±1.6920 | 0.7065±0.0500 | 11.0594±0.9880 |
| 50 | 0.4404±0.0069 | 1.8941±0.0347 | 1.4256±0.0602 | 1.7448±0.0856 | 1.1720±0.0426 | 0.5025±0.0142 | 0.4153±0.0104 | 0.5028±0.0436 | 6.8204±0.2040 | 0.4404±0.0366 | 6.4021±0.4725 |
| 30 | 0.2634±0.0069 | 1.9391±0.0814 | 0.8874±0.0252 | 1.3388±0.0205 | 0.8062±0.0750 | 0.2830±0.0113 | 0.2585±0.0043 | 0.2728±0.0145 | 4.2188±0.4268 | 0.2838±0.0237 | 3.8089±0.3879 |
| 10 | 0.0925±0.0017 | 1.1779±0.1149 | 0.2637±0.0336 | 1.0617±0.0060 | 0.6473±0.0297 | 0.1026±0.0080 | 0.0863±0.0036 | 0.1088±0.0061 | 1.2991±0.0956 | 0.1026±0.0062 | 1.4357±0.1469 |
| 8 | 0.0805±0.0066 | 1.1334±0.0718 | 0.1843±0.0303 | 0.8777±0.0231 | 0.5424±0.0050 | 0.1026±0.0049 | 0.0720±0.0038 | 0.0819±0.0019 | 1.1654±0.0456 | 0.0922±0.0077 | 1.1620±0.0320 |
| 5 | 0.0511±0.0016 | 0.8703±0.0221 | 0.0881±0.0173 | 0.6358±0.0074 | 0.3933±0.0074 | 0.0533±0.0052 | 0.0512±0.0027 | 0.0461±0.0020 | 0.0530±0.0013 | 0.0613±0.0094 | 0.0513±0.0030 |
| 3 | 0.0380±0.0004 | 0.6049±0.0037 | 0.0383±0.0082 | 0.4436±0.0070 | 0.2733±0.0050 | 0.0265±0.0031 | 0.0390±0.0005 | 0.0279±0.0015 | 0.0366±0.0013 | 0.0359±0.0020 | 0.0411±0.0050 |
| 1 | 0.0177±0.0003 | 0.3048±0.0111 | 0.0111±0.0010 | 0.2203±0.0029 | 0.1282±0.0017 | 0.0100±0.0004 | 0.0210±0.0050 | 0.0098±0.0004 | 0.0175±0.0003 | 0.0131±0.0008 | 0.0178±0.0002 |
| $8 \times 10^{-1}$ | 0.0132±0.0002 | 0.2803±0.0073 | 0.0144±0.0094 | 0.2008±0.0039 | 0.1082±0.0018 | 0.0080±0.0005 | 0.0132±0.0001 | 0.0079±0.0002 | 0.0130±0.0003 | 0.0104±0.0008 | 0.0129±0.0001 |
| $5 \times 10^{-1}$ | 0.0070±0.0002 | 0.2261±0.0033 | 0.0052±0.0006 | 0.1506±0.0017 | 0.0731±0.0010 | 0.0046±0.0001 | 0.0071±0.0001 | 0.0049±0.0002 | 0.0071±0.0002 | 0.0062±0.0001 | 0.0070±0.0001 |
| $3 \times 10^{-1}$ | 0.0035±0.0001 | 0.1716±0.0022 | 0.0036±0.0003 | 0.1090±0.0017 | 0.0521±0.0018 | 0.0097±0.0098 | 0.0049±0.0000 | 0.0029±0.0003 | 0.0037±0.0000 | 0.0037±0.0003 | 0.0035±0.0001 |
| $1 \times 10^{-1}$ | 0.0009±0.0000 | 0.0774±0.0000 | 0.0012±0.0001 | 0.0426±0.0005 | 0.0124±0.0001 | 0.0010±0.0000 | 0.0035±0.0000 | 0.0081±0.0101 | 0.0010±0.0000 | 0.0013±0.0001 | 0.0009±0.0000 |
| $8 \times 10^{-2}$ | 0.0007±0.0000 | 0.0675±0.0002 | 0.0009±0.0000 | 0.0357±0.0027 | 0.0093±0.0003 | 0.0008±0.0000 | 0.0007±0.0000 | 0.0008±0.0000 | 0.0007±0.0000 | 0.0010±0.0000 | 0.0007±0.0000 |
| $5 \times 10^{-2}$ | 0.0004±0.0000 | 0.0507±0.0009 | 0.0007±0.0000 | 0.0209±0.0006 | 0.0050±0.0001 | 0.0005±0.0000 | 0.0005±0.0000 | 0.0005±0.0000 | 0.0005±0.0000 | 0.0007±0.0000 | 0.0004±0.0000 |
| $3 \times 10^{-2}$ | 0.0189±0.0099 | 0.0342±0.0022 | 0.0004±0.0000 | 0.0122±0.0000 | 0.0129±0.0010 | 0.0116±0.0092 | 0.0075±0.0002 | 0.0195±0.0063 | 0.0081±0.0111 | 0.0004±0.0000 | 0.0003±0.0000 |
| $1 \times 10^{-2}$ | 0.2747±0.0110 | 0.0596±0.0018 | 0.0432±0.0029 | 0.0831±0.0027 | 0.1521±0.0144 | 0.2559±0.0187 | 0.2458±0.0317 | 0.2713±0.0137 | 0.1024±0.0050 | 0.0238±0.0042 | 0.1028±0.0059 |
| $8 \times 10^{-3}$ | 0.3540±0.0134 | 0.0722±0.0044 | 0.0560±0.0033 | 0.1025±0.0034 | 0.2334±0.0147 | 0.3502±0.0444 | 0.3782±0.0345 | 0.3498±0.0251 | 0.1400±0.0049 | 0.0539±0.0031 | 0.1535±0.0130 |
| $5 \times 10^{-3}$ | 0.5383±0.0144 | 0.1306±0.0024 | 0.1028±0.0079 | 0.1954±0.0088 | 0.4190±0.0172 | 0.4794±0.0088 | 0.5292±0.0509 | 0.5352±0.0263 | 0.2629±0.0107 | 0.1642±0.0141 | 0.2691±0.0017 |
| $3 \times 10^{-3}$ | 0.6450±0.0303 | 0.2361±0.0044 | 0.2410±0.0100 | 0.3577±0.0077 | 0.5810±0.0077 | 0.6048±0.0730 | 0.6123±0.0442 | 0.6584±0.0437 | 0.3916±0.0181 | 0.4047±0.0529 | 0.3907±0.0190 |
| $1 \times 10^{-3}$ | 0.7594±0.0267 | 0.5412±0.0211 | 0.6013±0.0194 | 0.6430±0.0222 | 0.7360±0.0238 | 0.7566±0.0276 | 0.7211±0.0482 | 0.7498±0.0312 | 0.6015±0.0460 | 0.6453±0.0495 | 0.7185±0.0149 |
| $8 \times 10^{-4}$ | 0.7160±0.0087 | 0.5678±0.0112 | 0.6477±0.0109 | 0.6684±0.0039 | 0.7186±0.0174 | 0.7828±0.0174 | 0.7589±0.0165 | 0.7703±0.0604 | 0.6718±0.0382 | 0.6836±0.0211 | 0.7163±0.0128 |
| $5 \times 10^{-4}$ | 0.7653±0.0262 | 0.6971±0.0526 | 0.7226±0.0039 | 0.6803±0.0215 | 0.7617±0.0215 | 0.7912±0.0393 | 0.7891±0.0067 | 0.7953±0.0266 | 0.7830±0.0425 | 0.7378±0.0145 | 0.6751±0.0220 |
| $3 \times 10^{-4}$ | 0.8457±0.0352 | 0.7321±0.0326 | 0.7639±0.0215 | 0.7465±0.0105 | 0.7948±0.0305 | 0.7908±0.0204 | 0.7646±0.0033 | 0.7864±0.0466 | 0.7100±0.0429 | 0.7491±0.0203 | 0.7938±0.0186 |
| $1 \times 10^{-4}$ | 0.8093±0.0227 | 0.7885±0.0195 | 0.7749±0.0105 | 0.7928±0.0380 | 0.8168±0.0380 | 0.8058±0.0158 | 0.8151±0.0481 | 0.8039±0.0251 | 0.7846±0.0221 | 0.7626±0.0589 | 0.8299±0.0233 |
| $8 \times 10^{-5}$ | 0.7885±0.0525 | 0.7841±0.0170 | 0.8303±0.0346 | 0.8011±0.0606 | 0.8028±0.0606 | 0.7600±0.0358 | 0.7743±0.0226 | 0.8303±0.0111 | 0.8039±0.0166 | 0.7977±0.0512 | 0.7738±0.0072 |
| $5 \times 10^{-5}$ | 0.7991±0.0525 | 0.7783±0.0233 | 0.7970±0.0311 | 0.8237±0.0238 | 0.8331±0.0238 | 0.7835±0.0377 | 0.7558±0.0366 | 0.8068±0.0126 | 0.7847±0.0526 | 0.7831±0.0308 | 0.7982±0.0385 |
| $3 \times 10^{-5}$ | 0.8101±0.0105 | 0.7992±0.0453 | 0.8197±0.0457 | 0.7834±0.0271 | 0.7962±0.0431 | 0.8351±0.0116 | 0.7937±0.0116 | 0.7483±0.0244 | 0.7932±0.0143 | 0.8096±0.0368 | 0.8043±0.0252 |
| $1 \times 10^{-5}$ | 0.8117±0.0263 | 0.8169±0.0375 | 0.7834±0.0271 | 0.8115±0.0069 | 0.8231±0.0475 | 0.8176±0.0199 | 0.8142±0.0199 | 0.8302±0.0278 | 0.7942±0.0126 | 0.8152±0.0308 | 0.8371±0.0359 |
| $8 \times 10^{-6}$ | 0.7595±0.0237 | 0.8265±0.0131 | 0.8115±0.0069 | 0.8196±0.0368 | 0.8357±0.0154 | 0.8271±0.0202 | 0.8130±0.0471 | 0.8007±0.0579 | 0.8018±0.0192 | 0.8395±0.0284 | 0.8022±0.0566 |
| $5 \times 10^{-6}$ | 0.7845±0.0314 | 0.8336±0.0266 | 0.7732±0.0185 | 0.8508±0.0186 | 0.7954±0.0368 | 0.8371±0.0248 | 0.8227±0.0695 | 0.8602±0.0203 | 0.8058±0.0152 | 0.8052±0.0382 | 0.8197±0.0247 |
| $3 \times 10^{-6}$ | 0.8083±0.0300 | 0.7975±0.0232 | 0.8054±0.0373 | 0.7585±0.0418 | 0.8474±0.0186 | 0.8087±0.0283 | 0.8302±0.0183 | 0.8099±0.0333 | 0.8230±0.0159 | 0.7857±0.0354 | 0.8152±0.0362 |
| $1 \times 10^{-6}$ | 0.7842±0.0003 | 0.7709±0.0266 | 0.8146±0.0384 | 0.8412±0.0330 | 0.7549±0.0221 | 0.7657±0.0264 | 0.8302±0.0183 | 0.8425±0.0392 | 0.7705±0.0302 | 0.8207±0.0338 | 0.7887±0.0366 |

Table 13: Numerical results for Gradient Flow with the Swiss dataset ($d = 100$)

| LR | SW | MaxSW | DSW | MaxKSW | iMSW | viMSW | oMSW | rMSW | EBSW | RPSW | EBRPSW |
|---|---|---|---|---|---|---|---|---|---|---|---|
| 100 | $294.00 \pm 0.38$ | $287.00 \pm 4.65$ | $227.00 \pm 4.40$ | $275.00 \pm 7.53$ | $286.00 \pm 7.55$ | $286.00 \pm 7.36$ | $294.00 \pm 0.38$ | $201.00 \pm 3.34$ | $291.00 \pm 0.71$ | $291.00 \pm 0.79$ | $291.00 \pm 0.52$ |
| 80 | $294.00 \pm 0.38$ | $286.00 \pm 4.00$ | $227.00 \pm 4.49$ | $275.00 \pm 7.71$ | $286.00 \pm 7.55$ | $286.00 \pm 7.36$ | $294.00 \pm 0.38$ | $168.00 \pm 4.27$ | $289.00 \pm 0.56$ | $289.00 \pm 0.54$ | $290.00 \pm 0.47$ |
| 50 | $294.00 \pm 0.38$ | $280.00 \pm 4.74$ | $228.00 \pm 5.45$ | $276.00 \pm 0.85$ | $286.00 \pm 7.55$ | $286.00 \pm 7.36$ | $294.00 \pm 0.38$ | $147.00 \pm 0.45$ | $282.00 \pm 0.86$ | $282.00 \pm 1.65$ | $284.00 \pm 0.81$ |
| 30 | $294.00 \pm 0.38$ | $263.00 \pm 6.38$ | $232.00 \pm 11.00$ | $258.00 \pm 1.30$ | $286.00 \pm 7.50$ | $286.00 \pm 7.36$ | $294.00 \pm 0.38$ | $162.00 \pm 7.25$ | $232.00 \pm 35.60$ | $272.00 \pm 8.80$ | $268.00 \pm 0.79$ |
| 10 | $294.00 \pm 0.38$ | $187.00 \pm 15.70$ | $179.00 \pm 1.86$ | $173.00 \pm 1.69$ | $286.00 \pm 7.50$ | $286.00 \pm 7.36$ | $145.00 \pm 1.13$ | $95.90 \pm 0.61$ | $161.00 \pm 3.78$ | $191.00 \pm 1.47$ | $163.00 \pm 1.10$ |
| 8 | $294.00 \pm 0.38$ | $166.00 \pm 17.30$ | $168.00 \pm 3.83$ | $151.00 \pm 3.05$ | $286.00 \pm 7.50$ | $286.00 \pm 7.36$ | $122.00 \pm 0.20$ | $89.70 \pm 0.32$ | $152.00 \pm 0.88$ | $169.00 \pm 4.85$ | $181.00 \pm 1.50$ |
| 5 | $165.00 \pm 5.93$ | $123.00 \pm 17.90$ | $134.00 \pm 10.20$ | $101.00 \pm 4.33$ | $286.00 \pm 7.50$ | $286.00 \pm 7.36$ | $53.40 \pm 0.29$ | $50.90 \pm 0.31$ | $143.00 \pm 5.02$ | $123.00 \pm 4.24$ | $133.00 \pm 5.45$ |
| 3 | $48.40 \pm 0.01$ | $85.60 \pm 15.60$ | $88.40 \pm 8.86$ | $71.40 \pm 1.34$ | $293.90 \pm 0.40$ | $293.90 \pm 0.40$ | $29.60 \pm 0.01$ | $34.60 \pm 1.36$ | $85.50 \pm 3.18$ | $43.30 \pm 2.77$ | $80.20 \pm 6.35$ |
| 1 | $0.03 \pm 0.02$ | $28.60 \pm 4.04$ | $27.70 \pm 3.07$ | $22.40 \pm 2.20$ | $293.90 \pm 0.40$ | $293.90 \pm 0.40$ | $11.60 \pm 0.01$ | $10.10 \pm 1.78$ | $29.20 \pm 1.93$ | $23.90 \pm 8.57$ | $30.00 \pm 2.60$ |
| $8 \times 10^{-1}$ | $0.04 \pm 0.04$ | $23.70 \pm 7.05$ | $25.50 \pm 6.84$ | $16.60 \pm 0.06$ | $293.90 \pm 0.40$ | $293.90 \pm 0.40$ | $8.59 \pm 0.43$ | $7.33 \pm 0.25$ | $27.50 \pm 0.30$ | $23.70 \pm 4.18$ | $25.90 \pm 0.43$ |
| $5 \times 10^{-1}$ | $0.04 \pm 0.04$ | $13.90 \pm 3.30$ | $11.50 \pm 4.65$ | $11.30 \pm 0.70$ | $293.90 \pm 0.40$ | $293.90 \pm 0.40$ | $4.99 \pm 0.01$ | $3.86 \pm 0.16$ | $14.00 \pm 0.19$ | $7.90 \pm 2.08$ | $15.40 \pm 0.36$ |
| $3 \times 10^{-1}$ | $0.09 \pm 0.07$ | $10.30 \pm 3.68$ | $9.24 \pm 0.24$ | $6.73 \pm 0.14$ | $293.90 \pm 0.40$ | $293.90 \pm 0.40$ | $2.97 \pm 0.00$ | $2.50 \pm 0.55$ | $10.20 \pm 0.38$ | $5.19 \pm 2.25$ | $11.20 \pm 0.88$ |
| $1 \times 10^{-1}$ | $0.03 \pm 0.01$ | $5.55 \pm 1.22$ | $2.98 \pm 0.39$ | $4.33 \pm 0.01$ | $293.90 \pm 0.40$ | $293.90 \pm 0.40$ | $0.97 \pm 0.00$ | $0.89 \pm 0.01$ | $4.99 \pm 0.22$ | $1.69 \pm 0.49$ | $5.55 \pm 0.00$ |
| $8 \times 10^{-2}$ | $0.04 \pm 0.00$ | $5.30 \pm 1.26$ | $2.90 \pm 0.09$ | $3.86 \pm 0.18$ | $293.90 \pm 0.40$ | $184.00 \pm 11.00$ | $0.78 \pm 0.00$ | $0.60 \pm 0.02$ | $4.17 \pm 0.00$ | $1.32 \pm 0.41$ | $4.89 \pm 0.01$ |
| $5 \times 10^{-2}$ | $0.03 \pm 0.01$ | $4.85 \pm 1.70$ | $1.91 \pm 0.04$ | $3.13 \pm 0.01$ | $210.00 \pm 7.20$ | $5.07 \pm 0.03$ | $0.49 \pm 0.00$ | $0.55 \pm 0.17$ | $3.81 \pm 0.03$ | $0.87 \pm 0.28$ | $3.85 \pm 0.03$ |
| $3 \times 10^{-2}$ | $0.06 \pm 0.04$ | $3.68 \pm 1.01$ | $1.18 \pm 0.06$ | $2.74 \pm 0.08$ | $5.59 \pm 0.00$ | $0.31 \pm 0.01$ | $0.29 \pm 0.00$ | $0.22 \pm 0.00$ | $3.30 \pm 0.00$ | $0.56 \pm 0.15$ | $3.47 \pm 0.06$ |
| $1 \times 10^{-2}$ | $1.18 \pm 0.08$ | $2.48 \pm 0.48$ | $0.25 \pm 0.07$ | $2.00 \pm 0.01$ | $0.35 \pm 0.00$ | $0.23 \pm 0.00$ | $0.10 \pm 0.00$ | $0.07 \pm 0.00$ | $2.45 \pm 0.04$ | $0.21 \pm 0.04$ | $2.55 \pm 0.01$ |
| $8 \times 10^{-3}$ | $1.36 \pm 0.10$ | $2.26 \pm 0.47$ | $0.17 \pm 0.04$ | $1.81 \pm 0.02$ | $0.25 \pm 0.00$ | $0.13 \pm 0.02$ | $0.07 \pm 0.00$ | $0.06 \pm 0.00$ | $2.34 \pm 0.02$ | $0.19 \pm 0.03$ | $2.41 \pm 0.02$ |
| $5 \times 10^{-3}$ | $1.73 \pm 0.08$ | $1.70 \pm 0.42$ | $0.17 \pm 0.00$ | $1.30 \pm 0.01$ | $0.13 \pm 0.03$ | $0.06 \pm 0.00$ | $0.05 \pm 0.00$ | $0.04 \pm 0.00$ | $2.00 \pm 0.03$ | $0.13 \pm 0.02$ | $2.03 \pm 0.02$ |
| $3 \times 10^{-3}$ | $2.12 \pm 0.05$ | $1.14 \pm 0.40$ | $0.11 \pm 0.01$ | $0.76 \pm 0.01$ | $0.09 \pm 0.00$ | $0.03 \pm 0.00$ | $0.04 \pm 0.00$ | $0.03 \pm 0.00$ | $1.48 \pm 0.02$ | $0.10 \pm 0.00$ | $1.50 \pm 0.01$ |
| $1 \times 10^{-3}$ | $3.05 \pm 0.03$ | $0.15 \pm 0.10$ | $0.03 \pm 0.00$ | $0.09 \pm 0.05$ | $0.03 \pm 0.00$ | $0.03 \pm 0.00$ | $0.03 \pm 0.00$ | $0.03 \pm 0.00$ | $0.03 \pm 0.00$ | $0.13 \pm 0.02$ | $0.10 \pm 0.07$ |
| $8 \times 10^{-4}$ | $3.34 \pm 0.02$ | $0.18 \pm 0.14$ | $0.03 \pm 0.00$ | $0.04 \pm 0.00$ | $0.03 \pm 0.00$ | $0.03 \pm 0.00$ | $0.06 \pm 0.03$ | $0.03 \pm 0.00$ | $0.03 \pm 0.00$ | $0.13 \pm 0.00$ | $0.04 \pm 0.01$ |
| $5 \times 10^{-4}$ | $4.25 \pm 0.00$ | $0.09 \pm 0.06$ | $0.03 \pm 0.00$ | $0.03 \pm 0.00$ | $0.03 \pm 0.00$ | $0.03 \pm 0.00$ | $0.05 \pm 0.03$ | $0.13 \pm 0.10$ | $0.07 \pm 0.01$ | $0.14 \pm 0.05$ | $0.03 \pm 0.01$ |
| $3 \times 10^{-4}$ | $6.02 \pm 0.04$ | $0.03 \pm 0.00$ | $0.05 \pm 0.02$ | $0.03 \pm 0.00$ | $0.03 \pm 0.00$ | $0.03 \pm 0.00$ | $0.06 \pm 0.03$ | $0.03 \pm 0.00$ | $0.05 \pm 0.03$ | $0.16 \pm 0.01$ | $0.08 \pm 0.03$ |
| $1 \times 10^{-4}$ | $13.91 \pm 0.32$ | $6.76 \pm 4.76$ | $0.84 \pm 0.07$ | $1.19 \pm 0.20$ | $0.03 \pm 0.00$ | $0.03 \pm 0.00$ | $1.38 \pm 0.11$ | $1.42 \pm 0.08$ | $1.32 \pm 0.11$ | $5.20 \pm 0.05$ | $1.57 \pm 0.02$ |

Table 14: Results for Color Transfer (Set 1).

| LR | SW | MaxSW | DSW | MaxKSW | iMSW | viMSW | oMSW | rMSW | EBSW | RPSW | EBRPSW |
|---|---|---|---|---|---|---|---|---|---|---|---|
| 100 | 271.54 ± 0.22 | 255.40 ± 2.41 | 268.04 ± 0.71 | 266.08 ± 0.17 | 250.17 ± 2.50 | 250.21 ± 2.46 | 269.68 ± 2.09 | 184.92 ± 0.00 | 269.68 ± 2.09 | 267.88 ± 0.27 | 267.89 ± 0.27 |
| 80 | 271.54 ± 0.22 | 267.88 ± 0.27 | 268.93 ± 0.93 | 265.72 ± 0.30 | 250.17 ± 2.50 | 250.21 ± 2.46 | 269.68 ± 2.09 | 183.33 ± 0.08 | 271.54 ± 0.22 | 267.84 ± 0.03 | 267.86 ± 0.03 |
| 50 | 271.54 ± 0.22 | 267.67 ± 0.25 | 270.75 ± 0.57 | 264.57 ± 0.25 | 250.17 ± 2.50 | 250.21 ± 2.46 | 269.68 ± 2.09 | 181.88 ± 0.37 | 271.54 ± 0.22 | 267.48 ± 0.22 | 267.66 ± 0.25 |
| 30 | 206.09 ± 0.32 | 266.45 ± 0.25 | 271.54 ± 0.22 | 263.23 ± 0.32 | 250.21 ± 2.46 | 250.21 ± 2.46 | 269.33 ± 1.63 | 150.88 ± 0.68 | 266.07 ± 0.27 | 265.47 ± 0.10 | 266.50 ± 0.30 |
| 10 | 267.89 ± 0.27 | 231.34 ± 0.18 | 237.76 ± 8.30 | 207.42 ± 14.88 | 250.45 ± 2.28 | 250.45 ± 2.28 | 145.92 ± 0.14 | 110.15 ± 0.05 | 258.87 ± 0.32 | 209.55 ± 8.08 | 188.90 ± 1.52 |
| 8 | 267.89 ± 0.27 | 202.37 ± 0.25 | 208.99 ± 0.19 | 170.61 ± 3.62 | 267.89 ± 0.27 | 267.89 ± 0.27 | 122.14 ± 0.03 | 70.60 ± 0.03 | 241.47 ± 0.33 | 197.62 ± 2.41 | 189.23 ± 7.49 |
| 5 | 267.89 ± 0.27 | 155.92 ± 7.23 | 160.85 ± 2.04 | 104.98 ± 7.47 | 250.96 ± 2.44 | 250.78 ± 2.66 | 63.83 ± 0.05 | 52.42 ± 0.06 | 145.10 ± 23.12 | 109.62 ± 6.34 | 138.06 ± 4.07 |
| 3 | 67.92 ± 3.09 | 81.71 ± 3.84 | 82.30 ± 16.46 | 62.80 ± 2.47 | 249.98 ± 2.76 | 249.98 ± 2.76 | 29.07 ± 0.02 | 26.25 ± 0.45 | 93.20 ± 1.46 | 66.69 ± 3.35 | 89.42 ± 2.56 |
| 1 | 4.82 ± 0.05 | 39.43 ± 1.40 | 17.67 ± 9.12 | 21.70 ± 0.29 | 259.34 ± 0.43 | 259.94 ± 0.03 | 9.43 ± 0.19 | 10.18 ± 1.32 | 30.18 ± 1.27 | 22.54 ± 3.17 | 34.46 ± 2.11 |
| $8 \times 10^{-1}$ | 0.01 ± 0.00 | 31.62 ± 2.06 | 17.77 ± 4.39 | 18.39 ± 0.09 | 267.62 ± 0.01 | 267.62 ± 0.01 | 9.74 ± 0.08 | 7.77 ± 1.12 | 26.50 ± 1.50 | 18.69 ± 0.09 | 25.70 ± 1.05 |
| $5 \times 10^{-1}$ | 0.01 ± 0.00 | 19.61 ± 0.76 | 11.12 ± 0.13 | 15.37 ± 1.67 | 267.89 ± 0.27 | 267.89 ± 0.27 | 5.38 ± 0.33 | 4.35 ± 0.32 | 17.35 ± 0.83 | 12.48 ± 0.86 | 17.65 ± 0.39 |
| $3 \times 10^{-1}$ | 0.03 ± 0.01 | 12.00 ± 0.85 | 1.31 ± 0.01 | 8.26 ± 0.09 | 267.89 ± 0.27 | 267.89 ± 0.27 | 3.31 ± 0.00 | 2.58 ± 0.29 | 10.92 ± 0.30 | 7.50 ± 0.91 | 11.34 ± 0.26 |
| $1 \times 10^{-1}$ | 0.03 ± 0.00 | 6.39 ± 0.52 | 3.40 ± 0.21 | 5.72 ± 0.00 | 267.89 ± 0.27 | 267.89 ± 0.27 | 1.06 ± 0.01 | 0.81 ± 0.04 | 6.36 ± 0.04 | 2.54 ± 0.29 | 7.46 ± 0.09 |
| $8 \times 10^{-2}$ | 0.03 ± 0.00 | 6.62 ± 0.65 | 2.65 ± 0.13 | 3.82 ± 0.02 | 267.89 ± 0.27 | 267.89 ± 0.27 | 0.95 ± 0.01 | 0.70 ± 0.01 | 5.10 ± 0.47 | 1.97 ± 0.23 | 6.87 ± 0.09 |
| $5 \times 10^{-2}$ | 0.33 ± 0.27 | 5.60 ± 0.38 | 2.12 ± 0.10 | 3.28 ± 0.08 | 250.80 ± 2.77 | 251.21 ± 2.88 | 0.59 ± 0.05 | 0.35 ± 0.00 | 5.81 ± 0.00 | 1.28 ± 0.15 | 6.03 ± 0.05 |
| $3 \times 10^{-2}$ | 0.63 ± 0.57 | 4.60 ± 0.46 | 1.18 ± 0.00 | 2.74 ± 0.00 | 3.36 ± 1.76 | 3.19 ± 1.57 | 0.30 ± 0.00 | 0.22 ± 0.01 | 5.41 ± 1.05 | 0.76 ± 0.10 | 5.51 ± 1.02 |
| $1 \times 10^{-2}$ | 2.20 ± 0.51 | 2.71 ± 0.12 | 0.25 ± 0.00 | 2.00 ± 0.00 | 0.34 ± 0.01 | 0.33 ± 0.01 | 0.10 ± 0.00 | 0.08 ± 0.01 | 4.22 ± 0.04 | 0.22 ± 0.02 | 4.25 ± 0.02 |
| $8 \times 10^{-3}$ | 2.57 ± 0.61 | 2.31 ± 0.02 | 0.15 ± 0.01 | 1.83 ± 0.01 | 0.24 ± 0.00 | 0.24 ± 0.00 | 0.08 ± 0.00 | 0.07 ± 0.01 | 3.96 ± 0.01 | 0.21 ± 0.02 | 4.00 ± 0.02 |
| $5 \times 10^{-3}$ | 3.00 ± 0.63 | 1.57 ± 0.06 | 0.10 ± 0.03 | 1.32 ± 0.01 | 0.13 ± 0.00 | 0.13 ± 0.00 | 0.05 ± 0.00 | 0.04 ± 0.00 | 3.28 ± 0.64 | 0.17 ± 0.02 | 3.30 ± 0.64 |
| $3 \times 10^{-3}$ | 4.07 ± 0.98 | 0.92 ± 0.11 | 0.11 ± 0.00 | 0.76 ± 0.00 | 0.09 ± 0.00 | 0.07 ± 0.01 | 0.04 ± 0.00 | 0.03 ± 0.00 | 1.49 ± 0.01 | 0.10 ± 0.00 | 1.50 ± 0.00 |
| $1 \times 10^{-3}$ | 8.48 ± 2.72 | 0.15 ± 0.00 | 0.04 ± 0.01 | 0.10 ± 0.06 | 0.03 ± 0.00 | 0.03 ± 0.00 | 0.03 ± 0.00 | 0.03 ± 0.00 | 0.03 ± 0.00 | 0.13 ± 0.00 | 0.07 ± 0.02 |
| $8 \times 10^{-4}$ | 9.10 ± 2.88 | 0.12 ± 0.03 | 0.04 ± 0.00 | 0.04 ± 0.00 | 0.03 ± 0.00 | 0.03 ± 0.00 | 0.06 ± 0.00 | 0.03 ± 0.00 | 0.03 ± 0.00 | 0.16 ± 0.02 | 0.04 ± 0.00 |
| $5 \times 10^{-4}$ | 11.10 ± 3.42 | 0.06 ± 0.01 | 0.04 ± 0.01 | 0.03 ± 0.00 | 0.03 ± 0.00 | 0.03 ± 0.00 | 0.05 ± 0.00 | 0.07 ± 0.03 | 0.04 ± 0.02 | 0.15 ± 0.00 | 0.03 ± 0.00 |
| $3 \times 10^{-4}$ | 13.97 ± 3.97 | 0.03 ± 0.00 | 0.04 ± 0.00 | 0.03 ± 0.00 | 0.03 ± 0.00 | 0.03 ± 0.00 | 0.05 ± 0.01 | 0.03 ± 0.00 | 0.04 ± 0.01 | 0.15 ± 0.00 | 0.06 ± 0.01 |
| $1 \times 10^{-4}$ | 11.85 ± 0.63 | 4.04 ± 1.36 | 0.86 ± 0.01 | 1.25 ± 0.03 | 0.03 ± 0.00 | 0.03 ± 0.00 | 1.15 ± 0.12 | 1.16 ± 0.13 | 1.32 ± 0.00 | 4.59 ± 0.31 | 1.49 ± 0.04 |

Table 15: Results for Color Transfer (Set 2).

| LR | SW | MaxSW | DSW | MaxKSW | iMSW | viMSW | oMSW | rMSW | EBSW | RPSW | EBRPSW |
|---|---|---|---|---|---|---|---|---|---|---|---|
| 100 | 320.55 ± 0.09 | 320.55 ± 0.09 | 279.16 ± 0.05 | 308.63 ± 0.18 | 320.55 ± 0.09 | 320.55 ± 0.09 | 320.55 ± 0.09 | 157.93 ± 0.04 | 320.55 ± 0.09 | 320.55 ± 0.09 | 320.55 ± 0.09 |
| 80 | 320.55 ± 0.09 | 320.55 ± 0.09 | 279.16 ± 0.05 | 307.70 ± 0.20 | 320.55 ± 0.09 | 320.55 ± 0.09 | 320.55 ± 0.09 | 168.12 ± 1.25 | 320.55 ± 0.09 | 320.55 ± 0.09 | 320.55 ± 0.09 |
| 50 | 320.55 ± 0.09 | 320.55 ± 0.09 | 279.16 ± 0.05 | 304.23 ± 0.21 | 320.55 ± 0.09 | 320.55 ± 0.09 | 320.55 ± 0.09 | 124.61 ± 0.79 | 320.55 ± 0.09 | 320.55 ± 0.09 | 320.55 ± 0.09 |
| 30 | 320.55 ± 0.09 | 320.55 ± 0.09 | 279.16 ± 0.05 | 298.63 ± 0.38 | 320.55 ± 0.09 | 320.55 ± 0.09 | 320.45 ± 0.07 | 123.85 ± 10.63 | 149.82 ± 0.07 | 320.52 ± 0.06 | 320.57 ± 0.05 |
| 10 | 320.55 ± 0.09 | 286.84 ± 0.29 | 279.14 ± 0.03 | 202.30 ± 1.04 | 320.17 ± 0.03 | 320.16 ± 0.03 | 183.41 ± 0.22 | 62.75 ± 0.48 | 157.47 ± 28.22 | 266.66 ± 2.21 | 205.07 ± 78.24 |
| 8 | 320.52 ± 0.07 | 251.89 ± 0.26 | 210.92 ± 6.75 | 165.78 ± 0.11 | 311.04 ± 0.20 | 311.02 ± 0.19 | 143.62 ± 0.14 | 91.55 ± 0.01 | 171.06 ± 18.71 | 222.87 ± 15.73 | 165.19 ± 37.58 |
| 5 | 254.50 ± 0.29 | 175.13 ± 4.67 | 161.67 ± 0.60 | 95.66 ± 15.39 | 219.26 ± 19.68 | 232.89 ± 0.29 | 72.48 ± 0.04 | 30.00 ± 1.59 | 123.75 ± 58.87 | 111.06 ± 3.36 | 124.90 ± 7.31 |
| 3 | 59.33 ± 0.07 | 108.52 ± 1.52 | 107.15 ± 8.57 | 57.36 ± 3.19 | 320.55 ± 0.09 | 320.55 ± 0.09 | 34.35 ± 0.22 | 29.95 ± 0.29 | 87.52 ± 10.23 | 66.06 ± 1.43 | 107.12 ± 1.10 |
| 1 | 0.00 ± 0.00 | 35.58 ± 1.42 | 39.61 ± 2.14 | 21.60 ± 0.55 | 320.55 ± 0.09 | 320.55 ± 0.09 | 9.70 ± 0.01 | 10.02 ± 0.49 | 34.00 ± 0.45 | 19.71 ± 3.98 | 32.68 ± 4.24 |
| $8 \times 10^{-1}$ | 0.00 ± 0.00 | 28.51 ± 1.06 | 17.42 ± 0.97 | 17.33 ± 0.41 | 320.55 ± 0.09 | 320.55 ± 0.09 | 7.68 ± 0.01 | 7.67 ± 0.39 | 27.35 ± 1.00 | 18.13 ± 2.59 | 27.86 ± 1.00 |
| $5 \times 10^{-1}$ | 0.00 ± 0.00 | 19.69 ± 1.77 | 12.75 ± 4.32 | 11.51 ± 0.11 | 320.55 ± 0.09 | 320.55 ± 0.09 | 4.80 ± 0.01 | 5.08 ± 0.89 | 16.78 ± 0.14 | 10.86 ± 0.59 | 18.69 ± 1.52 |
| $3 \times 10^{-1}$ | 0.20 ± 0.25 | 12.93 ± 1.49 | 7.88 ± 1.00 | 8.03 ± 0.05 | 320.55 ± 0.09 | 320.55 ± 0.09 | 3.38 ± 0.01 | 2.75 ± 0.22 | 11.10 ± 0.32 | 6.10 ± 0.27 | 11.51 ± 0.58 |
| $1 \times 10^{-1}$ | 0.03 ± 0.00 | 6.24 ± 0.71 | 3.97 ± 0.02 | 5.10 ± 0.24 | 320.15 ± 0.03 | 320.16 ± 0.03 | 0.95 ± 0.00 | 0.79 ± 0.01 | 6.17 ± 0.16 | 1.94 ± 0.23 | 6.96 ± 0.52 |
| $8 \times 10^{-2}$ | 0.20 ± 0.24 | 5.93 ± 0.44 | 2.75 ± 0.40 | 4.72 ± 0.06 | 311.04 ± 0.20 | 311.02 ± 0.19 | 0.77 ± 0.00 | 0.73 ± 0.01 | 5.51 ± 0.21 | 1.59 ± 0.29 | 6.35 ± 0.41 |
| $5 \times 10^{-2}$ | 0.17 ± 0.24 | 4.33 ± 0.74 | 2.02 ± 0.07 | 4.47 ± 0.08 | 219.26 ± 19.68 | 232.89 ± 0.29 | 0.48 ± 0.00 | 0.49 ± 0.06 | 5.03 ± 0.16 | 0.97 ± 0.14 | 5.30 ± 0.10 |
| $3 \times 10^{-2}$ | 0.95 ± 0.01 | 3.68 ± 0.00 | 1.17 ± 0.01 | 4.06 ± 0.08 | 5.99 ± 0.56 | 5.49 ± 0.59 | 0.29 ± 0.00 | 0.24 ± 0.02 | 4.56 ± 0.06 | 0.58 ± 0.05 | 4.78 ± 0.08 |
| $1 \times 10^{-2}$ | 2.16 ± 0.04 | 2.31 ± 0.12 | 0.28 ± 0.05 | 2.93 ± 0.01 | 0.34 ± 0.01 | 0.32 ± 0.01 | 0.10 ± 0.00 | 0.07 ± 0.00 | 3.51 ± 0.05 | 0.28 ± 0.08 | 3.59 ± 0.01 |
| $8 \times 10^{-3}$ | 2.37 ± 0.07 | 2.09 ± 0.12 | 0.22 ± 0.07 | 2.64 ± 0.01 | 0.24 ± 0.01 | 0.23 ± 0.00 | 0.09 ± 0.02 | 0.07 ± 0.01 | 3.32 ± 0.02 | 0.23 ± 0.06 | 3.42 ± 0.01 |
| $5 \times 10^{-3}$ | 2.99 ± 0.02 | 1.61 ± 0.06 | 0.18 ± 0.01 | 1.51 ± 0.01 | 0.13 ± 0.00 | 0.13 ± 0.00 | 0.05 ± 0.00 | 0.04 ± 0.00 | 2.41 ± 0.02 | 0.16 ± 0.05 | 2.44 ± 0.01 |
| $3 \times 10^{-3}$ | 3.24 ± 0.03 | 1.15 ± 0.01 | 0.12 ± 0.01 | 0.93 ± 0.12 | 0.09 ± 0.00 | 0.06 ± 0.00 | 0.04 ± 0.00 | 0.04 ± 0.01 | 1.76 ± 0.01 | 0.10 ± 0.00 | 1.79 ± 0.01 |
| $1 \times 10^{-3}$ | 3.48 ± 0.17 | 0.15 ± 0.00 | 0.05 ± 0.01 | 0.66 ± 0.37 | 0.03 ± 0.00 | 0.10 ± 0.09 | 0.03 ± 0.00 | 0.03 ± 0.00 | 0.03 ± 0.00 | 0.15 ± 0.01 | 0.11 ± 0.11 |
| $8 \times 10^{-4}$ | 3.84 ± 0.35 | 0.12 ± 0.05 | 0.03 ± 0.00 | 0.04 ± 0.00 | 0.08 ± 0.08 | 0.03 ± 0.00 | 0.03 ± 0.00 | 0.03 ± 0.00 | 0.05 ± 0.02 | 0.09 ± 0.05 | 0.05 ± 0.03 |
| $5 \times 10^{-4}$ | 4.93 ± 0.48 | 0.06 ± 0.02 | 0.03 ± 0.00 | 0.03 ± 0.00 | 0.03 ± 0.00 | 0.03 ± 0.00 | 0.05 ± 0.00 | 0.08 ± 0.07 | 0.03 ± 0.00 | 0.11 ± 0.02 | 0.03 ± 0.00 |
| $3 \times 10^{-4}$ | 6.37 ± 0.25 | 0.03 ± 0.00 | 0.08 ± 0.02 | 0.03 ± 0.00 | 0.03 ± 0.00 | 0.03 ± 0.00 | 0.05 ± 0.01 | 0.03 ± 0.00 | 0.07 ± 0.02 | 0.15 ± 0.01 | 0.05 ± 0.02 |
| $1 \times 10^{-4}$ | 13.61 ± 0.21 | 5.55 ± 0.86 | 0.89 ± 0.03 | 6.88 ± 0.15 | 0.70 ± 0.09 | 0.74 ± 0.14 | 11.42 ± 0.11 | 12.17 ± 0.14 | 3.64 ± 0.15 | 7.50 ± 0.16 | 3.68 ± 0.01 |

Table 16: Results for Color Transfer (Set 3).

| LR | SW | MaxSW | DSW | MaxKSW | iMSW | viMSW | oMSW | rMSW | EBSW | RPSW | EBRPSW |
|---|---|---|---|---|---|---|---|---|---|---|---|
| $1$ | $14.17_{\pm0.02}$ | - | - | - | - | - | - | - | - | - | - |
| $8 \times 10^{-1}$ | $14.16_{\pm0.01}$ | - | - | - | - | - | - | - | - | - | - |
| $5 \times 10^{-1}$ | $14.15_{\pm0.02}$ | - | - | - | - | - | - | - | - | - | - |
| $3 \times 10^{-1}$ | $14.16_{\pm0.00}$ | - | - | - | - | - | - | - | - | $17.78_{\pm0.18}$ | - |
| $1 \times 10^{-1}$ | $14.22_{\pm0.01}$ | - | - | $14.50_{\pm0.03}$ | - | - | $14.13_{\pm0.02}$ | $14.17_{\pm0.02}$ | - | $14.14_{\pm0.00}$ | - |
| $8 \times 10^{-2}$ | $14.26_{\pm0.01}$ | - | $14.78_{\pm0.02}$ | $14.29_{\pm0.01}$ | - | - | $14.12_{\pm0.01}$ | $14.16_{\pm0.01}$ | - | $14.15_{\pm0.02}$ | - |
| $5 \times 10^{-2}$ | $14.33_{\pm0.01}$ | $14.54_{\pm0.02}$ | $14.46_{\pm0.03}$ | $14.20_{\pm0.01}$ | - | - | $14.19_{\pm0.00}$ | $14.18_{\pm0.02}$ | - | $14.18_{\pm0.02}$ | - |
| $3 \times 10^{-2}$ | $14.37_{\pm0.01}$ | $14.26_{\pm0.02}$ | $14.25_{\pm0.02}$ | $14.27_{\pm0.01}$ | - | - | $14.20_{\pm0.01}$ | $14.19_{\pm0.02}$ | - | $14.18_{\pm0.01}$ | - |
| $1 \times 10^{-2}$ | $14.50_{\pm0.01}$ | $14.06_{\pm0.02}$ | $14.12_{\pm0.02}$ | $14.29_{\pm0.02}$ | - | - | $14.17_{\pm0.01}$ | $14.16_{\pm0.01}$ | $14.71_{\pm0.02}$ | $14.15_{\pm0.02}$ | $14.69_{\pm0.02}$ |
| $8 \times 10^{-3}$ | $14.54_{\pm0.01}$ | $14.02_{\pm0.02}$ | $14.11_{\pm0.02}$ | $14.27_{\pm0.01}$ | - | - | $14.16_{\pm0.02}$ | $14.16_{\pm0.01}$ | $14.72_{\pm0.04}$ | $14.16_{\pm0.02}$ | $14.71_{\pm0.02}$ |
| $5 \times 10^{-3}$ | $14.67_{\pm0.00}$ | $14.01_{\pm0.02}$ | $14.11_{\pm0.02}$ | $14.26_{\pm0.03}$ | - | - | $14.16_{\pm0.01}$ | $14.18_{\pm0.01}$ | $14.72_{\pm0.02}$ | $14.17_{\pm0.02}$ | $14.70_{\pm0.04}$ |
| $3 \times 10^{-3}$ | $14.84_{\pm0.01}$ | $14.01_{\pm0.02}$ | $14.16_{\pm0.02}$ | $14.27_{\pm0.02}$ | $14.06_{\pm0.01}$ | $14.09_{\pm0.01}$ | $14.19_{\pm0.01}$ | $14.19_{\pm0.01}$ | $14.77_{\pm0.03}$ | $14.20_{\pm0.01}$ | $14.77_{\pm0.05}$ |
| $1 \times 10^{-3}$ | $15.36_{\pm0.01}$ | $14.04_{\pm0.02}$ | $14.17_{\pm0.02}$ | $14.42_{\pm0.01}$ | $14.12_{\pm0.01}$ | $14.26_{\pm0.00}$ | $14.34_{\pm0.01}$ | $14.37_{\pm0.01}$ | $14.72_{\pm0.02}$ | $14.36_{\pm0.00}$ | $14.73_{\pm0.05}$ |
| $8 \times 10^{-4}$ | $15.47_{\pm0.02}$ | $14.08_{\pm0.01}$ | $14.17_{\pm0.02}$ | $14.47_{\pm0.01}$ | $14.14_{\pm0.02}$ | $14.31_{\pm0.01}$ | $14.35_{\pm0.02}$ | $14.38_{\pm0.01}$ | $14.74_{\pm0.02}$ | $14.38_{\pm0.01}$ | $14.76_{\pm0.02}$ |
| $5 \times 10^{-4}$ | $15.73_{\pm0.02}$ | $14.12_{\pm0.01}$ | $14.22_{\pm0.02}$ | $14.48_{\pm0.01}$ | $14.26_{\pm0.01}$ | $14.34_{\pm0.01}$ | $14.39_{\pm0.01}$ | $14.40_{\pm0.01}$ | $14.85_{\pm0.03}$ | $14.41_{\pm0.01}$ | $14.87_{\pm0.03}$ |
| $3 \times 10^{-4}$ | $16.59_{\pm0.04}$ | $14.75_{\pm0.02}$ | $14.91_{\pm0.04}$ | $14.94_{\pm0.02}$ | $14.81_{\pm0.02}$ | $14.81_{\pm0.02}$ | $14.88_{\pm0.03}$ | $14.88_{\pm0.01}$ | $14.97_{\pm0.02}$ | $14.48_{\pm0.01}$ | $14.99_{\pm0.04}$ |
| $1 \times 10^{-4}$ | $17.36_{\pm0.04}$ | $14.41_{\pm0.02}$ | $14.52_{\pm0.01}$ | $14.70_{\pm0.01}$ | $14.40_{\pm0.01}$ | $14.50_{\pm0.01}$ | $14.70_{\pm0.01}$ | $14.69_{\pm0.00}$ | $15.24_{\pm0.01}$ | $14.73_{\pm0.02}$ | $15.23_{\pm0.02}$ |
| $8 \times 10^{-5}$ | $17.71_{\pm0.11}$ | $14.41_{\pm0.01}$ | $14.53_{\pm0.01}$ | $14.79_{\pm0.01}$ | $14.42_{\pm0.01}$ | $14.53_{\pm0.01}$ | $14.77_{\pm0.01}$ | $14.77_{\pm0.01}$ | $15.15_{\pm0.02}$ | $14.81_{\pm0.01}$ | $15.15_{\pm0.02}$ |
| $5 \times 10^{-5}$ | $18.11_{\pm0.04}$ | $14.45_{\pm0.01}$ | $14.55_{\pm0.01}$ | $14.97_{\pm0.02}$ | $14.47_{\pm0.01}$ | $14.61_{\pm0.01}$ | $14.97_{\pm0.01}$ | $14.96_{\pm0.01}$ | $15.00_{\pm0.01}$ | $14.99_{\pm0.01}$ | $15.01_{\pm0.01}$ |
| $3 \times 10^{-5}$ | $18.49_{\pm0.07}$ | $14.50_{\pm0.01}$ | $14.61_{\pm0.01}$ | $15.22_{\pm0.01}$ | $14.55_{\pm0.01}$ | $14.72_{\pm0.01}$ | $15.20_{\pm0.00}$ | $15.23_{\pm0.01}$ | $14.96_{\pm0.01}$ | $15.57_{\pm0.01}$ | $14.95_{\pm0.02}$ |
| $1 \times 10^{-5}$ | $18.84_{\pm0.06}$ | $14.66_{\pm0.01}$ | $14.75_{\pm0.01}$ | $15.80_{\pm0.00}$ | $14.77_{\pm0.01}$ | $15.09_{\pm0.01}$ | $15.81_{\pm0.02}$ | $15.81_{\pm0.03}$ | $15.39_{\pm0.01}$ | $15.81_{\pm0.02}$ | $15.39_{\pm0.02}$ |
| $8 \times 10^{-6}$ | $18.89_{\pm0.04}$ | $14.70_{\pm0.01}$ | $14.80_{\pm0.01}$ | $15.94_{\pm0.02}$ | $14.84_{\pm0.01}$ | $15.20_{\pm0.01}$ | $15.95_{\pm0.02}$ | $15.95_{\pm0.02}$ | $15.50_{\pm0.02}$ | $15.93_{\pm0.02}$ | $15.50_{\pm0.01}$ |
| $5 \times 10^{-6}$ | $19.08_{\pm0.05}$ | $14.84_{\pm0.01}$ | $14.95_{\pm0.02}$ | $16.34_{\pm0.02}$ | $14.99_{\pm0.01}$ | $15.44_{\pm0.01}$ | $16.35_{\pm0.01}$ | $16.34_{\pm0.04}$ | $15.81_{\pm0.02}$ | $16.30_{\pm0.02}$ | $15.79_{\pm0.01}$ |
| $3 \times 10^{-6}$ | $19.10_{\pm0.08}$ | $15.03_{\pm0.01}$ | $15.20_{\pm0.02}$ | $16.93_{\pm0.05}$ | $15.26_{\pm0.02}$ | $15.77_{\pm0.01}$ | $16.95_{\pm0.03}$ | $16.94_{\pm0.04}$ | $16.25_{\pm0.02}$ | $16.81_{\pm0.03}$ | $16.23_{\pm0.05}$ |
| $1 \times 10^{-6}$ | $19.05_{\pm0.06}$ | $15.76_{\pm0.01}$ | $16.03_{\pm0.01}$ | $18.18_{\pm0.07}$ | $15.96_{\pm0.04}$ | $16.49_{\pm0.03}$ | $18.18_{\pm0.07}$ | $18.15_{\pm0.04}$ | $17.30_{\pm0.03}$ | $18.10_{\pm0.06}$ | $17.30_{\pm0.05}$ |

Table 17: Numerical results for the M2F task.

| LR | SW | MaxSW | DSW | MaxKSW | iMSW | viMSW | oMSW | rMSW | EBSW | RPSW | EBRPSW |
|---|---|---|---|---|---|---|---|---|---|---|---|
| $1$ | $14.58_{\pm0.03}$ | - | - | - | - | - | - | - | - | - | - |
| $8 \times 10^{-1}$ | $14.61_{\pm0.03}$ | - | - | - | - | - | - | - | - | - | - |
| $5 \times 10^{-1}$ | $14.62_{\pm0.03}$ | - | - | - | - | - | - | - | - | - | - |
| $3 \times 10^{-1}$ | $14.62_{\pm0.02}$ | - | - | - | - | - | - | - | - | $14.68_{\pm0.13}$ | - |
| $1 \times 10^{-1}$ | $14.70_{\pm0.02}$ | - | - | $14.98_{\pm0.02}$ | - | - | $14.62_{\pm0.02}$ | $14.63_{\pm0.01}$ | - | $14.65_{\pm0.03}$ | - |
| $8 \times 10^{-2}$ | $14.73_{\pm0.02}$ | $16.48_{\pm0.00}$ | $15.35_{\pm0.07}$ | $14.81_{\pm0.02}$ | - | - | $14.62_{\pm0.02}$ | $14.63_{\pm0.01}$ | - | $14.64_{\pm0.01}$ | - |
| $5 \times 10^{-2}$ | $14.80_{\pm0.02}$ | $15.17_{\pm0.08}$ | $14.95_{\pm0.02}$ | $14.65_{\pm0.02}$ | - | - | $14.63_{\pm0.01}$ | $14.67_{\pm0.03}$ | - | $14.65_{\pm0.03}$ | - |
| $3 \times 10^{-2}$ | $14.80_{\pm0.03}$ | $14.63_{\pm0.04}$ | $15.44_{\pm0.00}$ | $15.71_{\pm0.00}$ | - | - | $14.65_{\pm0.02}$ | $14.62_{\pm0.02}$ | - | $14.62_{\pm0.02}$ | - |
| $1 \times 10^{-2}$ | $14.95_{\pm0.02}$ | $14.60_{\pm0.02}$ | $14.63_{\pm0.04}$ | $14.69_{\pm0.03}$ | - | - | $14.58_{\pm0.03}$ | $14.62_{\pm0.02}$ | $15.18_{\pm0.04}$ | $14.60_{\pm0.01}$ | $15.16_{\pm0.03}$ |
| $8 \times 10^{-3}$ | $15.01_{\pm0.03}$ | $14.61_{\pm0.02}$ | $14.62_{\pm0.03}$ | $14.70_{\pm0.02}$ | - | - | $14.61_{\pm0.01}$ | $14.60_{\pm0.02}$ | $15.15_{\pm0.03}$ | $14.60_{\pm0.02}$ | $15.15_{\pm0.05}$ |
| $5 \times 10^{-3}$ | $15.09_{\pm0.03}$ | $14.53_{\pm0.03}$ | $14.60_{\pm0.04}$ | $14.69_{\pm0.05}$ | - | - | $14.61_{\pm0.03}$ | $14.63_{\pm0.02}$ | $15.11_{\pm0.04}$ | $14.64_{\pm0.03}$ | $15.16_{\pm0.04}$ |
| $3 \times 10^{-3}$ | $15.28_{\pm0.01}$ | $14.52_{\pm0.02}$ | $14.61_{\pm0.02}$ | $14.70_{\pm0.03}$ | $14.62_{\pm0.04}$ | $14.57_{\pm0.01}$ | $14.65_{\pm0.04}$ | $14.65_{\pm0.04}$ | $15.16_{\pm0.01}$ | $14.67_{\pm0.02}$ | $15.15_{\pm0.05}$ |
| $1 \times 10^{-3}$ | $15.78_{\pm0.01}$ | $14.57_{\pm0.03}$ | $14.65_{\pm0.01}$ | $14.89_{\pm0.01}$ | $14.59_{\pm0.01}$ | $14.72_{\pm0.03}$ | $14.80_{\pm0.02}$ | $14.83_{\pm0.03}$ | $15.13_{\pm0.01}$ | $14.78_{\pm0.02}$ | $15.10_{\pm0.05}$ |
| $8 \times 10^{-4}$ | $15.88_{\pm0.03}$ | $14.59_{\pm0.03}$ | $14.68_{\pm0.02}$ | $14.92_{\pm0.02}$ | $14.64_{\pm0.02}$ | $14.76_{\pm0.01}$ | $14.78_{\pm0.02}$ | $14.81_{\pm0.02}$ | $15.09_{\pm0.04}$ | $14.81_{\pm0.02}$ | $15.12_{\pm0.02}$ |
| $5 \times 10^{-4}$ | $16.17_{\pm0.03}$ | $14.62_{\pm0.02}$ | $14.74_{\pm0.03}$ | $14.92_{\pm0.02}$ | $14.75_{\pm0.01}$ | $14.77_{\pm0.03}$ | $14.82_{\pm0.02}$ | $14.85_{\pm0.02}$ | $15.26_{\pm0.06}$ | $14.85_{\pm0.02}$ | $15.25_{\pm0.02}$ |
| $3 \times 10^{-4}$ | $16.59_{\pm0.04}$ | $14.75_{\pm0.02}$ | $14.91_{\pm0.04}$ | $14.94_{\pm0.02}$ | $14.81_{\pm0.02}$ | $14.81_{\pm0.02}$ | $14.88_{\pm0.03}$ | $14.88_{\pm0.01}$ | $15.50_{\pm0.04}$ | $14.94_{\pm0.03}$ | $15.49_{\pm0.03}$ |
| $1 \times 10^{-4}$ | $18.05_{\pm0.05}$ | $14.89_{\pm0.04}$ | $14.96_{\pm0.02}$ | $15.16_{\pm0.02}$ | $14.81_{\pm0.02}$ | $14.97_{\pm0.02}$ | $15.14_{\pm0.04}$ | $15.13_{\pm0.02}$ | $15.63_{\pm0.04}$ | $15.17_{\pm0.02}$ | $15.66_{\pm0.02}$ |
| $8 \times 10^{-5}$ | $18.26_{\pm0.09}$ | $14.88_{\pm0.03}$ | $14.96_{\pm0.03}$ | $15.20_{\pm0.03}$ | $14.87_{\pm0.01}$ | $14.96_{\pm0.03}$ | $15.19_{\pm0.03}$ | $15.19_{\pm0.02}$ | $15.56_{\pm0.04}$ | $15.24_{\pm0.01}$ | $15.54_{\pm0.03}$ |
| $5 \times 10^{-5}$ | $18.92_{\pm0.06}$ | $14.92_{\pm0.04}$ | $15.03_{\pm0.04}$ | $15.38_{\pm0.03}$ | $14.94_{\pm0.02}$ | $15.05_{\pm0.04}$ | $15.37_{\pm0.03}$ | $15.38_{\pm0.04}$ | $15.43_{\pm0.01}$ | $15.42_{\pm0.02}$ | $15.41_{\pm0.05}$ |
| $3 \times 10^{-5}$ | $19.21_{\pm0.05}$ | $14.95_{\pm0.02}$ | $15.07_{\pm0.01}$ | $15.61_{\pm0.03}$ | $15.00_{\pm0.02}$ | $15.14_{\pm0.01}$ | $15.60_{\pm0.01}$ | $15.59_{\pm0.03}$ | $15.35_{\pm0.01}$ | $15.64_{\pm0.01}$ | $15.39_{\pm0.02}$ |
| $1 \times 10^{-5}$ | $19.64_{\pm0.04}$ | $15.12_{\pm0.01}$ | $15.23_{\pm0.02}$ | $16.21_{\pm0.03}$ | $15.22_{\pm0.04}$ | $15.55_{\pm0.03}$ | $16.23_{\pm0.03}$ | $16.22_{\pm0.03}$ | $15.72_{\pm0.04}$ | $16.22_{\pm0.02}$ | $15.70_{\pm0.04}$ |
| $8 \times 10^{-6}$ | $19.62_{\pm0.05}$ | $15.19_{\pm0.02}$ | $15.27_{\pm0.02}$ | $16.43_{\pm0.02}$ | $15.28_{\pm0.01}$ | $15.61_{\pm0.01}$ | $16.42_{\pm0.03}$ | $16.39_{\pm0.03}$ | $15.91_{\pm0.05}$ | $15.94_{\pm0.02}$ | $15.91_{\pm0.05}$ |
| $5 \times 10^{-6}$ | $19.79_{\pm0.06}$ | $15.32_{\pm0.01}$ | $15.38_{\pm0.02}$ | $16.86_{\pm0.02}$ | $15.47_{\pm0.03}$ | $15.87_{\pm0.03}$ | $16.85_{\pm0.02}$ | $16.85_{\pm0.05}$ | $16.25_{\pm0.03}$ | $16.30_{\pm0.02}$ | $16.26_{\pm0.04}$ |
| $3 \times 10^{-6}$ | $19.85_{\pm0.13}$ | $15.56_{\pm0.02}$ | $15.65_{\pm0.02}$ | $17.57_{\pm0.06}$ | $15.70_{\pm0.02}$ | $16.15_{\pm0.02}$ | $17.60_{\pm0.09}$ | $17.53_{\pm0.07}$ | $16.72_{\pm0.03}$ | $16.81_{\pm0.02}$ | $16.74_{\pm0.04}$ |
| $1 \times 10^{-6}$ | $19.84_{\pm0.10}$ | $16.23_{\pm0.02}$ | $16.46_{\pm0.02}$ | $18.95_{\pm0.06}$ | $16.39_{\pm0.05}$ | $16.93_{\pm0.05}$ | $18.93_{\pm0.08}$ | $18.88_{\pm0.06}$ | $17.92_{\pm0.07}$ | $18.05_{\pm0.09}$ | $18.02_{\pm0.04}$ |

Table 18: Numerical results for the A2C translation task.

