# OpenReview forum: "Understanding Learning with Sliced-Wasserstein Requires Re-thinking Informative Slices"
_ICLR.cc/2025/Conference — Submitted to ICLR 2025_

### Official Review · Reviewer_TZmm · 2024-11-03

**Soundness:** 3
**Presentation:** 3
**Contribution:** 3
**Rating:** 5
**Confidence:** 4

**Summary:**

This paper proposes a simplified approach to the Sliced-Wasserstein Distance (SWD) by rescaling the Wasserstein distance for each one-dimensional slice, ensuring that all slices carry the same informativeness. This method demonstrates strong performance in tasks such as image generation and color transfer, rivaling or even surpassing more complex SWD variants. The core innovation lies in using a global scaling factor to balance the informativeness of different slices, thereby avoiding the complexity of directly modifying slice distributions. Overall, the idea is innovative, and the experimental results are convincing, showcasing the potential of the classic SWD with suitable parameter settings.

**Strengths:**

1. Innovation: The rescaling method proposed in this paper offers a new approach for high-dimensional SWD applications. By using a global scaling factor to address the issue of selecting informative slices, it effectively simplifies the complex process of slice selection.
2. Theoretical support: Under the assumption of an effective subspace, this paper provides a theoretical derivation of the rescaling strategy, offering mathematical support that explains the rationale behind this method.
3. Experimental performance: In various machine learning tasks, the rescaled SWD method performs excellently, demonstrating that the performance gap between classic SWD and its complex variants can be bridged through appropriate hyperparameter tuning.

**Weaknesses:**

1. Limitation of assumption: The theoretical analysis of this method relies on the assumption that data is distributed within a low-dimensional subspace. However, this assumption may not hold in certain high-dimensional data, particularly in datasets with complex or high-dimensional structures, such as text data or more intricate time series data, where the method's effectiveness remains unclear. Therefore, its applicability may be limited to datasets that satisfy this assumption.
2. Single metric of informativeness: The proposed global scaling factor only considers the informativeness of the slices in terms of their projection onto the effective subspace. However, different tasks may involve slices with more complex characteristics, and this single metric may not fully capture the contributions of all slices. Incorporating task-specific measures of informativeness could potentially enhance the method’s applicability and robustness.
3. Limited breadth of experimental validation: Although the paper validates the method’s effectiveness on multiple visual datasets, the types of experimental data are relatively narrow, lacking validation on other types of data, such as text or audio. Additionally, the paper does not deeply analyze performance on extremely high-dimensional data, which is a significant motivation behind designing SWD variants.
4. Limited generalization: The method performs well on classic machine learning tasks, but its applicability in more complex deep generative models, such as large-scale language models, remains unclear. Further research is needed to determine whether the method maintains its efficiency and stability on complex data distributions.
5. Sensitivity to hyperparameters in experiments: The proposed scaling method relies heavily on the choice of learning rate. Although the authors suggest that the learning rate can be adjusted as a hyperparameter, the specific impact on model convergence and adaptability lacks detailed analysis. Questions such as whether the results are limited to certain learning rate ranges and whether reliance on learning rate could lead to instability in practice require further exploration.

**Questions:**

This paper provides new insights into optimizing Sliced-Wasserstein Distance, with a simplified method that is theoretically appealing and performs well across several tasks. However, due to limitations in assumptions, narrow experimental validation, and sensitivity to hyperparameters, the applicability of this method may be constrained. To enhance the comprehensiveness and generalizability of the work, it is recommended to validate the method on more complex tasks and datasets and further refine the theoretical analysis.

Suggested improvements based on the Weaknesses section:
1. Strengthen discussion and validation of assumptions: Further discuss the practicality and limitations of the effective subspace assumption in real-world applications, or explore ways to expand the method’s applicability without this assumption.
2. Expand experiments to diverse data types: Validate the method on a wider range of datasets, such as text and audio data, to verify its broad applicability. Greater diversity in experiments would enhance the generalizability and persuasiveness of the results.
3. More in-depth hyperparameter sensitivity analysis: Further analyze the effect of learning rate on the method's convergence. Testing across different learning rate ranges could ensure the robustness and stability of the method and clarify if it requires strict hyperparameter tuning.
4. Explore more flexible informativeness metrics: To adapt to a wider range of tasks, consider expanding the definition of the global scaling factor to incorporate more detailed task-specific information or introduce an adaptive informativeness metric.

---

### Official Review · Reviewer_3TbE · 2024-11-04

**Soundness:** 3
**Presentation:** 2
**Contribution:** 3
**Rating:** 6
**Confidence:** 3

**Summary:**

This manuscript presents a novel solution of using sliced Wasserstein distance in general learning tasks. It has solid theoretical and empirical results to support their claim.

**Strengths:**

1. There are a lot of derivations in the manuscript, provides a strong theoretical background for the discuss topic.

2. The ideas of 1D Wasserstein distance is quite novel.

3. Detailed experiments were conducted on some of the learning tasks.

**Weaknesses:**

1. The experiment seems not sufficient to prove the claim "Sliced-Wasserstein is all you need for common learning tasks". Some common learning tasks like image classification and object detection is not tested.

2. Larger dataset for experiment is needed. Especially, toy datasets and MNIST images, which is a pretty small and easy learning task.

3. How SWD is used during the training is not very clear, a systematic model framework or a detailed algorithm might be helpful.

4. The compared methods in the experiment only has abbreviation with no explanation.

5. How SWD is better is not clearly reflected in the experiments.

6. There are so many important things were discussed in the Appendix due to the page limit.

**Questions:**

1. What is the computational complexity for the proposed SWD?

2. What's the purpose of finding LR bound for SW Gradient Flow? How it impact the results for the mentioned learning tasks?

---

### Official Review · Reviewer_d93U · 2024-11-04

**Soundness:** 3
**Presentation:** 3
**Contribution:** 3
**Rating:** 8
**Confidence:** 3

**Summary:**

This paper revisits the classic Sliced-Wasserstein Distance (SWD) and proposes a new approach to enhance its effectiveness. The authors introduce a rescaling approach based on the "informativeness" of each slice, which they define as the slice's alignment with the data's effective subspace. Instead of modifying the slicing distribution to identify informative slices, this approach uses a single global scaling factor, which simplifies SWD calculations and also allows it to perform comparably with other variants.
The paper provides theoretical backing for the rescaling approach, showing how it aligns with different dimensionality settings and retains SWD's metric properties.
The authors validate this method through experiments on tasks like gradient flow, color transfer, and deep generative modeling, showing that a properly scaled SWD can match or even outperform other methods without added complexity.

**Strengths:**

The paper offers a simpler and more computationally efficient approach to improving SWD by introducing a single global rescaling factor based on slice informativeness.

The rescaling approach can be absorbed into the learning rate tuning process, making it easy to implement without extensive adjustments to existing models.

It also reduces dependency on strict dimensionality assumptions, making SWD adaptable to more tasks without requiring the knowledge of the data's effective subspace dimension k.

The authors provide a solid theoretical foundation for their approach.

**Weaknesses:**

I'm not an expert in this area, so I'll leave the weaknesses section blank for now.

**Questions:**

Why the strict subspace condition can be relaxed given Assumption 4.1 provides a theoretical basis?

What other criteria could be used to define informativeness, and would they offer any additional benefits over alignment with the effective subspace?

---

### Author Response · Authors · 2024-11-20
**Response to common questions and concerns**

Dear AC and Reviewers,

We would like to truly thank all reviewers for their valuable time and dedicated efforts to provide us with these constructive feedbacks. We are encouraged by the general endorsement of novelty from all reviewers. In this common section, we attempt to clarify our paper and provide some additional results, which we may reference in our detailed response to each reviewer. We will update the paper accordingly throughout our discussions.

---

# C1. The paper's primary contributions

**Problem**: In machine learning applications using sliced-Wasserstein, the current dominating paradigm in the literature is to design a slicing distribution that emphasizes the difference between input measures to address the projection complexity of the original Sliced-Wasserstein distances [1][2][3][4]. This often results in additional computational/memory overheads (especially for optimization-based methods [1][3]), numerical instabilities, and the loss of desirable theoretical properties (e.g., being a proper metric) [2][4].

**Proposed**: We propose a shift in perspective via the novel reweighting formulation. Instead of explicitly modifying the slicing distributions to emphasize informative slices, we rescale the contributions for all slices (e.g., less informative slices should be rescaled more). This new paradigm requires rethinking what it means for a projection to be *informative*. We discovered that this generalized notion can be dependent on prior knowledge about the data distribution. We show that, under a mild data assumption and an appropriate notion of informativeness, we can design the reweighting function to make all slices *equally informative*. This recovers the *subspace Sliced-Wasserstein* variant that is exactly a scalar multiple of the classical Sliced-Wasserstein distance between the input measures. This reduces the original problem to learning rate search (without any knowledge of the subspace in practice), which is already standard in the machine learning workflows. In the end, after an extensive search for the next SW variant, what we obtain is exactly the well-studied classical SWD with all theoretical properties preserved and no additional computational/memory overheads.

This naturally raises an important question: "Is Sliced-Wasserstein all you need for common learning tasks?". Here, by "common learning tasks", we refer to vision tasks that are often used in the existing SW literature [1][2][3][4][5]. There may exist less common settings with other data modalities (e.g., text, audio) that will not be the focus of our paper, although there are no known limitations restricting our results to vision tasks. Our choice of tasks simply allows direct comparison with existing methods within our computational budget.

Lastly, our proposed formulation provides novel insights and opens up interesting research directions to explore different data assumptions as well as different notions of slice informativeness.

# C2. The subspace assumption is reasonable in practice

Following the proposed reweighing formulation, we provide the subspace assumption upon which rest many of our theoretical contributions, Here, we aim to clarify that the effective subspace assumption is reasonable in practice because real datasets (images, audios, etc.) naturally have low-dimensional linear structure in two key ways. First, in minibatch processing, d-dimensional data of batch size $B \ll d$ mathematically cannot span more than min(2B, d), which provides a natural bound on the data effective dimensionality. Second, real features typically show strong linear correlations, as evidenced by the rapid eigenvalue decay in PCA across diverse datasets - practical success of PCA and autoencoders demonstrates that high-dimensional data can often be approximated in much lower-dimensional spaces while preserving essential structure for downstream tasks.

It is worth noting that our reweighting formulation does **not** depend on this assumption. One may make different assumptions based on prior knowledge of the data. In our paper, we find that using the subspace assumption leads to a conveninent notion of informativeness. This allows us to define the reweighting function that simplifies the math to a scalar multiple of the classical Sliced-Wasserstein in expectation. We note that this paradigm open up future research venues to explore other data assumptions and notions of informative slices, beyond the scope of our current paper.

Lastly, in section C4 below we show that our theoretical results remain relevant even when the data lives near a low-dimensional subspace (in parallel supported by extensive experimental results), addressing the comment of the reviewers about restrictive nature of our core assumption.

---

> ### Author Response · Authors · 2024-11-21
> **Response to common questions and concerns (cont)**
>
> # C3. The reweighting formulation recovers different Sliced-Wasserstein variants
>
> In our paper, we provide $2$ examples of Max-SW and Energy-based SW. Here, to supplement our response, we include additional instances, which will be added to the paper.
>
> - Max-SW [7], Markovian SW [3], and EBSW [2] implicitly use  $$\phi_{\mu,\nu}(\theta)=W_p^p(\theta_{\sharp} \mu,\theta_{\sharp} \nu)$$  to measure the informativeness of $\theta$. In other words, slices that have higher projected distances are considered more important/informative.
>
> - On the other hand, RPSW [4] implicitly uses $$\phi_{\mu, \nu}(\theta; \mu, \nu, \gamma_{\kappa}) = E_{(X,Y)\sim\mu \times \nu}\left[ \gamma_{\kappa}(\theta; P_{\mathbb{S}^{d-1}}(X - Y)) \right]$$ where $\gamma_\kappa$ is a location-scale distribution (i.e., von Mises-Fisher) and $P_{\mathbb{S}^{d-1}}$ is the projection onto $\mathbb{S}^{d-1}$. Slices that align well with random paths connecting $2$ distributions are considered more important/informative.
>
> For the above instances, one may also refer to $\phi_{\mu,\nu}$ as the **discriminant function**. However, note that we define the concept of *informativeness* more broadly, allowing for a broader set of assumptions about data structure rather than strictly referring to the ability to discriminate between distributions.
>
> For completeness, we show below how to recover different SW variants using our formulation with different notions of informative slices:
>
> **Note**: Since the \widetilde{SW} cannot be displayed, we resort to using \sim SW, referring to the rescaled sliced-Wasserstein.
>
> - Classical SWD [6]: We set $\phi\equiv 1$ and obtain the classical Sliced-Wasserstein distance $$\sim SW_p^p(\mu,\nu;\sigma, \rho_\phi) = SW_p^p(\mu,\nu;\sigma)$$
>
>
> - Max-SW [7] : We set $\phi_{\mu,\nu}(\theta)=W_p^p(\theta_\sharp\mu,\theta_\sharp\nu)$ and $\rho_\phi(r)=\delta_{r_{max}},\sigma=\mathcal{U}(\mathbb{S}^{d-1})$ where $r_{max}=\max\phi_{\mu,\nu}(\theta)$. Then, we have that $$\sim SW_p^p(\mu,\nu;\sigma,\rho_\phi)=Max\text{-}SW_p^p(\mu,\nu)$$
>
> - EBSW [2]: We set $\phi_{\mu,\nu}(\theta)=W_p^p(\theta_\sharp\mu,\theta_\sharp\nu)$, $\rho_\phi(r)=\frac{f(r)}{\int_{\mathbb{S}^{d-1}} f(W_p^p(\theta_\sharp\mu, \theta_\sharp\nu)) d\sigma(\theta)}$, $\sigma=\mathcal{U}(\mathbb{S}^{d-1})$ where $f : [0, \infty) \to (0, \infty)$ is an increasing energy function (e.g., $f(x) = e^x$). Then, we have that $$\sim SW_p^p(\mu,\nu;\sigma,\rho_\phi) = \int_{\mathbb{S}^{d-1}} \rho_\phi(\phi_{\mu,\nu}(\theta)) W_p^p(\theta_\sharp\mu,\theta_\sharp\nu) d\sigma(\theta) = \int_{\mathbb{S}^{d-1}} \frac{f(W_p^p(\theta_\sharp\mu,\theta_\sharp\nu))}{\int_{\mathbb{S}^{d-1}} f(W_p^p(\theta_\sharp\mu, \theta_\sharp\nu)) d\sigma(\theta)} W_p^p(\theta_\sharp\mu,\theta_\sharp\nu) d\sigma(\theta) \nonumber\\ =EBSW_p^p(\mu,\nu;f).$$
>
>
> - RPSW [4]: We set $\phi_{\mu,\nu}(\theta) = E_{(X,Y)\sim\mu\times\nu}[\gamma_\kappa(\theta; P_{S^{d-1}}(X - Y))]$,
> $\rho_\phi(r) = r$, $\sigma=\mathcal{U}(\mathbb{S}^{d-1})$, where $\gamma_{\kappa}$ is a location-scale distribution with parameter $\kappa$. Then, we have that $$\sim SW_p^p(\mu,\nu;\sigma,\rho_\phi) = \int_{\mathbb{S}^{d-1}} \phi_{\mu,\nu}(\theta)W_p^p(\theta_\sharp\mu, \theta_\sharp\nu) d\sigma(\theta) = E_{\theta\sim\phi_{\mu,\nu}(\cdot)}\left[W_p^p(\theta_\sharp\mu, \theta_\sharp\nu)\right] = RPSW_p^p(\mu,\nu;\gamma_\kappa)$$ where we abuse the notation $\phi_{\mu,\nu}$ to denote the distribution induced by function $\phi_{\mu,\nu}(\theta)$.
>
> - Exploring other notion of informative slices based on different data assumptions is a potential venue for further research.
>
> ---
>
> References:
>
> [1] Nguyen et al. "Distributional sliced-Wasserstein and applications to generative modeling." (2020)
>
> [2] Nguyen et al.. "Energy-based sliced wasserstein distance." (2024)
>
> [3] Nguyen et al. "Markovian sliced Wasserstein distances: Beyond independent projections." (2023)
>
> [4] Nguyen et al. "Sliced Wasserstein with random-path projecting directions." (2024)
>
> [5] Nadjahi et al. "Fast approximation of the sliced-Wasserstein distance using concentration of random projections." (2021)
>
> [6] Rabin et al. "Wasserstein barycenter and its application to texture mixing."  (2011)
>
> [7] Deshpande et al. "Max-sliced wasserstein distance and its use for gans." (2019)

---

> ### Author Response · Authors · 2024-11-21
> **Response to common questions and concerns (cont)**
>
> # C4. Relaxing the subspace assumption
>
> In this section, we discuss the situation where the subspace assumption does not strictly hold. Complete proofs will be updated in the paper.
>
> ## Proposition 1
> Let $U,U^\perp$ be defined in the subspace assumption, choose  $\mu^d,\nu^d\in P_p(\mathbb{R}^d)$, and let $\mu^k,\nu^k$ be defined by $U_\sharp\mu^d,U_\sharp\nu^d$,
> we claim the following:
> $$
> W_2^2(\mu^k,\nu^k)\leq W_2^2(\mu^d,\nu^d)\leq W_2^2(\mu^k,\nu^k)+2(m_2(U^\perp_\sharp\mu^d)+m_2(U^\perp_\sharp\nu^d,p))\tag{1}
> $$
> where $m_p(U^\perp_\sharp\mu^d)$ the $p$-th moment of measure $U^\perp_\sharp \mu^d$.
>
> ### Proof sketch for Proposition 1
> For each pair $(x,y)\in (\mathbb{R}^d)^2$, we have
> $$\|P_U(x)-P_U(y)\|^2 \leq \|x-y\|^2, \quad \text{(By definition of projection)}$$
>
> $$\|x-y\|^2 = \|P_U(x)-P_U(y)\|^2 + \|P_{U^\perp}(x)-P_{U^\perp}(y)\|^2, \quad \text{(Pythagorean theorem)}$$
>
> $$\|P_U(x)-P_U(y)\|^2 + \|P_{U^\perp}(x)-P_{U^\perp}(y)\|^2
> \leq \|P_U(x)-P_U(y)\|^2 + 2\|(U^\perp)^\top x\|^2 + 2\|(U^\perp)^\top y\|^2.$$
>
> Suppose $\gamma$ is an optimal transport plan for $W_2^2\left( (P_U)^\sharp \mu, (P_U)^\sharp \nu \right).$ By integrating both sides of the first inequality via $\gamma$, we prove the first inequality in (1): $$W_2^2(\mu^k, \nu^k) = W_2^2((P_U)^\sharp \mu, (P_U)^\sharp \nu) \leq W_2^2(\mu, \nu).$$
>
>
> Similarly, by integrating both sides of the second inequality via $\gamma$, we obtain the second inequality of (1).
>
> ## Proposition 2:
> $$
> \frac{k}{d} SW_2^2(\mu^k, \nu^k) \leq SW_2^2(\mu^d, \nu^d)
> \leq \frac{k}{d} SW_2^2(\mu^k, \nu^k) + 2 \frac{d-k}{d} \big( m_2(U^\perp_\sharp \mu^d) + m_2(U^\perp_\sharp \nu^d) \big)
> $$
>
> ### Proof sketch for Proposition 2
>
> Pick $\theta \in \mathbb{S}^{d-1}$ and $x, y \in \mathbb{R}^d$. We have that $$\|\theta^\top P_U(x) - \theta^\top P_U(y)\|^2 \leq \|\theta^\top x - \theta^\top y\|^2$$
>
> $$\|\theta^\top x - \theta^\top y\|^2 = \|P_U(\theta)^\top P_U(x) - P_U(\theta)^\top P_U(y)\|^2 + \|P_{U^\perp}(\theta)^\top P_{U^\perp}(x) - P_{U^\perp}(\theta)^\top P_{U^\perp}(y)\|^2$$
>
> $$\|P_U(\theta)^\top P_U(x) - P_U(\theta)^\top P_U(y)\|^2 + \|(U^\perp)^\top \theta\|^2 \|(U^\perp)^\top x - (U^\perp)^\top y\|^2, \quad \text{(Cauchy-Schwarz inequality)}$$
>
> $$\|\theta^\top P_U(x) - \theta^\top P_U(y)\|^2 + 2\|(U^\perp)^\top \theta\|^2 \big(\|(U^\perp)^\top x\|^2 + \|(U^\perp)^\top y\|^2\big).$$
>
> Similar to the proof of Proposition 1, we obtain:
>
> Choose $\theta \in \mathbb{S}^{d-1}$. From Proposition 1, we have that:
> $$W_2^2\big(\theta_\sharp (P_U)^\sharp \mu^d, \theta_\sharp (P_U)^\sharp \nu^d\big) \leq W_2^2\big(\theta_\sharp \mu^d, \theta_\sharp \nu^d\big)$$
>
> $$W_2^2\big(\theta_\sharp \mu^d, \theta_\sharp \nu^d\big) \leq W_2^2\big(\theta_\sharp (P_U)^\sharp \mu^d, \theta_\sharp (P_U)^\sharp \nu^d\big) + 2 \big(m_2(U^\perp_\sharp \mu^d) + m_2(U^\perp_\sharp \nu^d)\big).$$
>
> Take the expected value with respect to $\theta \sim \mathcal{U}(\mathbb{S}^{d-1})$, we complete the proof.

---

> ### Author Response · Authors · 2024-11-21
> **Response to common questions and concerns (cont)**
>
> # C5. Clarifying the global scaling factor
>
> Recall that in Proposition 4.4 in the paper, we have that:
> $$W_p^p(\theta^{d^\sharp} \mu^d, \theta^{d^\sharp} \nu^d) = W_p^p\big((U^\top \theta^d)^\sharp \mu^k, (U^\top \theta^d)^\sharp \nu^k\big)  = \|U^\top \theta^d\|^p W_p^p(\theta^{k^\sharp} \mu^k, \theta^{k^\sharp} \nu^k).$$
>
> where $\theta^k=\frac{U^\top\theta^d}{\|U^\top\theta^d\|}$ with convention $\theta^k=0_k$ if $\|U^\top \theta^d\|=0$.
>
> If we integrate both sides over $\theta^d \in \mathbb{S}^{d-1}$ with respect to the uniform measure $\sigma(\theta^d)$, then we obtain
>
> \begin{equation}
> SW_p^p(\mu^d, \nu^d) = \int_{\mathbb{S}^{d-1}} W_p^p(\theta^d_\sharp \mu^d, \theta^d_\sharp \nu^d) d\sigma(\theta^d) \ = \int_{\mathbb{S}^{d-1}} \| U^\top \theta^d \|^p  W_p^p\left( \theta^k_\sharp \mu^k, \theta^k_\sharp \nu^k \right)  d\sigma(\theta^d). \end{equation}
>
> Note that $\theta^k$ depends on $\theta^d$, and the distribution of $\theta^k$ induced by $\theta^d \sim \sigma$ is uniform over $S^{k-1}$. We introduce the change of variables from $\theta^d$ to $\theta^k$ and express the integral in terms of $\theta^k$:
> \begin{equation}
> SW_p^p(\mu^d, \nu^d) = \int_{\mathbb{S}^{k-1}} W_p^p\left( \theta^k_{\sharp} \mu^k, \theta^k_{\sharp} \nu^k \right) \left( \int_{\theta^d: \frac{U^\top \theta^d}{\| U^\top \theta^d \|} = \theta^k} \| U^\top \theta^d \|^p \, d\sigma(\theta^d|\theta^k) \right) dT_\sharp \sigma(\theta^k),
> \end{equation}
> where $\sigma(\cdot|\theta^k)$ is the conditional distribution of $\theta^d$ under $\theta^k$, and $T:x\mapsto \frac{U^\top x}{\|U^\top x\|}$ is the mapping from $\theta^d$ to $\theta^k$.
>
> The inner integral over $\theta^d$ can be evaluated as a scaling factor $C_{d,k}$ dependent on $\sigma$, $\theta^k$, $U$. When $\sigma=\mathcal{U}(\mathbb{S}^{d-1})$, $C_{d,k}$ is invariant for all $\theta^k$.
>
> Substituting back into  and let $\sigma_k=T_\sharp\sigma=\mathcal{U}(\mathbb{S}^{k-1})$ denote the distribution of $\theta^k$, we obtain
> \begin{equation}
> SW_p^p(\mu^d, \nu^d) = C_{d,k} \int_{\mathbb{S}^{k-1}} W_p^p\left( \theta^k_{\sharp} \mu^k, \theta^k_{\sharp} \nu^k \right) \, d\sigma_k(\theta^k).
> \end{equation}
>
> Since $\sigma_k(\theta^k)$ integrates to 1 over $S^{k-1}$, and $W_p^p\left( \theta^k_{\sharp} \mu^k, \theta^k_{\sharp} \nu^k \right)$ is integrated over all $\theta^k$, we can express the right-hand side as $C_{d,k} \cdot SW_p^p(\mu^k, \nu^k; \sigma_k)$. Intuitively speaking, this means the \textit{loss of information} is due to an implicit constant factor on $SW_p^p(\mu^d, \nu^d)$, which we denote as the \textbf{Effective Subspace Scaling Factor} (ESSF). Thus, rescaling the one-dimensional Wasserstein for all slices becomes multiplying the SWD by the reciprocal of the ESSF. The main theorem 4.5 in the paper makes this connection explicit.

---

> ### Author Response · Authors · 2024-11-21
> **Response to common questions and concerns (cont)**
>
> # C6. Additional results in the empirical case
>
> In practice, we have access to finite samples and a limited number of slices. We provide the following proposition to extend our results to this practical setting.
>
> ## Proposition 3
>
> Let $\mu^d, \nu^d \in \mathcal{P}(\mathbb{R}^d)$ satisfy Assumption~\ref{asp:lowdsupp}. Consider the empirical estimator $\widehat{ESSF}(L)$ defined as:
>
> \begin{equation}
>     \widehat{ESSF}(L) = \frac{1}{L}\sum_{l=1}^L \| U^{\top} \theta_l^d \|^p,
> \end{equation}
>
> where $\{\theta_l^d\}_{l=1}^L \overset{\text{i.i.d.}}{\sim} \mathcal{U}(\mathbb{S}^{d-1})$. Then:
>
>
> - $\mathbb{E}[\widehat{ESSF}(L)]=\frac{C_k}{C_d}$ and $\text{Var}(\widehat{ESSF}(L))= \mathcal{O}(\frac{1}{L})$
>
> - Let $\epsilon_L = \left| SW_p^p\left(\mu^d, \nu^d; \frac{1}{L}\sum_{l=1}^L \delta_{\theta_l^d}\right) - \widehat{ESSF}(L) \cdot SW_p^p\left(\mu^k, \nu^k; \frac{1}{L}\sum_{l=1}^L \delta_{\theta_l^k}\right) \right|$. Then $\epsilon_L \xrightarrow{\text{a.s.}} 0$ as $L \to \infty$
>
> - Furthermore, there exists a constant $K > 0$ depending only on $\mu^d$ and $\nu^d$ such that for any $\delta > 0$, we have that
> $$\mathbb{P}(\epsilon_L < \delta) \geq 1 - e^{-\delta^2L/K^2}$$
>
> ### Proof sketch for Proposition 3
>
> - It is straight forward to prove $\|U^\top\theta_l^d\|^2\sim Beta(\frac{k}{2},\frac{d-k}{2})$. Thus $\hat{ESSF(L)}$ is essentially of sample mean of $L$ i.i.d Beta distribution variables. We can derive its expected value and variance.
>
> - It is straight forward to derive
> \begin{equation}
> \epsilon_L \leq K\cdot \left|\frac{1}{L}\sum_{l=1}^L\left(\|U^\top \theta_l\|^p-\sum_{l'=1}^L\|U^\top\theta_{l'}\|^p\right)\right|
> \end{equation}
> where $K=\max_{x,y}\|x-y\|^p$.
> Let's denote the sum term as $B_n$. By law of large numbers, with probability 1, $B_n\to 0$. Thus, $\epsilon_L\to 0$.
> - It remains to show the convergence rate of $\epsilon_L$. Since each $\|U^\top\theta_l\|^p\in[0,1]$, for each $t>0$, by Hoeffding's we have
> \begin{equation}
> \mathbb{P}(|B_n|\ge \epsilon)\leq e^{-2\delta^2L}
> \end{equation}
> Replacing $\delta$ by $\delta/K$, we have $\mathbb{P}(\epsilon_L\leq \delta)\ge 1-2 e^{\frac{2\delta^2L}{K^2}}$ and we complete the proof.
>
>
> ## Proposition 4
>
> Consider discrete measures $\mu=\sum_{i=1}^nq_i^1\delta_{x_i},\nu=\sum_{j=1}^mq_i^2\delta_{y_j}$.
> Define the empirical gradient error for each $x_i$:
>
> $$\delta_d = \sum_{l=1}^L\delta_l^d$$
>
> $$\delta_k = \sum_{l=1}^L\delta_l^k$$
>
> $$G_d = \nabla_{x_i} SW_p^p(\hat{\mu}_d,\hat{\nu}_d;\delta_d)$$
>
> $$G_k = \nabla_{x_i} SW_p^p(\hat{\mu}_k,\hat{\nu}_k;\delta_k)$$
>
> $$\epsilon_L(x_i) = \|G_d - \widehat{ESSF}(L) \cdot G_k\|$$
>
> Then the following holds
> - $\epsilon_L(x_i) \xrightarrow{\mathbb{P}} 0$ as $L \to \infty$
>
> - $\mathbb{P}(\|\epsilon_L(x_i)\| \leq \epsilon) \geq 1-2e^{-\epsilon^2L/(pq^1_iK)^2}$,
> where $K=\max_{x_i,y_j}\|x_i-y_j\|^{p-1}<\infty$.
>
> ### Proof sketch for Proposition 4
>
> - Similar to previous proof, we can show
> $$\|\epsilon_L(x_i)\|\leq pq_1^1 K |B_L|$$
> where $K=\max_{x_i,y_j}\|x_i-y_j\|^p-1$, $B_L=\frac{1}{L}\sum_{l=1}^L(\|U^\top \theta_l\|^p-\frac{1}{L}\sum_{l=1}^L\|U^\top \theta_l\|^p)$.
> By law of large number, with probability 1,
>  $B_L\to 0$ as $L\to \infty$.
>  Thus, $\epsilon_L\to 0$.
> - Since $\|U^\top \theta\|\in[0,1]$, by Hoeffding inequality,
> $$\mathbb{P}(|B_L|\ge \epsilon)\leq 2e^{-\epsilon^2/L},$$
> thus we prove the inequality in the statement.

---

### Author Response · Authors · 2024-11-25
**Engaging in Final Discussions and Requesting Reevaluations as the Discussion Period Closes**

Dear Reviewers,

Thank you once again for your time and dedication to maintaining the high standards of the ICLR conference. As we near the end of the discussion period, we would like to check in and engage in constructive dialogue to address any remaining concerns you may have.

We have provided detailed responses to your thoughtful feedback and revised the paper according to your constructive comments. We would be happy to engage with you further to clarify or expand on any points. Please let us know if there are additional questions we can address.

Best, The Authors

---

### Meta-Review · Area_Chair_qJrf · 2024-12-19

**Metareview:**

This paper revisits Sliced-Wasserstein Distances (SWDs) and addresses their relative inefficiency in high dimensions by introducing a simple global rescaling factor to enhance slice informativeness while preserving theoretical properties. The method avoids the complexity of existing SWD variants and is aimed at improving their applicability in gradient-based learning workflows. Experiments demonstrate that their variant of SWD achieves competitive  performance compared to more complex alternatives in some machine learning tasks, mostly generative.

Through the analysis of the reviews and the discussion, I find that the paper is still lacking decisive arguments both theoretical and experimental to make this variant of SWD stand out with respect to all the sliced variants of Wasserstein, and I will mostly follow reviewer  TZmm. In order to explain further my decision, I will detail some of those limitations:
  - limited theoretical insights: the subspace assumption is insufficiently described. Assumption 4.1 assumes that both target ad source distributions live both in the same subspace, which can be perceived as a strong limitation. Equivalences with other SWD variants are not very informative.
  - the evaluation is somehow limited, and mostly focused on SWD variants. For equivalent batch sizes, why not considering directly Wasserstein ?  In the color transfer example, the idea of limited subspace of dimension < 3 is not discussed. In the case of FFHQ: the considered dimension is the one from the latent space (512). The metrics used for comparing the different variants of SWD are not not discussed, nor the results are analyzed. Results presented in the appendix should be chosen more carefully, and do not aim for exhaustivity (tables 5 to 18)

I encourage the authors to sharpen their arguments on why their variant of SWD should be preferred to the classical sliced Wasserstein, and to design experiments that clearly show benefits.

**Additional Comments On Reviewer Discussion:**

I mostly considered review from TZmm, and weighed down the first two reviews, as I did not find them very informative.

---

### Decision · Program_Chairs · 2025-01-22

Reject